# Statistics of chromatin organization during cell differentiation revealed by heterogeneous cross-linked polymers

O. Shukron[1], V. Piras [2], D. Noordermeer [2] & D. Holcman [1]

Chromatin of mammalian nucleus folds into discrete contact enriched regions such as Topologically Associating Domains (TADs). Folding hierarchy and internal organization of TADs is highly dynamic throughout cellular differentiation, and are correlated with gene activation and silencing. To account for multiple interacting TADs, we developed a parsimonious randomly cross-linked (RCL) polymer model that maps high frequency Hi-C encounters within and between TADs into direct loci interactions using cross-links at a given base-pair resolution. We reconstruct three TADs of the mammalian X chromosome for three stages of differentiation. We compute the radius of gyration of TADs and the encounter probability between genomic segments. We found 1) a synchronous compaction and decompaction of TADs throughout differentiation and 2) high order organization into meta-TADs resulting from weak inter-TAD interactions. Finally, the present framework allows to infer transient properties of the chromatin from steady-state statistics embedded in the Hi-C/5C data.

[1] Group of data modeling, computational Biology and predictive medicine, Ecole Normale Supérieure, 46 rue d'Ulm, 75005 Paris, France. [2] Chromatin Dynamics Group, Institute for Integrative Biology of the Cell (I2BC), CEA, CNRS, Université Paris-sud, University Paris-Saclay, 1 avenue de la Terrasse, 91198 Gif-sur-Yvette, France. Correspondence and requests for materials should be addressed to D.H. (email: david.holcman@ens.fr)

Mammalian chromosomes fold into discrete mega-basepairs (Mbp) contact enriched regions termed topologically associating domains (TADs). Although the precise role of TADs remain unclear, they participate in synchronous gene regulation[1,2] and replication[3]. Gene regulation within TAD is modulated by transient loops at sub-TAD scale[1,4–6], formed by regulatory elements such as enhancers and promoters[7]. However, sparse connectors between TADs (at a scale >Mbps) can significantly influence chromatin dynamics and gene regulation within TADs[8,9]. We present and apply here methodology based on polymer model that accounts for interactions between TADs to study chromatin dynamic at long scales, an area that remains largely unexplored.

Genome organization can be probed by chromosome conformation capture (CC) methods[2,4,10], which report simultaneous genomic contacts (loops) at scales of kilo bps (kbps) to Mbps. TADs appear as matrix block in the contact maps by summing encounter events over an ensemble of millions of cells[1,10]. Conformation capture contact maps provide a statistical summary of steady-state looping frequencies, but do not contain neither direct information about the size of the folded genomic section nor any transient genomic encounter times[11]. Throughout cell differentiation stages, the boundaries between TADs remain stable[1,12], but their internal looping pattern is highly variable. Moreover, TADs can form hierarchy structures into meta TADs, formed by inter TAD connectivity, correlated with transcription state of the chromatin[12].

Coarse-grained polymer models are used to study steady-state and transient properties of chromatin at a given scale. Starting with the Rouse polymer[13], composed of $N$ monomers connected sequentially by harmonic springs, the linear connectivity in the Rouse model does not account for the complexity of molecular interactions and therefore cannot account for contact enriched regions such as TADs. To include short and long-range looping, other polymer models have been developed, with self-avoiding interactions[11,14–22], random loops[8,15,23,24], or epigenomic state[25,26], that in addition account for TADs.

To account for interactions between multiple TADs, we develop a parsimonious polymer model of the chromatin. We extend the construction of the randomly cross-linked (RCL) polymer model, introduced in refs. [9,11], and account for multiple connected TADs of various sizes and connectivity within and between TADs. Random cross-links can be generated by binding molecules such as CTCF[1,27] or by loop extrusion mechanism[28], although the exact mechanism by which they form is not really important for the present work. The random positions of the cross-links capture the heterogeneity of chromatin structure sampled in a large ensemble of cells.

We then introduce our methodology to construct an RCL polymer from the empirical encounter probability (EP) extracted from 5C or Hi-C data, we determine their geometrical organization and characterize their distribution in space by computing the volume of TADs using the mean radius of gyration (MRG). We further investigate the role of several parameters such as the number and positions of the cross-links or the polymer length. We show that TADs and higher-order genome organization are modulated by the inter-TAD connectivities. These connectivities determine the steady-state and time-dependent statistical properties of monomers within each TAD. We validate the present method by comparing Hi-C and 5C data and the reconstructed statistics between two 5C replicas. Finally, we study the reorganization of three neighboring TADs on the mammalian X chromosome throughout stages of cellular differentiation. The key step of our method was to derive an expression for the EP of the RCL polymer model (see Eqs. (3) and (5)), this step was a bottleneck when fitting the empirical 5C data[1]. We then determine the rate of compaction and decompaction of TADs that can be correlated to gene silencing and activation, respectively. These properties cannot be studied by a direct comparison of the overall number of interactions within and between TADs from 5C data, because individual realizations cannot be discerned from the ensemble. To conclude, the present framework allows to study chromatin dynamics from Hi-C or 5C data, to derive its statistical properties, to connect steady-state Hi-C statistics with time-dependent analysis of single locus trajectories and to reveal the influence of multiple connected TADs on each other and genome reorganization throughout cellular differentiation.

## Results

**Statistical properties of heterogeneous RCL.** We will study the statistical properties of the heterogeneous RCL polymer, representing multiple TADs using first the mean-square-radius of gyration (MSRG), second the EP in two cases: between monomers of the same TAD (intra TAD) and across TADs (inter-TAD), third, mean-square-displacement (MSD) of single monomers, and fourth, the distribution of distance between any two monomers. Details are given in Supplementary Methods.

(1) The MSRG $\langle R_g^2 \rangle$ characterizes the folding of a TAD inside a ball of radius $\sqrt{\langle R_g^2 \rangle}$. When the condition of a dominant intra-TAD connectivity (assumption $H_1$, Supplementary Methods Equation 29) is satisfied, the MSRG for $\text{TAD}_i$ (see derivation in Supplementary Methods Equation 24–34) is given by

$$\left\langle R_g^2 \right\rangle^{(i)} \approx \frac{b^2}{(1 - \xi_{ii})(\zeta_0^{(i)}(\Xi) - \zeta_1^{(i)}(\Xi))}, \qquad (1)$$

where

$$\begin{aligned}
\zeta_0^{(i)}(\Xi) &= y^{(i)}(\Xi) + \sqrt{y^{(i)}(\Xi)^2 - 1}, \\
\zeta_1^{(i)}(\Xi) &= y^{(i)}(\Xi) - \sqrt{y^{(i)}(\Xi)^2 - 1},
\end{aligned} \qquad (2)$$

$y^{(i)}(\Xi) = 1 + \frac{\sum_{k=1}^{N_T} \xi_{ik} N_k}{2(1 - \xi_{ii})}$, and $\xi_{ij}$ is the connectivity matrix defined in Eq. (14).

(2) Under the condition of non-vanishing connectivities (Supplementary Methods Equation 50), the EP between monomer $m$ and $n$ within TAD $A_i$ (see derivation in Supplementary Methods) is given by

$$P_{m^{(i)}, n^{(i)}}(\Xi) \propto \left( \frac{d}{2\pi \sigma_{m,n}^2(\Xi)} \right)^{d/2}, \qquad (3)$$

where

$$\sigma_{m,n}^2(\Xi) = \begin{cases} \left\langle R_g^2 \right\rangle^{(i)} \left( \frac{(\zeta_0^{(i)}(\Xi)^{m-n} - 1)^2 - 2\zeta_0^{(i)}(\Xi)^{m+n-1}}{\zeta_0^{(i)}(\Xi)^{2m-1}} + 2 \right), & m \geq n; \\ \left\langle R_g^2 \right\rangle^{(i)} \left( \frac{(\zeta_0^{(i)}(\Xi)^{n-m} - 1)^2 - 2\zeta_0^{(i)}(\Xi)^{m+n-1}}{\zeta_0^{(i)}(\Xi)^{2n-1}} + 2 \right), & m < n, \end{cases} \qquad (4)$$

and $\left\langle R_g^2 \right\rangle^{(i)}$ is the MSRG of TAD $A_i$, defined by relation (1). When monomers belong to distinct TADs $i$ and $j$ ($i \neq j$), the EP formula is modified to (see Supplementary Methods subsection Encounter probability of monomers of the heterogeneous RCL polymer)

$$P_{m^{(i)}, n^{(j)}}(\Xi) \propto \left( \frac{d}{2\pi \sigma_{mn}^2(\Xi)} \right)^{d/2}, \qquad (5)$$

where

$$\sigma_{mn}^2(\Xi)$$
$$= \left\langle R_g^2 \right\rangle^{(i)} (1 + \zeta_0^{(i)}(\Xi)^{1-2m}) + \left\langle R_g^2 \right\rangle^{(j)} (1 + \zeta_0^{(i)}(\Xi)^{1-2n})$$
$$+ b^2 \left( \frac{1}{N_i \sum_{k \neq i}^{N_T} N_k \xi_{ik}} + \frac{1}{N_j \sum_{k \neq j}^{N_T} N_k \xi_{jk}} \right).$$

$$(6)$$

(3) The MSD of a monomer $r_m^{(i)}$ located inside $A_i$ for intermediate times (see Supplementary Methods subsection Mean-square displacement of monomers of the heterogeneous RCL polymer) is given by

$$\langle \langle (r_m^{(i)}(t+s) - r_m^{(i)}(s))^2 \rangle \rangle \approx 2dD_{cm}t + \frac{db^2 Erf\left[ \sqrt{2dDt \sum_{k=1}^{N_T} \frac{N_k \xi_{ik}}{b^2}} \right]}{2\sqrt{(1 - \xi_{ii}) \sum_{k=1}^{N_T} N_k \xi_{ik}}},$$

$$(7)$$

where $D_{cm} = \frac{D}{\sum_{k=1}^{N_T} N_k}$ and $Erf$ is the Gauss error function.

(4) The distribution $f_{D_{mn}}(x)$ of the distance $D_{mn} = \| r_m - r_n \|$ between any two monomers $r_m$ and $r_n$ (see Supplementary Methods subsection Distribution of the distance between monomers of the RCL polymer) is given by

$$f_{D_{mn}}(x) = \frac{2\Gamma\left(\frac{d}{2}\right)}{2^{\frac{d}{2}}} \left( \frac{x}{\sigma_{mn}(\Xi)} \right)^{d-1} e^{-\left( \frac{x}{\sqrt{2}\sigma_{mn}(\Xi)} \right)^2},$$

$$(8)$$

where $\Gamma$ is the $\Gamma$-function.

To validate formulas (1)–(7) so that we can use them to extract statistical properties of 5C/Hi-C data, we decided to test them against numerical simulations of three synthetic interacting TADs. We constructed a RCL polymer containing three TADs with $N_1 = 50$, $N_2 = 40$, $N_3 = 60$ total monomers, so that condition $H_1$ (Supplementary Methods Equation 29) about dominant intra-connectivity is satisfied. We impose the number of connectors in each TAD to be at least twice compared to the one between TADs (see Fig. 1a).

To construct the encounter frequency matrix, we simulated Eq. (17) in dimension $d = 3$, with $b = 0.2 \,\mu m$ and diffusion coefficient $D = 8 \times 10^{-3} \,\mu m^2 \, s^{-1}$ [29] starting with a random walk initial polymer configuration. Connectors were placed between monomers with a uniform probability in each $TAD_i$ and in between TADs, as indicated in Fig. 1a. We ran 10,000 simulations until polymer relaxation time (see Supplementary Methods Equation 23 and ref. [11]). The longest relaxation time of RCL chains containing $N_T$ TADs is defined by tens of thousands of simulation steps. At the end of each realization, we collected the monomer encounters falling below the distance $\epsilon = 40 \, nm$, and constructed the simulation encounter frequency matrix. This matrix shows three distinct diagonal blocks (Fig. 1a) resulting from high intra-TAD connectivity, and further reveals a high-order organization (cyan blue in Fig. 1a), which resembles the meta-TADs discussed in ref. [12]. We thus propose here that hierarchical TAD organization is a consequence of weak inter-connectivity properties.

We then computed the steady-state EP from the simulation encounter frequency matrix (Fig. 1a) by dividing each row with its sum. We then compared simulations and theoretical EPs (Eqs. (3) and (5)) in Fig. 1b: the three sample curves for monomer $r_{20}$ (upper left), $r_{70}$ (upper right), and $r_{120}$ (lower) located in the middle of each TAD, are in good agreement with the theory. Furthermore, the theoretical and simulated EPs for monomer $r_1$, $r_{51}$, and $r_{91}$, located at boundaries of TADs (Fig. 1b, bottom) are in good agreement. Finally, we computed the MRG $\bar{R}_g = \sqrt{\langle R_g^2 \rangle}$,

for $TAD_1$, $TAD_2$, and $TAD_3$ given by 0.177, 0.13, 0.165 μm (simulations), compared to 0.178, 0.13, 0.167, respectively, obtained from expression (1), which agree.

To validate the MSD expression (Eq. (7)), we simulated Eq. (17) for 2500 steps with a time step $\Delta t = 0.01 \, s$, past the relaxation time $\tau(\Xi)$ (Supplementary Methods Equation 29) and computed the average MSD over all monomers in each $TAD_i$, $i = 1, 2, 3$. In Fig. 1c, we plotted the average MSD in each TAD against expression (Eq. (7) (dashed)), which are in good agreement. The overshoot of the MSD of $TAD_1$, results from the weak coupling of centers of masses of TADs (see Supplementary Methods Equation 22). The amplitude of the MSD curve is inversely proportional to the total connectivity of each TAD as shown in Fig. 1c, $TAD_1$ (blue, 26 connectors), $TAD_2$ (red, 44 connectors), and $TAD_3$ (yellow, 37 connectors). We conclude, that the present approach (numerical and theoretical) capture the steady-state properties (Eqs. (1), (3)–(5), (7)) of multi-TAD.

In addition, we found that adding an exclusion forces with a radius of 40 nm did not lead to any modifications of the statistical quantities defined above (see Supplementary Fig. 3 compared to Fig. 1c). However, when the exclusion radius increases to 67 nm, deviations started to appear (Supplementary Fig. 4). To conclude, an exclusion radius of the order of 40 nm, also used in ref. [30], is consistent with the physical crowding properties of condensin and cohesin[31] to fold and unfold chromatin. Thus, using the present RCL polymer models we will now reconstruct statistical properties of chromatin in different cellular differentiation phases.

**Reconstructing genome reorganization during cell differentiation.** To extract chromatin statistical properties, we constructed systematically an RCL model from 5C data of the X chromosome[1]. We focus on the chromatin organization during three stages of differentiation: undifferentiated mouse embryonic stem cells (mESC), neuronal precursor cells (NPC), and mouse embryonic fibroblasts (MEF). We first used the average of two replica of a subset of 5C data generated in ref. [1], and then each replica separately. The two replica harbor three TADs: TAD D, E, and F, which span a genomic section of about 1.9 Mbp. We coarse-grained the 5C encounter frequency data at a scale of 6 kb (Fig. 2a, upper), which is twice the median length of the restriction segments of the HindII enzyme used in producing the 5C data[1,8,9]. At this scale, we found that long-range persistent peaks of the 5C encounter data are sufficiently smoothed out to be able to use expressions (3) and (5) for fitting the 5C EP using standard norm minimization procedure. The result is a coarse-grained encounter frequency matrix that includes pairwise encounter data of 302 equally-sized genomic segments. To determine the position of TAD boundaries, we mapped the TAD boundaries reported in bps (see ref. [1]) to genomic segments after coarse-graining. We then constructed a heterogeneous RCL polymer with $N_D = 62$, $N_E = 88$, $N_F = 152$ monomers for TAD D, E, and F, respectively. To compute the minimum number of connectors within and between TADs, we fitted the EP of each monomer in the coarse-grained empirical EP matrix using formulas (3) and (5). In Fig. 2a (bottom), we present the fitted EP matrices for mESC (left), NPC (middle), and MEF (right).

We computed the number of connectors $Nc$, within and between TADs, by averaging the connectivity values $\xi_m$ for monomers in each TAD obtained from fitting the EP (Eqs. (3) and (5)) to all 302 monomers. After averaging we obtained the connectivity matrix $\Xi$, and used it in relations (Eqs. (13) and (14)) to recover the number of connectors within and between TADs. The mean number of connectors in the differentiation from

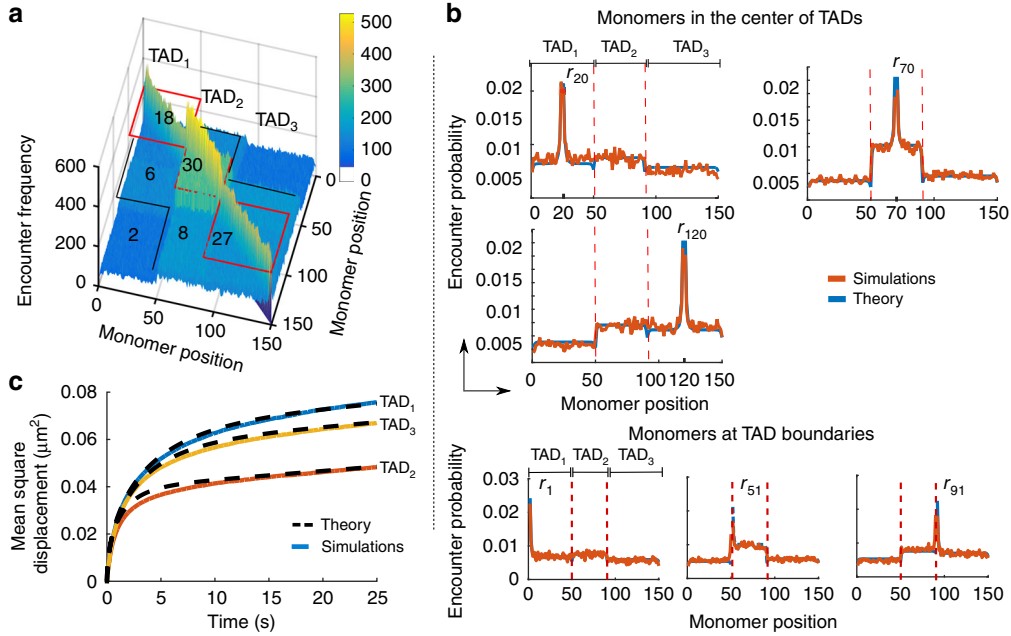

**Fig. 1** Statistical properties of the heterogeneous RCL polymer. **a** Encounter frequency matrix of a polymer with three TAD blocks (TAD₁, TAD₂, TAD₃) of $N_1 = 50$, $N_2 = 40$, $N_3 = 60$ monomers, computed from 10,000 simulations of Eq. (17) with $\Delta t = 0.01$ s, $D = 8 \times 10^{-3}$ μm²/s, $d = 3$, $b = 0.2$ μm[1]. The number of added connectors appears in each block. Three distinct diagonal TADs are visible (red boxes) where secondary structure appears (black lines) due to weak inter-TAD connectivity. **b** Encounter probability (EP) of the heterogeneous RCL described in panel (**a**), where the simulated (orange) and theoretical (blue, Eqs. (3) and (5)) EP agree. In the upper panel, we plotted the EP for middle monomers in each TAD: monomer $r_{20}$ (upper left), monomer $r_{70}$ (top right), and monomer $r_{120}$ (bottom left). In the bottom panel, we plotted the simulated (orange) vs. theoretical EPs (blue) for monomers $r_1$ (left), $r_{51}$ (center), and $r_{91}$ (right), located at TAD boundaries. **c** Average mean-square displacement of monomers in each TAD of the heterogeneous RCL polymer, simulated as described in panel (**a**), simulations (continuous) vs. theory (dashed, Eq. (7)) until $t = 25$ s

mESC to NPC showed an increase by 145–200% (Fig. 2b) within and between TADs. The number of connectors within TAD F increases by 145% from 22 for mESC to 31 in NPC, the inter-connectivity between TAD F and E increases by 150% from 9 at mESC to 14 for NPC and the connectivity between TAD F and D doubled from 6 connectors for mESC to 12 for NPC, whereas the number of connectors within TAD E remained constant of 15. In MEF stage the number of connectors within TADs D, E, and F returned to values comparable to mESC, whereas the inter-TAD connectivity between TAD F and E was 9 for MEF.

To evaluate the size of the folded TADs, we computed the MRG of the three TADs throughout differentiation stages. The MRG is the square root of the MSRG (Eq. (1)) for each TAD using the calibrated connectivity matrix Ξ, obtained from fitting the experimental 5C EP (Fig. 2c, left). We found that the MRG can both increase and decrease depending on the number of connectors within TADs, but is also affected by inter-TAD connectivity, as revealed by Eq. (1). The MRG of all TADs decreased in average from 0.21 μm at mESC stage to 0.19 μm for NPC and was 0.2 μm for MEF cells. From mESC to NPC, the MRG of TAD E exceeded that of TAD D (Fig. 2c, red squares) despite the higher numbers of added connectors in TAD D and its smaller size $N_D = 66$ compared to $N_E = 88$. This result shows how the inter-TAD connectivity contributes to determine the MRG and the volume of TADs. In addition, using the calibrated RCL model at 3 kb, we were able to reproduce the distributions of three-dimensional distances between seven DNA FISH probes reported in ref. [8] (Supplementary Fig. 2).

Finally, we recall that a ball having a radius of gyration is insufficient to characterize the degree of compaction inside a TAD, because it does not give the density of bps per nm³. To obtain a better characterization of chromatin compaction,

we use the compaction ratio for TAD$_i$, defined by the ratio of volumes:

$$C_r^i = \left( \frac{\left\langle R_g^2 \right\rangle^{(i)}}{N_i b^2 / 6} \right)^{3/2} \tag{9}$$

where $\left\langle R_g^2 \right\rangle^{(i)}$ is given by formula (1) and the denominator is the MSRG for a linear Rouse chain of size $N_i$[13]. We find that TAD F ($N_F = 154$ monomers) has the highest compaction ratio among all the TADs among the three stages of differentiation (Fig. 2c, circles): indeed for TAD F, $C_r^F = 91, 135$, and 97 fold more compact than the linear Rouse chain with $N = 150$ monomers, associated with mESC, NPC, and MEF stages, respectively. For TAD E, ($N_E = 88$ monomers) the compaction ratio is 51, 66, and 45, thus it is more compact than the linear Rouse chain with $N = 88$ monomers (Fig. 2c right, red squares), despite retaining 15 intra-TAD connectors in all stages of differentiation (panel b). This effect is due to an increased inter-TAD connectivities between TAD E and F at NPC stage to 15. Finally, TAD D ($N = 62$ monomers), characterized by $C_r^D = 28, 44$, and 35 (blue diamonds) is more compact than a Rouse chain of $N = 62$ monomers, for mESC, NPC, and MEF stages, respectively.

To examine the consistency of our approach and the ability of RCL model to represent chromatin, we fitted independently the EPs $P^{(1)}$, $P^{(2)}$ of the 5C data of replica 1 and 2 at 10 kb resolution (Supplementary Fig. 5A–C) using Eqs. (3)–(5) (Methods). We found that the number of added connectors in replica 1 and 2 differs by at most five connectors for TAD F. This difference between replica may arise from intrinsic fluctuations in the statistics of encounter frequencies. We further compared the EP $P$

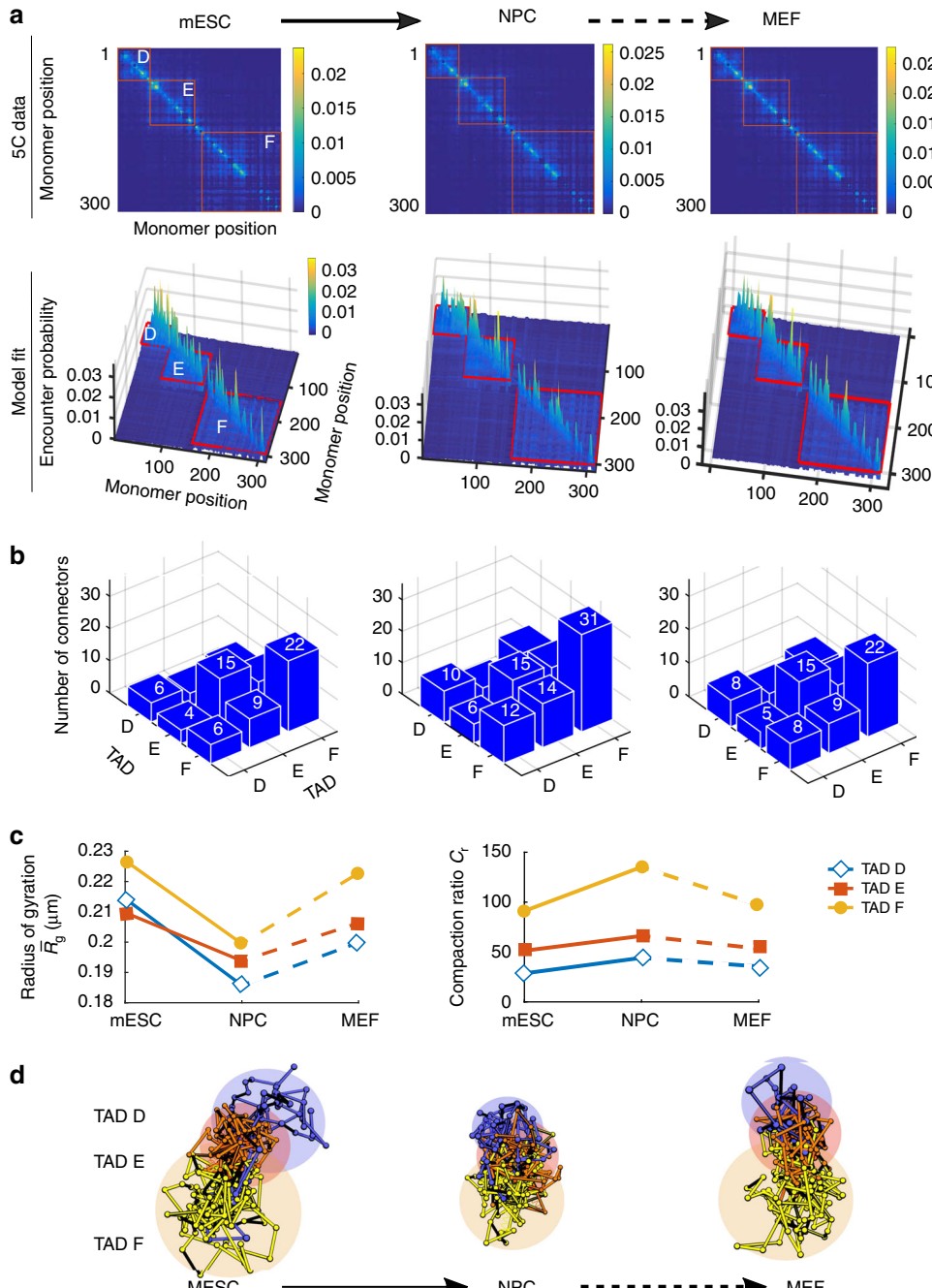

**Fig. 2** Reorganization of the X Chromosome during differentiation. **a** 5C encounter maps and result of fitting expressions (3) and (5) to the empirical 5C encounter probability data[36] at a scale of 6 kb for 3 TADs (red rectangles) at three stages of cell differentiation: mouse embryonic stem cell (mESC, left), neuronal precursor cells (NPC, middle) and embryonic fibroblasts (MEF, right). **b** Mean number of connectors in each TAD, obtained by fitting the empirical EP as described in panel (**a**), within and between TAD D, E, and F for mESC (left), NPC (middle), and MEF (right). The number of connectors within TAD F grows from 22 for MSEC to 31 for NPC and drops back to 22 for MEF cells. The inter-TAD F and E connectivity increases from 9 for MSEC to 14 for NPC and is 9 for MEF stage. The number of connectors within TAD E remains 14 throughout the three stages, whereas the number of connectors for TAD D increases from 6 for MSEC to 10 for NPC and decreases to 8 for MEF. **c** Mean radius of gyration (MRG, left) for the three TADs at three stages of differentiation shows synchronous compaction in the transition from mESC to NPC. The compaction ratio $C_r$ (right) is computed as the cube ratio of $\sqrt{Nb^2/6}$ (MRG of the Rouse chain) to the MRG of each TAD with the same number of monomers $N$, shows that the compaction in TAD E is higher than that of D at NPC stage despite having a higher MRG (0.181 and 0.19 μm for TAD D and E, respectively). **d** Three realizations of the RCL polymer showing the compaction of TAD D (blue), E (red), and F (orange) during the transition from mESC to NPC. Shaded areas represent a ball with of gyration radius listed in panel (**c**) for each TAD in each stage

(1) with the empirical EP $E^{(2)}$ of replica 2 (Supplementary Fig. 5D, left); We found that $\langle \| P^{(1)}(m) - E^{(2)}(m) \| \rangle_m$, averaged over monomers $m$ (Supplementary Methods Equation 65), equals 0.17. Note that the main contribution of this difference arises from

monomers forming long-range loops (Supplementary Fig. 5D) consistent with off-diagonal peaks of the 5C data. Similarly, we found $\langle \| P^{(2)} - E^{(1)} \| \rangle_m = 0.17$ (Supplementary Fig. 5D, right). In addition, the mean radii of gyration for all three TADs in both

replicas were comparable for all three stages of differentiation (Supplementary Fig. 6A, B, left).

To evaluate the consequences of boundaries between TADs on the number of added connectors necessary to reconstruct the heterogeneous RCL polymer (Supplementary Fig. 7), we subdivided each TAD D-F into two equal parts, and repeated the fitting of the heterogeneous RCL model to the EP of the six resulting sub-TADs. We tested the scenario where the number and boundaries of TADs differ from the one in ref. [1], which can result from various TAD-calling algorithms[32,33]. We extracted the intra and inter-TAD connectivity fractions Ξ and computed the number of connectors by fitting the empirical EP (Supplementary Fig. 6A, B). We computed the difference in the average number of connectors between the three TADs case (Supplementary Fig. 6C) and the six sub-TADs case. We found a maximal difference of six connectors for intra-TAD connectivity of TAD E in the MEF stage (Supplementary Fig. 7D). For inter-TAD connectivity, the average difference is two connectors. In addition, we find that the compaction of TADs throughout differentiation is preserved for the six TAD case (Supplementary Fig. 8A), and further find a qualitative agreement between the compaction ratios of the three and six TADs case (see comparison, Supplementary Fig. 8B).

Finally, to determine the robustness of the predictions of the heterogeneous RCL polymer, we compared the reconstructed 5C statistics (Fig. 2) to the statistics reconstructed from Hi-C data[34] of the X chromosome, harboring TAD D, E, and F, binned at 10 kb, with $b = 1.814\,\mu m$ computed from Supplementary Methods Equation 67, and for three successive stages of differentiation: mESC, NPC, and cortical neurons (Fig. 3). Note that the polymer model reconstructed from Hi-C and 5C data, are not necessary identical, although their share some similar statistics, because for each one, the data are generated at a different resolution. However, we found a good agreement between the intra-TAD connectivity of TADs D and E of the 5C and Hi-C data for mESC and NPc stages (Fig. 3a, c). In general, the inter-TAD connectivity in the Hi-C data was lower (average of 1.5 connectors) than that of the 5C (average of 4), which resulted in an increased MRG for all TADs (Fig. 3b, d, left; and a decreased compaction ratios, right). A direct comparison between the reconstructed statistics of the 5C MEF and Hi-C CN was not possible. To conclude, inter-TAD connectivity plays a key role in the compaction of TADs and therefore recovering their exact number is a key step for precisely recovering genome reorganization from 5C data.

**Distribution of anomalous exponents for single monomer trajectories**. Multiple interacting TADs in a cross-linked chromatin environment, mediated by cohesin molecules can affect the dynamics of single loci trajectories. Indeed, analysis of single particle trajectories (SPTs)[35–39] of a tagged locus revealed a deviation from classical diffusion as measured by the anomalous exponent. We recall briefly that the MSD (Eq. (7)) is computed from the positions $r_i(t)$ of all monomers $i = 1, …, N_T$. In that case, the MSD, which is an average over realization, behaves for small time $t$, as a power law

$$\langle (r_i(s + t) - r_i(s))^2 \rangle \propto t^{\alpha_i}. \tag{10}$$

It is still unclear how the value of the anomalous exponent $\alpha_i$ relates to the local chromatin environment, although it reflects some of its statistical properties, such as the local cross-link interaction between loci[14,35]. Thus we decided to explore here how the distribution of cross-links extracted from EP of the Hi-C data could influence the anomalous exponents. For that purpose, we simulate a heterogeneous RCL model, where the number of cross-links was previously calibrated to the data. The number and

position of the connectors remain fixed throughout all simulations (for tens of seconds).

We started with a heterogeneous RCL model with three TADs, reflecting the inter and intra-TAD connectivity as shown in Fig. 2. We generated a hundred chromatin realizations $\mathcal{G}_1, … , \mathcal{G}_{100}$. In each realization $\mathcal{G}_k$, the position of added connectors is not changing. We then simulated in time each configuration a hundred times until relaxation time (Supplementary Methods Equation 23). After the relaxation time is reached, defined as $t = 0$, we followed the position of each monomer and computed the MSD up to time $t = 25$ s. To compute the anomalous exponent $\alpha_i$, we fitted the MSD curves using a power law (Eq. (10)) to estimate the anomalous exponents $\alpha_i, i = 1, … , 302$ along the polymer chain. We repeated the procedure for each stage of cell differentiation: mESC, NPC, and MEF.

In Fig. 4, we plotted the anomalous $\alpha_i$ for each monomer of the three stages mESC (left), NPC (middle), and MEF (right), and for TAD D (dark blue), TAD E (cyan), and TAD F (brown). We find a wide distribution of $\alpha_i$ with values in the range $\alpha_i \in [0.25, 0.65]$ for all TADs in the three cell types. The average anomalous exponent in TAD D is $\alpha_D = 0.46$, in mESC stage, reduced to $\alpha_D = 0.41$ in NPC, due to the increases intra-TAD connectivity, and increased to $\alpha_D = 0.435$ in NPC stage. The average anomalous exponent in TAD E, $\alpha_E = 0.425, 0.41, 0.426$ at mESC, NPC, and MEF stages, respectively. The average anomalous exponent of TAD F was $\alpha_F = 0.443, 0.405, 0.44$ at mESC, NPC, and MEF stages, respectively. The anomalous exponent $\alpha$ decreases with adding connectors, observed throughout differentiation in all TADs, which is in agreement with the compaction and decompaction of TADs (Fig. 2c and Supplementary Figs. 6 and 7). Furthermore, we obtain an average anomalous exponent of 0.4, previously reported experimentally in ref. [38].

To complement the anomalous exponent, we estimated the space explored by monomers by computing the length of constraint $L_c$[35] (computed empirically along a trajectory of $N_p$ points for monomer R as $L_c \approx \sum_i \left( \frac{1}{N_p} R(i\Delta t) - \langle R \rangle \right)^2$) for three monomers in each TAD D, E, F: $r_{20}, r_{70}, r_{120}$. For a single connector realization, we obtain $L_c \approx 0.3, 0.25, 0.26\,\mu m$, respectively, which is about twice the simulated MRG of TAD D, E, F: $0.18, 0.13.0.17\,\mu m$, respectively. Thus we conclude that random distributions of fixed connectors can reproduce the large variability of anomalous exponents reported in experimental systems using single locus trajectories, especially for bacteria and yeast genome[35,38] in various conditions.

## Discussion

Here, we report a general framework based on the RCL model[9,11] to extract statistical and physical properties of multiple interacting TADs. The present polymer model differs from others by several aspects: Our construction of a polymer model from Hi-C is parsimonious. It uses a minimal number of added connectors at a given scale to match the experimental steady-state of Hi-C/5C data, in contrast to the model[8], which is based on a full monomer-monomer interaction, described by pairs of potential wells. In addition, at the scale of few µm occupied by TAD D, E and F of chromosome X, we neglected crowding effects from neighboring chromosomes. Furthermore, we do not use here several types of diffusing binders that need to find a binding site in order to generate a stable link, as introduced in ref. [40]. In addition, contrary to the random loop model (RLM)[16], we do not consider here transient binding, because the positions of random connectors within TADs does not matter (as long as they are uniformly randomly placed). By placing connectors randomly, we capture the heterogeneity in chromatin organization across cell

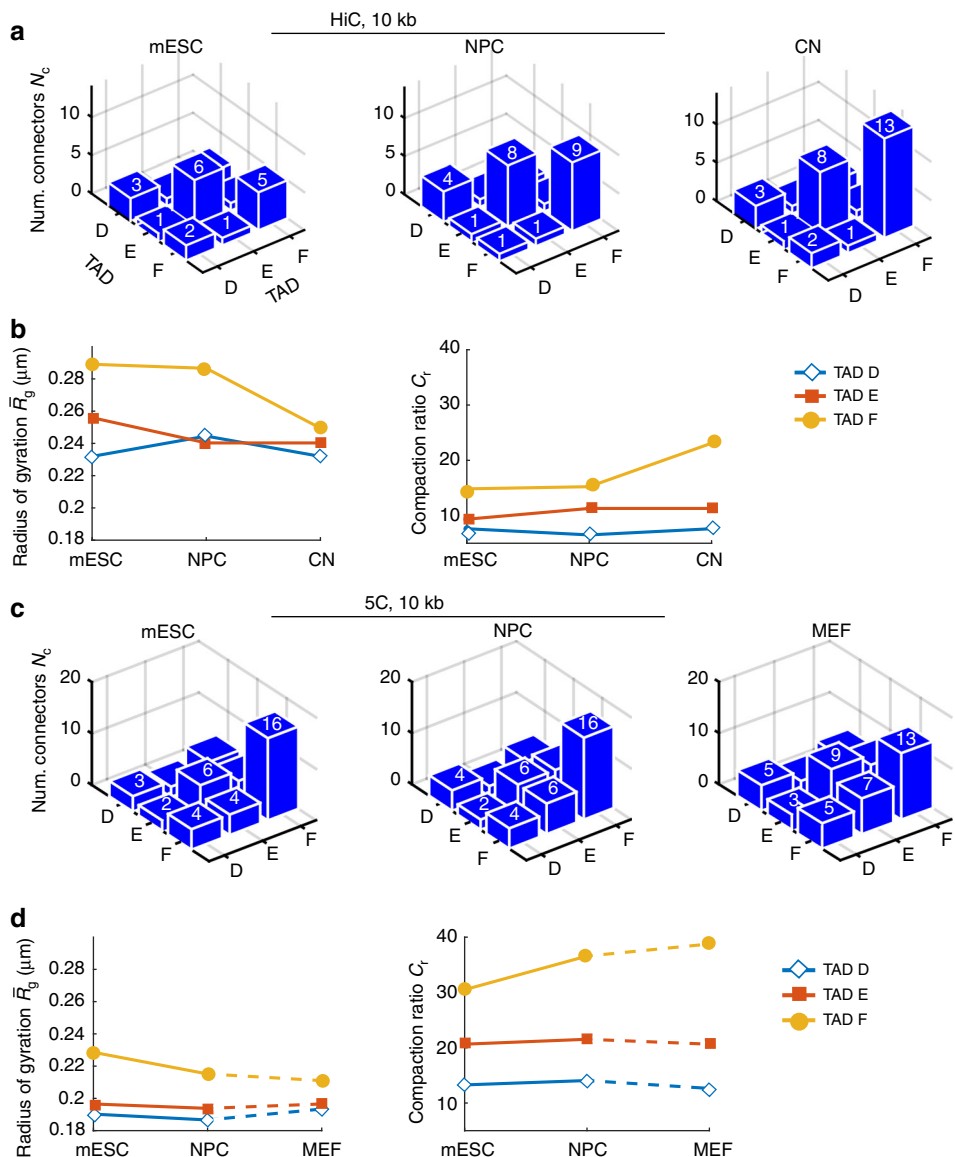

**Fig. 3** Comparing TADs reconstruction during cell differentiation between Hi-C and 5C. **a** Average number of connectors within and between TADs D, E, and F of Hi-C data[34] of the X chromosome binned at 10 kb, obtained by fitting the empirical EP with Eqs. (3) and (5), where TAD boundaries were obtained in ref. [1] for mouse embryonic stem cells (mESC, left), neuronal progenitor cells (NPC, middle), and cortical neurons (CN, right). The average number of connectors within and between TADs are presented in each blue box. **b** Mean radius of gyration (left) for TAD D, E, and F, throughout three successive stages of differentiation of the Hi-C data, with $b = 0.18$ μm obtained from Supplementary Methods Equation 67, and the compaction ratio (right, Eq. (9)). **c** Average number of connectors within and between TADs D, E, and F of the 5C data[1] of the X chromosome binned at 10 kb, obtained by fitting the empirical EP with Eqs. (3) and (5) for mESC (left), NPC (middle), and MEF (right). **d** Mean radius of gyration (left) for TAD D, E, and F of the 5C data, and the compaction ratio (right, Eq. (9))

population. Here we fix connectors, which are stable in the time scale of minutes to hours. The present polymer construction is motivated by the evidence of many stable loci–loci interactions, which are common to the majority of chromosomes in 5C (e.g., peaks of the 5C data)[1,32]. These stable interactions are also found at TAD boundaries, which are conserved in both human and mouse. There are several conflicting studies[5,32,41] about the binding time of connectors (CTCF-cohesin, etc...), which suggest that cross-links can remain stable for minutes to hours and even during the entire phase cycle. Here, we study the chromatin dynamics within this time range where cross-links are stable[32,41]. One final difference between the present RCL model and the RLM model, is the possibility to account for several interacting

TADs and our expressions for statistical quantities such as the radius of gyration, EP, or MSD.

We applied our framework to reconstruct chromatin organization across cell differentiation from conformation capture (3C, 5C, and Hi-C) data, where we accounted for both intra and inter-TAD connectivities. The RCL polymer model allows estimating average number of cross-links within and between each TAD, length-scales such as the MSRG that characterizes the size of folded TADs, and the MSD of monomers in multiple interacting TADs. The present method allowed us to estimate the volume occupied by TADs. These quantities cannot be derived directly from the empirical conformation capture data and are usually extracted from SPT experiments[42,43]. Finally, the present

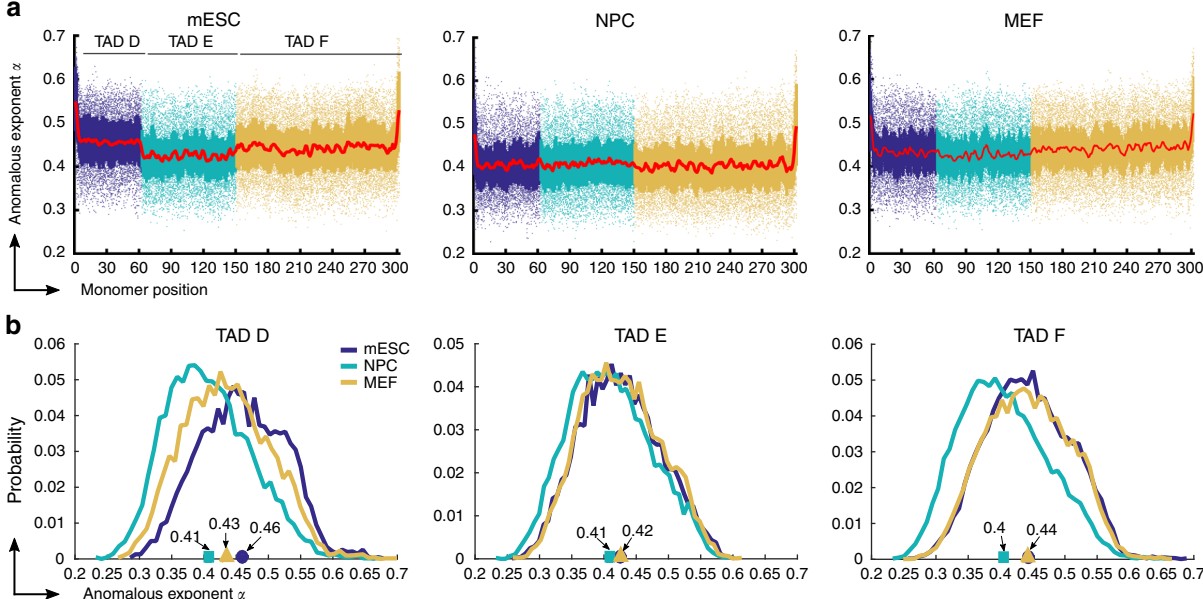

**Fig. 4** Anomalous exponents in three stages of differentiation. **a** Anomalous exponents computed for 302 monomers of the RCL polymer reconstructed in Fig. 2, corresponding to 5C data of three TADs, TAD D (dark blue), TAD E (cyan), and TAD F (brown) of the X chromosome[1]. A hundred realizations are simulated for each configuration using Eq. (17). For each realization, we choose the positions of added connectors uniformly distributed within and between TADs and repeated simulations 100 times, after relaxation time has been reach (Supplementary Methods Equation 23). We then run simulation for 25 s. The anomalous exponents $\alpha_i$, $i = 1, ..., 302$ are obtained by fitting the MSD curve of each monomer using model 10. **b** Distribution of the anomalous exponent in TAD D (left) TAD E (middle), and TAD F (right) for three cell stages: mESC (dark blue), NPC (cyan), and MEF (brown). The average anomalous exponents in TAD D, E, and F, are $\alpha_D = 0.46$, 0.41, 0.435, $\alpha_E = 0.425$, 0.41, 0.426, and $\alpha_F = 0.443$, 0.405, 0.44 for mESC (circle), NPC (square), and MEF (triangle) stages, respectively

approach can also be used to study the phase-liquid transition at the chromatin level, depending on the number of connectors. We computed here the radius of gyration that clarifies how chromatin compaction depends on the number of added connectors. Similar questions were recently discussed in a general perspective[44] and we presented here some answers.

We have applied the present approach to reconstruct multiple TAD reorganization during three stages of cell differentiation: mouse embryonic stem cell (mESC), neuronal precursor cells (NPC), and mouse embryonic fibroblasts (MEF). We fitted expressions (3) and (5) to the empirical 5C encounter matrices of three differentiation stages (Fig. 2a) and showed that the RCL model produced contact enriched TADs with a variability in monomer connectivity within each TAD (Fig. 2a, super and sub-diagonals of EP matrices). We use the average connectivity obtained to compute the average number of connectors within and between TADs (Fig. 2b). At a scale of 6 kb, our results show that the X-chromosome acquires connectors within and between TADs D, E, and F in the transition between mESC to NPC, and the number of connectors is comparable to those of mESC at MEF stage. Increased connectivity for NPC cells is correlated with an increase of LaminB1 (see ref. [1], Fig. 3). Similarly, we reported (Fig. 2a, b) an increase in the number of connectors within TADs D, E, and F in NPC stage in comparison to the mESC stage, which is associated with TAD compaction. Indeed, the MRG curves (Fig. 2c, left) decrease from mESC to NPC stage, indicating a higher chromatin compaction for all TADs. The compaction ratio (compared to Rouse polymer) showed a high compaction at NPC stage (Fig. 2c, right) for all TADs, which can be associated with a heterochromatin state, suppressed gene expression and lamina associating domains[45]. Inter-TAD connectivity remained quite stable as the number of connectors for TAD E did not change, but the MRG decreased and the compaction ratio increased at NPC

stage. This result shows that an accurate description of chromatin from 5C data by polymer models has to account for inter-TAD connectivity. Indeed, despite having similar intra-TAD connectivity, the reconstruction of 5C and Hi-C differs in inter-TAD connectivity, which is reflected in a change in the MRG (Fig. 3). In addition, using Eq. (8), we were able to reproduce the distribution of three-dimensional distances between seven genomic loci, measured by DNA FISH probes (Supplementary Fig. 2). Overall, the RCL polymer captures the correlated reorganization of TAD D, E, and F during differentiation. TAD reorganization is also compatible with transcription co-regulation in these TADs[1]. Moreover, we found here multiple connected regions with weak inter-TAD connectivity, as suggested by experimental conformation capture data[1,4,5,12] that can affect all loci dynamics inside a TAD (Fig. 4).

To describe 5C EP of Hi-C data, we used theory (Eqs. (3) and (5)) and simulations so we could capture both TADs and higher order structures (meta-TADs[12]) resulting from weak inter-TAD connectivity (Fig. 1a, b). This representation allowed us to clarify how the dynamics of monomers are affected by the local connectivities within and between TADs[46]. Using the MSD curves (Fig. 1c), we found that local connectivities (number and positions) are responsible to shape the value of the anomalous exponents, leading to a large spectrum with a mean of 0.4 (Fig. 4). This result explains the large variability in MSD and anomalous exponent behavior reported experimentally in SPT experiments[9,22,38,39,43,47−49]. Indeed, the local chromatin organization can vary in cell population, as cohesin could bind randomly at various places. Furthermore, we have shown here that the anomalous exponents are modulated by the number and the positions of cross-linkers. The value of anomalous exponent $\alpha$ does not depend in general on the diffusion coefficient $D$ or $b$, as known in various other polymer models, such as for Rouse or

β-polymers[14,46]. We have shown here (Fig. 4) that the anomalous exponent crucially depends on the number and the distribution of connectors (see also Supplementary Fig. 3D). In the mean-field approximation, the mean exponent is 0.5, which does not depend on the polymer scale. Finding the exact relation between the number of connectors for a specific connector configuration (not in the mean-field case) remains challenging and relevant to reconstruct the local connectors environment from measured anomalous exponent. To conclude, we propose that measuring the anomalous exponents of loci, positioned at different locations inside a TAD, could reveal the amount of connectors and thus chromatin condensation beyond the exact position of these loci. In summary, the present method allows to reconstruct a polymer model from Hi-C, to generate numerical simulations and to estimate the MSD and the anomalous exponents, relating Hi-C with SPT statistics.

We emphasize here that the present approach can be used to describe chromatin with volume exclusion lower than 40 nm (Supplementary Figs. 3 and 4). Furthermore, we obtained here estimations for the number of connectors within and between TADs, which was consistent across two replicas of the 5C data (Supplementary Figs. 7 and 8). Comparing the predicted number of connectors between 5C and Hi-C data, we found a good agreement in the intra-TAD connectivity. However, the inter-TAD connectivity was reduced in the Hi-C vs. 5C (Fig. 3). We attributed these discrepancies to the elimination of inter-TAD interactions, which are smoothed out at 10 kb resolution for the Hi-C data. Connectors are likely to represent binding molecules such as cohesin and the present results suggest that only a few of them are actually needed to condense chromatin and their exact positions inside a TAD does not necessarily matter.

The interpretation of the number of connectors $Nc$ is not straightforward. This number characterize the amount of connectors at a given scale, which suggests that in the limit of 1 bp resolution $Nc$, it would be equal to the endogenous number of linkers, but as the resolution decreases, the number of connectors that are reported by the Hi-C data should also decrease, because two connectors in the same bin are not counted (Fig. 5a). It is always possible to coarse-grain a polymer model, but reconstructing a refined polymer from a coarse grained kb-scale is an ill-posed problem, because the refined EP at high kb-resolution cannot be inferred from a low resolution (yellow and red arrows in Fig. 5b).

To conclude, the presented framework is a tool to systematically reconstruct chromatin structural reorganization from Hi-C matrices, and can be used to interpret chromatin capture data. In particular, using any experimental data of ligation proximity experiments (Hi-C, 5C 4C), we obtained here statistical properties beyond conformation capture data. The present analysis could also be applied to reveal the statistics of complex matter at the transition point between liquid and gel[44,50] based on heterogeneous random architecture at the level of chromatin sub-organization.

## Methods

**RCL polymer model for multiple interacting TADs.** We now describe the construction of a heterogeneous RCL polymer for multiple interacting TADs. The RCL polymer models were previously used in ref. [11] and (Supplementary Methods subsection Constructing a RCL polymer for a single TAD) to compare the statistical properties of simulations vs. theory for a single TAD. Heterogeneous RCL polymers consist of $N_T$ sequentially connected RCL polymers (Fig. 6a, Supplementary Methods subsection Constructing a RCL polymer for a single TAD) that account for connectivity within TADs of $N = [N_1, N_2, \ldots, N_{N_T}]$ monomers, respectively (Fig. 6b). The polymer is constructed as follows: the position of the ensemble of monomers $\boldsymbol{R}$ in all TADs is described by $N_T$ vectors represented in a block matrix

$$\boldsymbol{R} = \left[\, [R^{(1)}], [R^{(2)}], \ldots, [R^{(N_T)}]\, \right]^T, \tag{11}$$

where the superscript indicates TAD 1, …, $N_T$ it belongs to and square brackets indicate a block matrix, such that $R^{(i)} = \left[ r_1^{(i)}, \ldots, r_{N_i}^{(i)} \right]$. The linear backbone of the polymer is a chain of $N_T$ Rouse matrices (Supplementary Methods Equation 2), defined in the block matrix

$$
\begin{aligned}
\boldsymbol{M} &= diag\left([M^{(1)}], [M^{(2)}], \ldots, [M^{(N_T)}]\right) \\
&= \begin{bmatrix}
[M^{(1)}] & 0 & 0 & \ldots & 0 \\
0 & [M^{(2)}] & 0 & \ldots & 0 \\
0 & 0 & [M^{(3)}] & \ldots & 0 \\
. & . & 0 & \ldots & . \\
. & . & . & \ldots & . \\
. & . & . & \ldots & . \\
0 & . & . & \ldots & [M^{(N_T)}]
\end{bmatrix},
\end{aligned} \tag{12}
$$

where each $[M^{(j)}]$ contains $N_j$ monomers. Note, that during the numerical simulations, we connect the last monomer of block $[M^{(j)}]$ to the first one of $[M^{(j+1)}]$ by a spring connector. One of the key ingredients of the present model is the addition of random spring connectors between non nearest-neighbor (non-NN) monomer pairs. The maximal possible number $N_L$ of non-NN connected pairs within (between) TAD $A_i$ ($A_i$ and $A_j$) is given by

$$
N_L = \begin{cases}
(N_i - 1)(N_i - 2)/2, & |i - j| = 0; \\
N_i N_j - 1, & |i - j| = 1; \\
N_i N_j, & \text{otherwise.}
\end{cases} \tag{13}
$$

We add $Nc(i, j)$ spring connectors randomly between non-NN monomer pairs (Fig. 6b) between TADs $A_i$ $A_j$. We define the square-symmetric connectivity fraction matrix $\Xi = \{\xi_{ij}\}$, $1 \le i, j \le N_T$, as the ratio of number of connectors to the total possible number of non-NN monomer pairs

$$\xi_{ij} = \frac{Nc(i, j)}{N_L}. \tag{14}$$

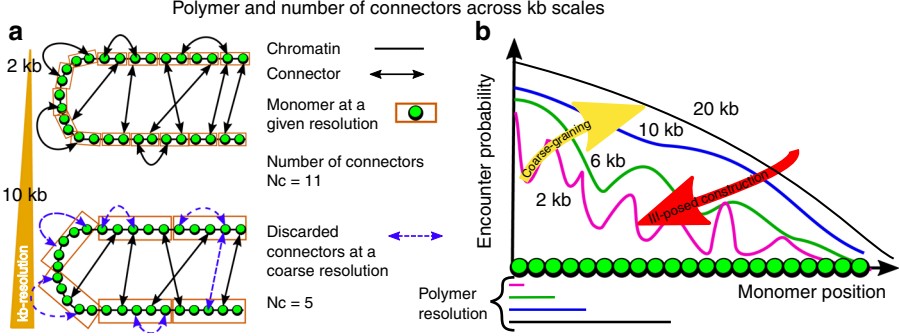

Polymer and number of connectors across kb scales

**Fig. 5** Illustration of polymer reconstruction of chromatin at different kb-scales. **a** Two examples of polymer reconstruction at 2 (Upper) and 10 kb (bottom) resolution, where the number of connectors $Nc$ vary with the scale $Nc = 11$ (2 kb) and $Nc = 5$ (10 kb). For coarse scale, connectors within the same bin (orange boxes) are discarded. **b** Effect of coarse-graining on the encounter probability. It is possible to coarse-grain (yellow arrow), while the construction of a refined polymer model from a coarse-grained resolution (red arrow) is an ill-posed problem

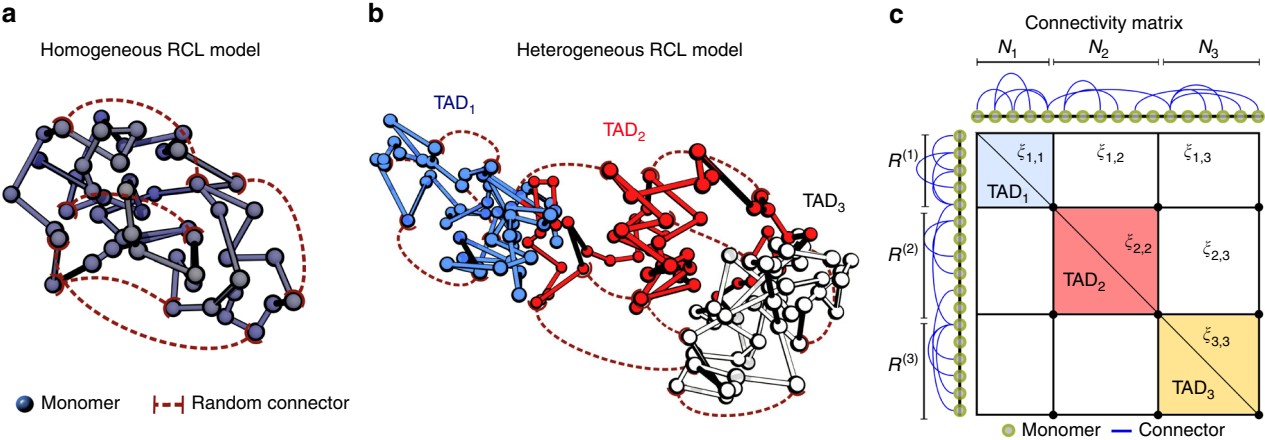

**Fig. 6** RCL polymer model for single and multiple interacting associating domains. **a** Schematic representation of the RCL polymer for a single topologically associating domain (TAD). Monomers (circles) are connected sequentially by harmonic springs to form the polymer's backbone (gray). Spring connectors (dashed red) are added between randomly chosen non nearest-neighbor monomers. The choice of monomer pairs to connect is randomized in each realization. All monomers share similar average level of connectivity (homogeneous RCL model). **b** Schematic representation of the RCL polymer model for three connected TADs (blue, red, white), where monomers (circles) are randomly connected (dashed red) within and between TADs by springs. Monomer connectivity within and between TADs can vary (heterogeneous RCL model). **c** Schematic representation of the symmetric connectivity matrix $B^{\mathcal{G}}(\Xi)$, corresponding to the connected monomers over the three TADs in panel (**b**) (diagonal blocks), where each TAD is of variable number of monomer $N$ and connectivity fraction $\xi_{ij}$

A realization $\mathcal{G}$ of the RCL polymer is a random choice of $Nc(i,j)$ monomer pairs to connect. We define the added connectivity matrix $B^{\mathcal{G}}(\Xi)$ by

$$B^{\mathcal{G}}_{mn}(\Xi) = \begin{cases} -1, & |m-n|>1, r^{(i)}_m, r^{(j)}_n \text{ connected}; \\ -\sum\limits_{j \neq m}^{\sum_{k=1}^{N_T} N_k} B^{\mathcal{G}}_{mj}(\Xi), & m = n; \\ 0, & \text{otherwise}. \end{cases} \quad (15)$$

The spring constants of the linear backbone and the added $Nc$ connectors are similar. The added springs keep distal connected monomers into close proximity. The energy $\phi^{\mathcal{G}}_{\Xi}$ of realization $\mathcal{G}$ of the RCL polymer is the sum of the spring potential of the linear backbone plus that of added random connectors:

$$\phi^{\mathcal{G}}_{\Xi}(\boldsymbol{R}) = \frac{\kappa}{2} Tr(\boldsymbol{R}^T(\boldsymbol{M} + B^{\mathcal{G}}(\Xi))\boldsymbol{R}), \quad (16)$$

where $\kappa$ is the spring constant, $Tr$ is the trace operator. The dynamics of monomers $\boldsymbol{R}$ is induced by the potential energy 16 plus thermal fluctuations:

$$\begin{aligned} \frac{d\boldsymbol{R}}{dt} &= \frac{dD}{b^2} \nabla \phi^{\mathcal{G}}_{\Xi}(\boldsymbol{R}) + \sqrt{2D} \frac{d\omega}{dt} \\ &= -\frac{dD}{b^2}(\boldsymbol{M} + B^{\mathcal{G}}(\Xi))\boldsymbol{R} + \sqrt{2D} \frac{d\omega}{dt}, \end{aligned} \quad (17)$$

for a dimension $d = 3$, the mean distance $b$ between neighboring monomers is defined for a linear chain when there are no added connectors ($Nc = 0$), a diffusion coefficient $D$, and the standard $d$-dimensional Brownian motions $\omega$ have mean 0 and variance 1. We note that added connectors describe the cross-linking by binding molecules such as cohesin or CTCF, and their random positions in $\mathcal{G}$ realization capture the heterogeneity over cell population. In addition, after connectors have been added the mean distance between two consecutive monomers $r_m$, $r_{m+1}$ can be smaller than the initial mean distance $b$ for a Rouse chain ($Nc = 0$), as computed based on Eqs. (4)–(6).

**Reporting summary**. Further information on research design is available in the Nature Research Reporting Summary linked to this article.

## Data availability
The datasets analyzed during the current study are available in the spatial organisation of the X inactivation center repository (www.ncbi.nlm.nih.gov/geo/query/acc.cgi?acc=GSE35721) for the 5C encounter matrices; and in the multi-scale 3D genome rewiring during mouse neural development repository (www.ncbi.nlm.nih.gov/geo/query/acc.cgi?acc=GSE96107) for the Hi-C data.

## Code availability
Codes for performing simulation and fit in the manuscript were constructed using Matlab 2017b. All codes are available from our repository website: http://bionewmetrics.org/statistics-of-chromatin-organization-during-cell-differentiation-revealed-by-heterogeneous-cross-linked-polymers/.

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

## Acknowledgements

V.P. and D.N. acknowledge funding from the Fondation pour la Recherche Médicale (AJE20140630069) and the Agence Nationale de la Recherche (ANR-14-ACHN-0009-01 and ANR-16-TERC-0027-01). D.H.'s research is supported by FMR team (DEQ20160334882).

## Author contributions

O.S. and D.H. wrote the manuscript. O.S. and D.H. conceived the research. O.S. designed the code, performed numerical simulations, and mathematical derivations. O.S. and D.H. established the model. D.N. and V.P. performed analysis of the contact matrices and analyzed the data.

## Additional information

**Competing interests:** The authors declare no competing interests.

