## [Peer Review File · Nature Communications]

Reviewer #1 (Remarks to the Author):

Referee Report on NCOMMS-17-34203 Statistics cross-linked chromatin organization during cell differentiation reconstructed by heterogeneous polymer models

The manuscript aims at reconstructing the three-dimensional structure of a fraction of the chromatin fiber from chromosome conformation capture experiments data.

In the first part, the authors extend the randomly cross-linked (RCL) polymer model (also known as random-loop model) to capture the experimentally observed higher contact probability within topologically associated domain (TAD) in contrast to contact probability between different TADs. Each TAD i is modeled as a linear Rouse-like polymer backbone where a fraction ξ_{ii} of all possible non-consecutive monomer pairs are cross-linked at random by other harmonic springs. Similarly, the contacts between TADs i and j are realized by a fraction ξ_{ij} of randomly chosen harmonic cross-links. The authors use mean-field (MF) approximation to derive various quantities characterizing the conformation and dynamics of TADs, such as mean squared radius of gyration (MSRG), encounter (contact) probability (EP) and mean squared displacement (MSD). The MF approximation in essence means that instead of counting with the fraction ξ of the cross-linked monomers, all monomers are considered cross-linked, however with a coupling constant that is ξ -times smaller. Using this, Brownian dynamics and an assumption of dominating intra-TAD contacts over inter-TAD, the authors derive closed form expressions for the above-mentioned quantities.

In the second part, the authors use their results to fit the parameters of their model to a chromatin segment representing three TADs at three different differentiation stages of the mouse embryonic stem cell.

The main ideas of the manuscript are written clearly with the aid of the appealing visual presentation. However, as detailed below, the extension of the RCL model represents a small improvement to the known model (introduced more than ten years ago) and does not seem to bring a substantial new viewpoint on the TADs or chromatin organization. Moreover, the validity of the model, its assumptions, and simplifications, are not assessed by detailed comparison with experiments and the reconstruction of the 3D structure is not compared to other methods or models.

To summarize, based on the limited novelty and insufficient model validation I would suggest to reject the manuscript from the publication in Nature Communications. This does not mean that I do not consider the work interesting. I think it is important and interesting to have exactly solvable models and their various extensions, even if their direct applicability is somewhat questionable. From this point of view, after some restructuring and clarifications I think the manuscript would be

more appropriate for a publication in a specialized journal that favors comprehensive subject exploration over novelty and paradigm shift. An option would be Physical Review E, where a close predecessor of this model was presented by the authors last year.

Major questions and issues in order of appearance

1. In the RCL model and its present extension the chromatin fiber is assumed to be phantom: there is neither (i) excluded volume interaction nor (ii) any kind of topological constraints that would prevent co-localization or self-crossings of different segments. In dilute conditions in theta solvent this could be a good approximation, but what makes these to be valid assumptions in the case of highly compacted chromatin structure within TADs? This affects not only the statistical properties of the conformation, but also the dynamics of the polymer.
2. The mean-field approximation is also discussed in one of the early works on random-loop model (Bohn et al PRE (2007) - Ref [15] in the manuscript). It was shown there numerically that the size of a chromatin domain could be fitted with the model with MF approximation, but it also says: "Although one could fit the data with these averaged attraction potentials, we see no biological reason for such a potential to exist in the cell." Indeed, the MF represents interactions (although weak) between all the monomers within the domain that is easily $1\mu m$ large. What would be the physical and biological mechanisms behind such long-range attractions? I understand that this assumption is necessary for having an exactly solvable model, but isn't this on the expense of its direct applicability? i.e. What is the validity of the mean-field assumption? (See also related question 9 below.)
3. Along similar lines, as far as I understand, the 5C data represent the contact probability as a population average over many cells, while contacts within single cell can exhibit large variability. Why should we hope that for the RCL model the procedures of averaging and dynamical evolution commute? i.e. evolving an average system is not the same as averaging over evolved systems. (See also related question 9 below.)
4. It would help to clarify what are the different scaling regimes of the MSRG with polymer length. From the presented results, the older paper (Ref [15]) as well as similar result in the authors preceding paper (eq. (33) of Shukron and Holcman

PRE (2017) - Ref [10] in the present manuscript) it's clear that the MSRG initially grows with N , but then the growth is decreasing. This is because adding one monomer adds other N attractive springs i.e. the number of cross-links scales with N^2 , while the length of the polymer is only linear in N . For fixed spring constants this would eventually lead to a collapse in the absence of other repulsive interactions. To my understanding, the collapse in the present model is prevented by the fact that the spring constants decrease as N^{-2} . Could such a fine tuning be present in a biological system? What would be its physics?

Moreover, the above-mentioned scaling could give an answer to the range of polymer lengths the model could be applicable to, by comparison of different scaling regimes of chromatin fiber. What are the lower and upper bounds on the lengths of the polymer, where the model is assumed to work? How would it behave under a different coarse-grained representation, e.g. when a smaller fraction of chromatin is represented by a single bead? This deserves some comments.

5. The scaling exponent of the average contact probability with the genomic separation length of the polymer is a good measure to distinguish between different polymer models [see e.g. S. Sazer, H. Schiessel, Traffic (2017)]. Is the exponent and the behavior predicted here (eq. (15)) consistent with experiments? I understand one can fit the ξ 's to reproduce the behavior to some extent, but I also expect that the exponent will not depend on this fit.
6. Is the dynamics prediction (eq. (17)) and Fig 2C consistent with any experimental evidence?
7. Are the values chosen for the diffusion coefficient and the standard deviation of the spring connector b realistic? What would such a coupling $b = \sqrt{3}\mu m$ represent? How do the model properties depend on this choice?
8. Why is the encounter distance ϵ chosen so large i.e. $b/10$? Does the simulation encounter frequency matrix depend on this choice?
9. The description of the simulations is not completely clear. Firstly, the simulations are running within the MF approximation i.e. using eq. (9). Wouldn't it be better to run it without the MF approximation (because we don't need the MF for numerics) and compare it with the analytical results with MF? This would to some extent elucidate whether MF makes sense. Secondly, the authors say "The connectors are placed uniformly in each A_i ..." do they mean uniformly or uniformly-randomly? This is not clear, because eq. (9) suggests connectors between all the pairs of monomers. Thirdly, the relation between the time step used (Δt) and a natural time scale of the model (D/b^2 ?) should be clarified. Lastly, the time step in the text $\Delta t = 0.05s$ is not consistent with the one stated in the caption of Figure 2 $\Delta t = 0.1s$.
10. Why is there no comparison for the MSRG between the simulations and the theory?
11. Why are the 5C data coarse-grained to $6kb$ before fitting? As far as I understand this is to "smooth out" the peaks found in the data to resemble the model's EP. But doesn't this say simply that the model does not represent the data well and therefore should not be used for this purpose? What predictive power has it then? (See related questions 4 and 5)
12. It was found that "The MSRG increases and decreases in correlation with the acquisition and lose of connectors in all TADs". This seems obvious. Is there a reason why it is not a trivial result? If so, this should be stated. (lose \rightarrow loss)
13. The presentation of the resulting MSRG from the simulations is a bit confusing. Firstly, the numbers are presented as e.g. $2b^2\mu m$ which is inconsistent with MSRG having units of μm^2 and b having units of μm . I presume the μm unit is just a typo and the authors meant e.g. $2b^2$. More importantly, MSRG results are compared only with random walk model and no connection to experiment is made. Plugging in the value of b gives MSRG about $6\mu m^2$, which seems different from the Fluorescence In-Situ Hybridization (FISH) experiment performed on the very same TADs (Nora et al Nature 2012 - Ref. [1] in the manuscript), where they found a 3D distance below $1\mu m$ - see TAD E in Fig. 1.c and d of that publication.
14. The equation (18) seems to depend on the number of monomers used and therefore on the chosen resolution. Why is it interesting to present? What is behind the resulting huge compaction ratios - is it only different scaling of MSRG with number of monomers, or the negligence of some other effects such as the excluded volume interactions?
15. The authors claim "Inter-TAD connectivity determines the compaction of TADs and therefore recovering their exact number is a key step of the reconstruction method and allows to *precisely* recover genome organization from 5C data.". In the light of the assumptions of the model and the absence of its physical microscopic understanding, this seems to me as an excessively strong statement.

16. The present method is claimed to be a new tool to reconstruct chromatin structural reorganization from contact probability matrices. However, no comparison with other such methods is shown and no discussion on advantages of the present tool over others is presented. An example of such tool is e.g. [L. Rieber, S. Mahony, Bioinformatics (2017)], which cites (e.g. section 2.4.) at least about ten other such tools.

Minor issues:

- There is a typo in the equation for the connectivity matrix $B_{mn}^G(\xi)$. The sum should go over $j \neq m$ in the case $m = n$
- There is typo in the encounter frequency plot label in Fig. 2 A and missing units for the quantities D and b in the caption of Fig. 2A.
- There are missing units in the MSD plot in Fig 2 C.
- The units in Fig 3C are wrong.
- There is extra “the” in the first line below eq. (34).
- There is missing “of” in the first line below eq (48).
- There is an extra “gku698” in Reference [21]

Reviewer #2 (Remarks to the Author):

Review of Shukron and Holcman

This manuscript provides a method to analyze data from chromosome conformation studies based on Rouse chain dynamics. It is still a qualitative assessment since there are several assumptions in the model that do not reflect physiological conditions; such as the fact that Rouse chains assume linear springs and the absence of excluded volume. The authors go through a detailed description of the mathematical methods in order to extract numerical averages out of the physical equations without running the code multiple times.

There are several specific issues that need to be justified. There are linear springs linking the chains as well as cross-links between chains. The chain springs (linking beads in a Rouse chain) reflect the behavior of the polymer and are very different from cross-linking springs that are surrogates for cohesin and condensin. They need to justify giving these different physical entities the same spring constant. In addition, the ends of the chains are free (eqn 2), which is probably not the situation in cells. Again, this needs to be justified, or show that tethering the ends does not influence the outcome. In addition, there is no confinement. Again, the experimental evidence is that the chains behave as confined polymers. The absence of confinement needs to be justified. The authors model 3 chains, assuming each one is a TAD. They then put springs (crosslinks) between the chains and examine the change in organization. This is a common strategy in these type of simulations (see Cheng et al., 2015; Arbona et al., 2017; Vasquez et al. 2016; Hult et al., 2017). The major differences are the additional constraints in the prior publications and the lack thereof in this manuscript.

The authors go through very detailed mathematical methods to justify how they can extract numerical averages without running multiple simulations. Unfortunately, if this is intended for a biological audience, it is not appropriate. It requires a deep dive into the equations and some knowledge of statistical mechanics to see the assumptions, which a biological reader will be unprepared to do.

The connectivity matrix has no physical basis. If two monomers are set to be "connected" they can be separated in space. Maybe this is okay in the model since they are doing averages, but it is difficult to conceptualize. The loops in the chains are fixed. The dynamics of loops have a major influence on the position of behavior of the chains (see Hult et al., NAR 2017). Since the springs between loops are dynamic in live cells, the authors need to justify why they utilize fixed, uniform looping.

The statement that the model is robust is very misleading. There is so much "under the hood" that one cannot assess robustness. For instance, on page 3 eqn 10, they have Kuhn length b and value ξ that I believe is the # of connections and is comprised of 6 parameters, however, my understanding is that they performed fittings to 302 sets of data corresponding to the total number of monomers (beads) in their simulation. How 302 fittings were used to obtain 6 parameters is unclear. They authors should at least provide a statement on how robust is this fitting. In addition, the diffusion coefficient $D = 1 \mu\text{m}^2/\text{s}$ is extremely large. In yeast, the diffusion coefficient for the chromosome is $D = 5 \times 10^{-12} \text{ cm}^2/\text{sec}$ or $D = 5 \times 10^{-4} \mu\text{m}^2/\text{sec}$ (Marshall et al., 1997). The diffusion of protein is approx. $5\text{-}15 \mu\text{m}^2/\text{sec}$. Thus I don't know the justification of their $1 \mu\text{m}^2/\text{sec}$ diffusion (p.4). In addition, they approximate the Kuhn length as $\text{sq. rt. of } 3 \sim 1.73 \mu\text{m}$. The Kuhn length of DNA is $2(l_p) = 100 \text{ nm}$. The Kuhn length for the chromosome ranges from 100 to a couple of hundred nm. Again, what is the justification for such a large (and unrealistic) Kuhn length?

The loops are uniform (p. 4) which is also an unrealistic situation. What are the consequences of introducing heterogeneity in loop distribution?

The data in figure 2 was problematic. It seems that they use eqns 13 and 15 to create heat maps, and compare heat maps to experimental. The problem is that they use the encounter probability (EP) to build the equations, then show that the output matches experimental. This is not particularly surprising.

At the conclusion in Fig 2D, they show that increasing cross-links results in compaction. The

statement “we found that MSRG increases and decreases with the acquisition and lose of connectors in all TADs” (p. 7) is a natural extension of many polymer models, most recently in Hult et al (NAR 2017).

The major conclusion is the development of a tool to interpret chromatin capture data. There is little new insight into chromosome structure, organization or dynamics that justify the detailed simulation to a biologically-inclined audience. One of the shortcomings of capture data is that it provides static, averaged information, one would expect the role of modeling is to extend this information to single cell, dynamic information, however I don't see how the proposed procedure can accomplish that.

Reviewer #3 (Remarks to the Author):

This manuscript reports a theoretical and data-analysis study on a model consisting of an heterogeneous ensemble of macromolecules consisting of a set of defined polymers with additional inter- and intra- connectivity expressed in terms of random Gaussian springs.

Focusing on the important problem of the modeling of the emergence of topological association domains (TADs) in chromosome conformation capture experiment, the authors formulate their model in terms of a stochastic differential equation describing Brownian motion in a network of Gaussian potentials. As a follow-up on their previous theoretical work (same authors PRE 2017), where the mean field theory for a single randomly crosslinked polymer was developed, they chose in this work a more complex network connectivity matrix: consisting of N_T independent polymers of length $\{ N_1, N_2, \dots, N_{N_T} \}$ connected by a random matrix B^G . The matrix is generated according to the square-symmetric connectivity fraction matrix ξ_{ij} that for each couple of independent polymers i and j prescribes the number of crosslinks.

From this model, they make analytical mean-field calculation substituting, in the stochastic equation, the random matrix, B^G , by its mean over the ensemble of random matrices. Obtaining formulas describing the encounter probability, the mean square radius of gyration and the monomer dynamics. The mean-field calculations are shown to work excellently in reproducing simulated configurations. I consider this part of the paper the most insightful: I think it is novel, well executed and presented. It will be generally useful to the community.

Finally, the authors utilize their model to describe results from an already published chromosome conformation capture data of the mammalian X chromosomes on a specific region consisting of 3 TADs, in 3 different differentiation stages. In this part of the work I feel major fundamental issues that I would like to be taken into consideration and replied by the authors. I do not think that the work is ready to be published by a selective Journal for a general audience such as Nature communications in the present form. However I think that the authors should have the opportunity to reply to my comments and possibly improve their manuscript.

Major issues:

- The authors merge the data from the two replicas before performing the analysis, could this data be used instead to evaluate the level of noise of the measurements in relation to the differences identified between the different cell lineages, in order, also, to assess their significance?
- The fit from the Data to the model first consists in the identification of TADs: the authors use the reported position of TAD boundaries consisting of the values N_D , N_E and N_F . This step consists alone in the fitting of 4 parameters (the number of TADs and their length) necessary as input to the definition of the model and I wonder if the choice of those parameters does not affect the results? What would happen to results by splitting a TAD in two?

- Is there an advantage in using a physical model to measure the two quantities in question respect to the measurements of raw number of contacts between TADs from experiments? One naively would expect the degree of compaction to be proportional to the intra- over inter- TADs contacts and the number of crosslinks to be proportional to the specific inter- interactions between the TADs in question vs all the rest. Does your model provide additional insight regarding the identification of direct vs indirect interactions? Does your model provide results more apt to be tested by other experimental methods?

- The author choose a Rouse chain as the base for their polymer model, I consider the choice rational since it allows the authors to make very precise analytical calculations. However it neglects steric and knot-topological interactions between different chromatin sections. Could the authors provide some convincing arguments about the adequacy of such a model in this specific context?

- Importantly, regarding meta-TADs, the authors state that the hierarchical organization is a consequence of weak inter- connectivity properties. Personally, I am convinced that the hierarchical organization comes often from the fact that the chromosome is a single polymer and that, for this reason, TADs will have a decreasing contact probability in function of genomic distance, despite any special interactions. How could those views on the matter be conciliated? In particular I would like the author to consider matrix M of eq. 6, that being a block matrix, neglects the link constraints between the frontiers of single-TAD Rouse chains: in this matrix the order of TADs on the diagonal does not matter, while this is generally not true in reality. What changes in the results if adjacent blocks would be connected by single links between their borders?

Minor issues:

- there is a typo at the definition of NN: non-nearest neighboring should be non-NN considering the use of NN later in the manuscript. Otherwise better a less ambiguous rephrasing

- before eq. 10: appendix 37 -> appendix Eq. 37

Reviewers' comments:

Reviewer #2 (Remarks to the Author):

Review of Shukron and Holcman

Comment 1

This manuscript provides a method to analyze data from chromosome conformation studies based on Rouse chain dynamics. It is still a qualitative assessment since there are several assumptions in the model that do not reflect physiological conditions; such as the fact that Rouse chains assume linear springs and the absence of excluded volume.

>Answer 1:The Rouse chain is a quite a good approximation for analyzing single particle trajectories of a locus, but cannot be used for the analysis of 5C and Hi-C data. Many years ago, we developed the beta-polymer with A. Amitai to produce a polymer model with a prescribed anomalous exponent.

Here, unfortunately, we could not use neither Rouse nor the beta-polymer model and we had to extend the RCL-polymer framework that we have introduced last year. We think that in the manuscript, we have now even a superior model, even closer to reality. Indeed, we developed here a method to simulate the multi-TAD heterogeneous RCL polymer, which after this revision include volume exclusion forces, added pairwise between monomers.

We have now added the results of simulations in the Supplementary Information, subsection “Heterogeneous RCL model with volume exclusion forces”, where we report the results of simulations of the RCL polymer with volume exclusion, and compared with theoretical encounter probability and mean square displacement. We compare our theoretical results for the steady-state encounter probability, radius of gyration and mean squared-displacement with simulation of the RCL with volume exclusion, where we tested for two values of the exclusion radius, denoted by c . The added exclusion potential is modeled by the potential

$$\phi_V(\mathbf{R}) = \frac{\kappa}{2} \sum_{m,n}^N (r_m - r_n)^2 H(r_m, r_n, c),$$

where the indicator function H is defined by

$$H(r_m, r_n, c) = \begin{cases} 1, & \|r_m - r_n\| < c; \\ 0, & \text{else.} \end{cases}$$

and c is the exclusion sphere radius. The total potential of the RCL polymer is given by

$$\phi(\mathbf{R}) = \phi_\xi^{\mathcal{G}}(\mathbf{R}) + \phi_V(\mathbf{R})$$

where $\phi_\xi^{\mathcal{G}}(\mathbf{R})$ is the spring potential from the linear backbone and of added random connectors (Eq. 5, main text).

For $c=0.4\text{nm}$, where b is the mean square distance between adjacent monomers, we find that the steady-state encounter probability of the RCL model with volume exclusion matches the theoretical results (Eqs. 10, 12, main text). The results for 10,000 simulations of the RCL model with volume exclusion of radius $c=0.04 \mu\text{m}$, $b=0.2 \mu\text{m}$, $D=8 \times 10^{-3} \mu\text{m}^2/\text{s}$, $\Delta t = 0.01 \text{ s}$ are shown in SI Fig. 2 below:

SI Figure 2: **Statistical properties of the heterogeneous RCL polymer with added volume exclusion.** **A.** Encounter frequency matrix of a polymer with three TAD blocks (TAD₁, TAD₂, TAD₃) of $N_1=50$, $N_2=40$, $N_3=60$ monomers is computed from 10,000 simulations of the RCL polymer using Eq. 6 (main text) with the added volume exclusion potential 58, and $\Delta t = 0.01$ s, $D=8 \times 10^{-3}$ $\mu\text{m}^2/\text{s}$, $d=3$, $b=0.2$ μm , $\varepsilon=0.02$ μm , and $c=0.04$ μm . The number of added connectors within and between TADs appears in each block. Three distinct diagonal TADs are visible (red boxes) where secondary structure appears (black lines) due to weak inter-TAD connectivity. **B.** Encounter probability (EP) of the heterogeneous RCL described in panel A, where the simulated EP (red) agrees with the theoretical one (blue, Eqs. 42, 52). We plot the EP of the middle monomer in each TAD: monomer r_{20} (top left), monomer r_{70} (top right) and monomer r_{120} (bottom left), where TAD boundaries are in vertical dashed lines. **C.** Averaged mean squared displacement of monomers in TAD₁ (blue), TAD₂ (orange) and TAD₃ (yellow), using simulations of RCL polymer, as described in panel A: simulation (continuous line) vs theory (dashed, Eq. 54).

The simulation mean radius of gyration for TAD₁, TAD₂, and TAD₃, are 0.179, 0.134, 0.168 vs. 0.178, 0.132, and 0.167 μm , from theory (Eq. 8, main text), respectively. An average error of 1%.

We then increase the exclusion radius to $c=0.0667\mu\text{m}$, and found several disagreements between the theoretical and the simulated encounter probability, mean square displacement, and radius of gyration, as shown in Supplementary Information SI Fig. 3 below

SI Figure 3: **Statistical properties of the heterogeneous RCL polymer with added volume exclusion.** **A.** Encounter frequency matrix of a polymer as in Fig. SI Fig. 2A, with radius of exclusion $c=0.067 \mu\text{m}$. The number of added connectors appears in each block. Three distinct diagonal TADs are visible (red boxes) where secondary structure appears (black lines) due to weak inter-TAD connectivity. **B.** Encounter probability (EP) of the heterogeneous RCL described in panel A, where the simulation EP (orange) deviates from the theoretical EP (blue, Eqs. 42, 52), plotted for the middle monomer in each TAD: monomer r_{20} (top left), monomer r_{70} (top right) and monomer r_{120} (bottom left), and TAD boundaries appear in dashed red lines. **C.** Averaged mean squared displacement of monomers in TAD₁ (blue), TAD₂ (red), and TAD₃ (yellow), using simulations of RCL polymers, as described in panel A: simulations (continuous line) vs. theory (dashed, Eq. 54), that do not agree for $t > 5$ s.

The results of the simulation mean radius of gyration for radius of exclusion 67nm , for TAD₁, TAD₂, and TAD₃, are 0.192 , 0.144 , and $0.181 \mu\text{m}$ vs. 0.178 , 0.132 , and $0.167\mu\text{m}$ from theory, for TADs 1 to 3, respectively. An average error of 8.5% .

To conclude, these new results show that for $b=0.2\mu\text{m}$, $c=0.0667\mu\text{m}$, we can expect a deviation from the theory presented in this paper. But, and this should be considered a remarkable achievement: the present theory, which neglected initially the exclusion volume is actually quite accurate for polymers with an exclusion radius of up to 40 nm , which is a reasonable scale.

We have now added in the Results section, subsection “Statistical properties of the heterogeneous RCL and numerical validations“, last paragraph:

“In addition, we found that adding an exclusion forces with a radius of 40nm did not lead to any modifications of the statistical quantities defined above (see SI Fig. 2 compared to Fig. 2C). However, when the exclusion radius increases to 67nm , deviations started to appear (SI Fig. 3). To conclude, an exclusion radius of the order of 40 nm , also used in [29], is consistent with the physical crowding properties of condensin and cohesin [30] to fold and unfold chromatin.”,

We also added two references:

[29] T. M. Cheng, S. Heeger, R. A. Chaleil, N. Matthews, A. Stewart, J. Wright, C. Lim, P. A. Bates, and F. Uhlmann, *Elife* 4 (2015).

[30] D. E. Anderson, A. Losada, H. P. Erickson, and T. Hirano, *The Journal of cell biology* 156, 419 (2002).

Comment 2

The authors go through a detailed description of the mathematical methods in order to extract numerical averages out of the physical equations without running the code multiple times.

>Answer 2: We agree with the reviewer, our model allows to extract the average number of connectors directly from the empirical encounter probability for a given coarse-grained scale without to rely on further simulations. To further clarify, what we did, we shall add that:

1- we do not extract averaged values only, but also estimate the probability density function, the correlation function, etc... of monomer dynamics and local chromatin organization.

2-later on, when we needed to validate our formulas (Eqs. 8-14, main text) using stochastic simulations, we did run the code multiple times, at least 10,000 times (see Results, Figs. 2-3, SI Figs. 2-3).

Comment 3

There are several specific issues that need to be justified. There are linear springs linking the chains as well as cross-links between chains. The chain springs (linking beads in a Rouse chain) reflect the behavior of the polymer and are very different from cross-linking springs that are surrogates for cohesin and condensin.

>Answer 3: Following the reviewer recommendations, we have now better explained and justified several unclear statements: cross-linking reflect here the binding of molecules such as CTCF or cohesin and we use a tether which is equivalent of a binding force to a monomer located on the linear backbone.

We use the same spring force for binding as that for the connection between monomers of the linear backbone. We are aware that the force holding to two genomic loci by binding molecules might differ than that of the linear backbone and might affect dynamical and transient properties of the chromatin. However, here we are computing steady-state properties of the chromatin, for which the added springs for the cross-links are only used to draw two distal loci into close proximity, and their position recorded at the end of each realization (relaxation time, SI Eq. 23).

In our analysis we only collect the end configuration of each polymer realization, which matches the 5C protocol, in which an ensemble of millions of ‘fixed’ configurations are used to construct the encounter matrices. In these fixed chromatin configurations, the strength of the springs has little importance.

To emphasize this point, we have now added in the Methods section, subsection “RCL polymer for multiple interacting TADs”, second paragraph:

“The spring constants of the linear backbone and the added N_c connectors are similar. The added springs keep distal connected monomers into close proximity.”

In Ref. 8 (main text), short range random cross links were assigned one spring constant, while long-range persistent connectors were assigned a higher value, to account for frequent interactions (seen as off-diagonal peaks of the empirical 5C matrix) at 3kb resolution. Here we use 5C matrices binned at 6 kb resolution, for which long-range peaks in the data are smoothed out. Therefore, the increased encounter probability in TAD vs. Non-TAD region is only attributed to short-range random

connectors, associated with the activity of CTCF/cohesin molecules. For these molecules we assume similar binding strength for all cross-link positions. Hence, we assign a constant spring constant for the random cross-links.

Comment 4

They need to justify giving these different physical entities the same spring constant.

>Answer 4: As long as the cross-linker is bound, the strength of the force should be strong enough to maintain two parts of the chromatin together. There are no specific values available in the literature. So in the absence of specific values, we have chosen the values of the chromatin backbone. This choice is also the one made in Shukron, 2017 PLoS Computational Biology [Ref. 8], Shukron 2017, PRE [Ref 10], and Shukron et al., 2017, Scientific Reports and were accepted in the literature.

Comment 5

In addition, the ends of the chains are free (eqn 2), which is probably not the situation in cells. Again, this needs to be justified, or show that tethering the ends does not influence the outcome.

>Answer 5: We have verified that the free ends of the polymer chain do not change our analytical results for the encounter probability (Fig. 2, main text). In the RCL polymer, often there are no free ends, due to the fact that the ends are cross-linked to inner monomers.

This fact can be appreciated by the excellent agreement between theoretical and simulation results of the encounter probability (Fig. 2B), for which we reported the EP of 3 monomers r_{20} , r_{70} , and r_{120} located at the middle of TADs. Furthermore, monomers in the middle TAD (TAD₂, Fig. 2, and TAD E, Fig. 3) are tethered on both ends, but our analytical EP is in excellent agreement with simulation EP.

Comment 6

In addition, there is no confinement. Again, the experimental evidence is that the chains behave as confined polymers. The absence of confinement needs to be justified. The authors model 3 chains, assuming each one is a TAD. They then put springs (crosslinks) between the chains and examine the change in organization. This is a common strategy in these type of simulations (see Cheng et al., 2015; Arbona et al., 2017; Vasquez et al. 2016; Hult et al., 2017). The major differences are the additional constraints in the prior publications and the lack thereof in this manuscript.

>Answer 6: Confinement is actually not easy to add, because it is only reflected in SPTs. Most of the models do not use confinement extracted from experimental data, but assumed that there are forces. In the RCL model, the added connectors act precisely as a resulting confining force, which could correspond to a quadratic potential of radius of confinement.

In addition, the goal of this research is to connect the physical characteristics of the polymer (number of connectors) to the encounter probability in a multi-TAD chromatin and the experimental 5C data.

There is not any information about the boundary in the experimental data that we can incorporate in the model. The boundary in the work by Cheng et al., Arbona et al., Vasquez et al., and Hult et al., is synthetic, and its exact mechanism of interaction with the chromatin is completely hypothetical.

Comment 7: The authors go through very detailed mathematical methods to justify how they can extract numerical averages without running multiple simulations.

>Answer 7: We do make multiple runs for each of empirical quantities we are computing. The description we are presenting is the minimum information, that is the mean number of cross-links. The number of realizations for each simulation type (Fig. 2, main text, and SI Figs. 2-7) are presented in Figure captions (10,000 runs). We perform 10,000 runs of the RCL polymer (See Results, subsection “Statistical properties of heterogeneous RCL and numerical validations”).

Comment 8

Unfortunately, if this is intended for a biological audience, it is not appropriate.

>Answer 8: It is very hard to satisfy all communities from biology, biophysics, statistical physics, applied mathematics, statisticians, etc. but we are open to suggestions and we think that the revised version is much more balanced for a large readership.

At this stage, we have now added new HiC data that we analyzed and presented our new statistical approach as an algorithm that we will provide with the publication to anyone interested to use it. We have now reorganized the manuscript so that the experimental biology community can pass quickly over the detail description of the model to focus on the result of the reconstruction.

We now insist on the relationship between the empirical encounter probability decay and the number of physical connectors (measurable concentration of binding molecules) we have derived, and our observation of the qualitative compaction and expansion of TADs throughout differentiation stages. Our method allows to quantify structural changes the chromatin organization throughout differentiation and understand their effect on the chromatin within and between TADs as the cell differentiates. These structural changes are correspond, in turn, to the acquisition of heterochromatic and euchromatin characteristics of TADs.

Comment 9

It requires a deep dive into the equations and some knowledge of statistical mechanics to see the assumptions, which a biological reader will be unprepared to do. The connectivity matrix has no physical basis.

Answer 9: We would like to explain what is the physical meaning of the connectivity matrix: The mean-field connectivity matrix is used here to obtain an analytical approximation of the steady-state configuration and dynamic properties of the chromatin: it allows us to derive expressions for the variance, encounter probability, and mean-squared displacement. This matrix represents an average connectivity, in agreement with the expected ensemble of conformation of the HiC or 5C data. However, it is not used for simulations.

In the present stochastic simulations, we use an ensemble of connectivity matrices, each with its own random configuration, which corresponds to an ensemble of many chromatin realizations (Eq. 7, main text, rather than SI Eq. 12, as we stated by mistake in the previous version of the manuscript), in which there are N_c monomer pairs chosen to be connected (Eq. 5, main text), rather than the mean-field average connectivity matrix (SI Eq. 5).

This methodology corresponds to the average ensemble obtained from empirical experimental data, in which cross-links are present in the organization of the chromatin, and are distributed between genomic loci. Simulation results and verification of the analytical results is done here for an ensemble of realizations, each having a connectivity matrix with N_c connected pairs (Eq. 5, main text).

To conclude, we have corrected our mistake (quoting the wrong equation about the mean instead of the stochastic equation) and we have also corrected the following statements in the manuscript:

1. Fig.2 caption: “Encounter frequency matrix of a polymer with three TAD blocks (TAD₁, TAD₂, TAD₃) of N₁=50, N₂=40, N₃=60 monomers each, result of 10,000 simulations of the system Eq. 6 with $\Delta t = 0.01$ s, $D = 8 \times 10^{-3} \mu\text{m}^2/\text{s}$, $d = 3$, $b = 0.2 \mu\text{m}$.
2. Result section, subsection “Statistical properties of heterogeneous RCL and numerical validations”, third paragraph: ”To construct the encounter frequency matrix, we simulated Eq. 6 in dimension $d = 3$, with $b = 0.2 \mu\text{m}$ and diffusion coefficient $D = 8 \times 10^{-3} \mu\text{m}^2/\text{s}$ starting with a random walk initial polymer configuration.”

Comment 10

If two monomers are set to be “connected” they can be separated in space. Maybe this is okay in the model since they are doing averages, but it is difficult to conceptualize. The loops in the chains are fixed. The dynamics of loops have a major influence on the position of behavior of the chains (see Hult et al., NAR 2017). Since the springs between loops are dynamic in live cells, the authors need to justify why they utilize fixed, uniform looping.

>Answer 10: There is a misunderstanding here, probably due to our lack of explanations: the position loops can actually vary across realizations because each time we ran a simulation, we changed how monomers were connected. This procedure represents the possible change of the cohesin/CTCF binding organization across cells and captures the heterogeneous chromatin organization at the time of 5C.

Furthermore, we are only interested here in the steady-state end-configuration of an ensemble of cells to construct the encounter frequency matrix and not the dynamics of connectors. This is exactly what is captured in the 5C experiments, in which an ensemble of chromatin is extracted from cells, and fixed using formaldehyde, to give a single fixed configuration for each chromatin. Thus, the dynamics of springs in live cells has no role in the computation of steady-state properties, because no temporal dynamics is involved. In addition, the probability of any two pairs of monomers to be connected is uniform, rather than a uniform spread of connectors, as implied by the reviewer.

To conclude, to clarify this last point, we have modified in the Results section, subsection “Statistical properties of heterogeneous RCL and numerical validations”, third paragraph, the sentence:

“Connectors are placed with uniform probability in each TAD_i and in between TADs, as indicated in Fig. 2A”.

Comment 11

The statement that the model is robust is very misleading. There is so much under the hood that one cannot assess robustness.

>Answer 11: To clarify the robustness of the model, we have now added new experimental data and added 2 new experimentalists as co-authors.

Indeed, we have now performed fitting of the multi-TAD heterogeneous RCL encounter probability on a second independent data-set, the HiC data of the X chromosome, published in Bonev et al. Cell, 2017. For that, we extracted the average number of connectors within and between TADs D, E, and F, of the X chromosome for three cell stages mESC, NPC, and Cortical neurons (CN) at a resolution of 10kb and compared to fitting of the 5C data at the same resolution.

The results are summarized in the Result section, subsection “Genome reorganization with multiple TADs from the HiC data reconstructed from the heterogeneous RCL polymer during cellular differentiation”. The new Figure 4 is below.

Figure 4: Comparing TADs reconstruction during cell differentiation between HiC and 5C. A. Average number of connectors within and between TADs D, E, and F of HiC data [31] of the X chromosome binned at 10 kb, obtained by fitting the empirical EP with Eq. 10, 12, where TAD boundaries were obtained in [1] for mouse embryonic stem cells (mESC, left), neuronal progenitor cells (NPC, middle), and cortical neurons (CN, right). The average number of connectors within and between TADs are presented in each blue box. **B.** Mean radius of gyration (left) for TAD D, E, and F, throughout three successive stages of differentiation of the HiC data, with $b=0.18\mu\text{m}$ obtained from SI Eq. 62, and the compaction ratio (right, Eq. 15). **C.** Average number of connectors within and between TADs D, E, and F of the 5C data [1] of the X chromosome binned at 10 kb, obtained by fitting the empirical EP with Eq. 10, 12 for mESC (left), NPC (middle), and MEF (right). **D.** Mean radius of gyration (left) for TAD D, E, and F of the 5C data, and the compaction ratio (right, Eq. 15).

The HiC data included the first two cell types as in the Nora 5C data, i.e., mESC and NPC, for which we find reasonable agreement between the average number of connectors in the 5C vs the HiC data (see Figure 3, main text). In addition, the mean radius of gyration for mESC and NPC stages agrees between the 5C and HiC, i.e., mean of 0.22-0.25 μm , at mESC, and NPC stages. Several discrepancies appear in our comparison between the reconstruction of the 5C and HiC data, but they highlight the importance of inter-TAD connectivity.

In addition, we have added in the Supplementary Information, subsection “Sensitivity of the RCL polymer model to TAD boundary locations”, where we report the results of testing the sensitivity of the RCL fitting procedure to the position of TAD boundaries.

We further subdivided the three TADs D, E, and F, of Nora et al. Nature 2012, into 2 sub TADS each and repeated fitting procedure of the encounter probability (Eq. 10, 12 main text), as described in the Method section. We found that the average number of connectors found by fitting the sub TADs sum up to those found for 3 TADs. This result is shown in the Supplementary Figure 6 below:

SI Figure 6 **Robustness of TAD boundaries in RCL representation.** **A.** Average number of connectors for the genomic section spanning TAD D, E, and F, when each TAD is further divided in half to give rise to six TADs, binned at 6kb resolution, for mESC (left), NPC (middle), and MEF (right) cell types. **B.** Average number of connectors obtained by fitting SI Eq. 42, 52 to the empirical EP, in each non-overlapping 2x2 blocks of the number of connectors in panel A, corresponding to the initial subdivision of the genomic segment into TADs D, E, and F. **C.** Average number of added connectors for TADs D, E, and F presented in Fig. 3 (main text). **D.** Difference between the average number of connectors shown in panel B and panel C.

We further computed the radius of gyration for the case of 3 and 6 TADs in Supplementary Information SI Fig. 7. The figure is brought below

SI Figure 7: **Comparing three and six sub-TADs throughout cellular differentiation.** **A.** Mean radius of gyration of the 5C data [3] at 6 kb for TADs D, E, and F when each TAD is further divided in half (left), such that TADs D E and F comprises TADs 1-2, 3-4, and 5-6, respectively, and (right) with no TAD sub-division. **B.** Compaction ratio (left, Eq. 15, main text) of TADs 1-6 throughout differentiation, and with no TAD sub-division (right).

We have added to the Result section, subsection “Reconstructing genome reorganization with multiple TADs from 5C data during cellular differentiation” The following paragraph:

“To evaluate the consequences of boundaries between TADs on the number of added connectors necessary to reconstruct the heterogeneous RCL polymer (SI Fig. 6), we subdivided each TAD D-F into two equal parts, and repeated the fitting of the heterogeneous RCL model to the EP of the six resulting sub-TADs. We tested the scenario where the number and boundaries of TADs differ from the one in [1], which can result from various TAD-calling algorithms [29, 30]. We extracted the intra and inter-TAD connectivity fractions Ξ and computed the number of connectors by fitting the empirical EP (SI Fig. 5A-B). We computed the difference in the average number of connectors between the three TADs case (SI Fig. 5C) and the six sub-TADs case. We found a maximal difference of six connectors for intra-TAD connectivity of TAD E in the MEF stage (SI Fig. 6D). For inter-TAD connectivity, the average difference is two connectors. In addition, we find that the compaction and decompaction of TADs throughout differentiation is preserved for the six TAD case (SI Fig. 7A), and further find a qualitative agreement between the compaction ratios of the three and six TADs case (see comparison, SI Fig. 7B).”

Additionally, we have tested the consistency of the RCL fit (encounter probability) between the two replica of the 5C data (Nora et al. 2012). We have now

added to the Supplementary Information the subsection “Independent fit and comparison of the two replicas of the 5C data at 10kb resolution”, where we fitted the RCL model on one 5C replica, to extract the connectivity fractions X_i within and between TADs, and used the fitted X_i matrix from one replica to compute the square difference between the EP formula (Eqs 10-12, main text) and the empirical 5C data of the second replica. When we reverse the roles of the two replicas, we found a consistent result. The result is shown in Supplementary Information SI Fig. 4 below

SI Figure 4. **Comparing the number of connectors between two replica of 5C data at a resolution of 10kb.** **A.** Average numbers of inter and intra-TAD connectors for TAD D, E, and F, obtained by fitting the RCL model EP (Eq. 42, 52) to each monomer of replica 1 of the dataset presented in [3], binned at 10 kb resolution, and for three stages of cell differentiation: mESC (left), NPC (middle), and MEF (right). **B.** Same as in panel A for replica 2. **C.** Absolute difference between the average number of connectors found independently for replica 1 and 2 (panels A and B), and for the three cell stages. **D.** Norm of the difference $\|P^{(1)} - E^{(2)}\|$ (Eq. 59), between the EP $P^{(1)}$ fitted to replica 1, and the empirical EP $E^{(2)}$ of replica 2 (left) and then we show the opposite (right), where we compute the norm of the difference $\|P^{(2)} - E^{(1)}\|$.

We have also computed the mean radii of gyration and compaction ratios for each replica individually and added in the Supplementary Information, SI Fig. 5, below, showing that the compaction and decompaction of TAD throughout differentiation is consistent with the observation of the mean of two replica (Fig. 3, main text). SI Fig. 5 is below.

SI Figure 5. **Statistics of individual replicas of the 5C data.** **A.** Mean Radius of Gyration (MRG) of replica 1 of the X chromosome (left) at 10 kb resolution for TADs D (blue diamonds), E (orange squares), and F (yellow circles), and for three stages of cell differentiation, shows a synchronous TAD compaction in the transition from mESC to NPC cells, and the compaction ratio at the MEF stage. The compaction ratio (right, Eq. 15, main text) further reveals the chromatin changes across cell differentiation. **B.** Mean radius of gyration (left) of TADs D (blue diamonds), E (red, squares), and F (yellow, circles) of 5C replica 2 of the X chromosome indicates a synchronous compaction of all three TADs in the transition from mESC to NPC. The changes in the compaction ratio (right) of replica 2 is comparable to that of replica 1 (panel A, right).

Comment 12

For instance, on page 3 eqn. 10, they have Kuhn length b and value ξ_m ; that I believe is the number of connections and is comprised of 6 parameters, however, my understanding is that they performed fittings to 302 sets of data corresponding to the total number of monomers (beads) in their simulation. How 302 fittings were used to obtain 6 parameters is unclear. They authors should at least provide a statement on how robust is this fitting.

>Answer 12: In subsection “Genome reorganization with multiple TADs”, we state in paragraph 2:

“We computed the number of connectors N_c , within and between TADs, by averaging the connectivity values ξ_m for monomers in each TAD, obtained from fitting the EP (Eq. 10 and 12, main text) to all 302 monomers. After averaging we then obtained the connectivity matrix Ξ , and used it in relations Eqs. 3 and 4 (main text) to recover the number of connectors within and between TADs.”

Comment 13

In addition, the diffusion coefficient $D = 1 \mu\text{m}^2/\text{s}$ is extremely large. In yeast, the diffusion coefficient for the chromosome is $D = 5 \times 10^{-7}$; $10 \times 10^{-12} \text{ cm}^2/\text{sec}$ or $D = 5 \times 10^{-7}$; $10 \times 10^{-12} \text{ m}^2/\text{sec}$ (Marshall et al., 1997). The diffusion of protein is approx. $5\text{-}15 \times 10^{-5} \text{ m}^2/\text{sec}$. Thus I don't know the justification of their $1 \mu\text{m}^2/\text{sec}$ diffusion (p.4). In addition, they approximate the Kuhn length as $\text{sqrt. of } 3 \sim 1.73 \mu\text{m}$. The Kuhn length of DNA is $2(l_p) = 100 \text{ nm}$. The Kuhn length for the chromosome ranges from 100 to a couple of hundred nm. Again, what is the justification for such a large (and unrealistic) Kuhn length?

>Answer 13: Indeed, we need to clarify what we meant here: we have used the value of $D=1 \mu\text{m}^2/\text{sec}$, for comparing stochastic simulations with the theoretical results (Eqs. 8-13) - formulas for the encounter probability, radius of gyration, and mean square displacement- and not in connection with any empirical data.

We have now replaced Fig. 2, with a new Fig. 2, using more realistic parameters: $D=8 \times 10^{-3} \mu\text{m}^2/\text{sec}$, $b=0.2 \mu\text{m}$. This change of parameter values, did not alter the validity of our theoretical results as can be appreciated by the new Figure 2 that we now replaced in the manuscript. The new Figure 2 is shown below:

Figure 2: **Statistical properties of the heterogeneous RCL polymer.** **A.** Encounter frequency matrix of a polymer with three TAD blocks (TAD_1 , TAD_2 , TAD_3) of $N_1=50$, $N_2=40$, $N_3=60$ monomers, computed from 10,000 simulations of Eq. 7 with $\Delta t = 0.01 \text{ s}$, $D=8 \times 10^{-3} \mu\text{m}^2/\text{s}$, $d=3$, $b=0.2 \mu\text{m}$, $\epsilon=0.02 \mu\text{m}$. The number of added connectors appears in each block. Three distinct diagonal TADs are visible (red boxes) where secondary structure appears (black lines) due to weak inter-TAD connectivity. **B.** Encounter probability (EP) of the heterogeneous RCL described in panel A, where the simulation EP (orange) is in agreement with theoretical EP (blue, Eqs. 10, 12), plotted for the middle monomer in each TAD: monomer r_{20} (top left), monomer r_{70} (top right) and monomer r_{120} (bottom left). **C.** Average mean square displacement of monomers in each TAD of the heterogeneous RCL polymer, simulated as described in panel A, simulation (full) versus theory (dashed Eq. 14) are in good agreement for time up to 25 s.

We corrected in the Result section, subsection “Statistical properties of heterogeneous RCL and numerical validations”, second paragraph, the values of the parameters: “To construct the encounter frequency matrix, we simulated equation 7 in dimension $d=3$, with $b=0.2\mu m$ and diffusion coefficient $D=8\times 10^{-3}\mu m^2/sec$ starting with a random walk initial polymer configuration”

In addition, we have computed the MRG of the 5C data (Fig. 3C) using the new parameter $b=0.2\mu m$. The new panel C in Figure 3 is below

We have corrected in the Results section, subsection "Reconstructing genome reorganization with multiple TADs from 5C data during cellular differentiation, 4th paragraph, the sentence “The MRG of all TADs decreased from average of $0.21\mu m$ at MESC stage to $0.9\mu m$ for NPC and increased back to $0.2\mu m$ for MEF cells”

Comment 14

The loops are uniform (p. 4) which is also an unrealistic situation. What are the consequences of introducing heterogeneity in loop distribution?

>Answer 14: There is probably a misunderstanding here. We allow N_c loops, for which two monomers are connected chosen with a uniform distribution. We chose the connector uniformly distributed from 1 to N because cohesin/CTCF could bind uniformly on the chromatin. The distribution of length has already been computed in Shukron, et al. Scientific Reports 2017 for a single TAD, where we find that the expected loop length is

$$\langle L \rangle = (b^2(1-\exp(-(N\xi)^{0.5})/(N\xi)^{0.5})^{0.5}(N^3-7N+6)/(3(N-1)(N-2)),$$

where N is the number of monomer in a single TAD, and $\xi=2N_c/(N-1)(N-2)$ is the connectivity fraction.

We have now emphasized that the probability of any two pairs in a TAD to be connected is uniform in the Results section, subsection “Statistical properties of heterogeneous RCL and numerical validations”, third paragraph, we modified the sentence: “Connectors were placed between monomer with uniform probability in each TAD_i and in between TADs according to the numbers indicated in Fig. 2A”

Comment 15

The data in figure 2 was problematic. It seems that they use eqns 13 and 15 to create heat maps, and compare heat maps to experimental. The problem is that they use the encounter probability (EP) to build the equations, then show that the output matches experimental. This is not particularly surprising.

>Answer 15: This comment is possibly a misunderstanding due to the lack of specification from our side: In Fig. 2, we have compared numerical and analytical

results only, but no real data: the heat maps in Fig. 2A are the encounter probabilities obtained from numerical simulations, but were not generated using the theoretical EP Eqs. 10-12.

The comparison between simulations and theoretical EP is done in Fig. 2B, where we were quite surprised and satisfied to find an excellent agreement between theoretical and simulation results. To emphasize this point, we have now added in the caption of Figure 2A:

“Encounter frequency matrix of a polymer with 3 TAD block (TAD₁, TAD₂, TAD₃) of N₁=50, N₂=40, N₃=60 monomers, computed from 10,000 simulations of Eq. 7 with $\Delta t=0.01$ s, $D=8 \times 10^{-3} \mu m^2/s$, $d=3$, $b=0.2 \mu m$ ”

Comment 16

At the conclusion in Fig. 2D, they show that increasing cross-links results in compaction. The statement “we found that MSR_G increases and decreases with the acquisition and loss of connectors in all TADs” (p. 7) is a natural extension of many polymer models, most recently in Hult et al (NAR 2017).

>Answer 16: We agree with the review: The fact that increasing cross-links results in compaction is clear, but the question is by how much and how do inter-TAD sparse connectors affect the MSR_G of each TAD separately? This is much less trivial: we already gave the full analytic treatment with new formula for the MSR_G presented for the first time in Shukron PRE 2017.

We extend here this result to the case of several interacting TADs, which was not done before, and not in Hult et al (NAR 2017). The inter-TAD connectivity affect the compaction of each TAD, and thus the MSR_G.

We emphasize this by editing in Result section, subsection “Reconstructing genome reorganization with multiple TADs from 5C data during cellular differentiation” third paragraph:

“We found that the MRG can both increase and decreases depending on the number of connectors within TADs but is also affected by inter TAD connectivity, as revealed by Eq. 8.”

Comment 17

The major conclusion is the development of a tool to interpret chromatin capture data. There is little new insight into chromosome structure, organization or dynamics that justify the detailed simulation to a biologically-inclined audience.

>Answer 17: due to the lack of precision, we have create here a misunderstanding in the interpretation of our analysis. With the new validation of the model, we would like now to recall that the main message here is to be able to find the average numbers of connectors directly from 5C/Hi-C map to study chromatin dynamic reorganization throughout cell differentiation stages.

In the absence of references in the literature, to our knowledge, we have introduced here a new method that clearly provides a tool to study how the chromatin is organized and quantify changes the chromatin undergoes throughout differentiation stages, in terms of folding and the concentration of binding molecules.

The average number of connectors in a given scale of the data, is a key parameter for studying the dynamics of a polymer model constrained on the HiC data. The dynamics of the chromatin and length scales, such as the radius of gyration, are not present in the experimental HiC data. Our model allows to compute these statistical quantities of the chromatin in complex organization and justifies, in our view, the

detailed simulations we have performed in this work. We are unaware of any other existing algorithm that could generate similar results.

Comment 18

One of the shortcomings of capture data is that it provides static, averaged information, one would expect the role of modeling is to extend this information to single cell, dynamic information, however I don't see how the proposed procedure can accomplish that.

>Answer 18: In the present manuscript, we developed a framework to study dynamics of the chromatin directly from the static empirical 5C and HiC encounter frequency matrices. The RCL model outputs the average number of added connectors to a linear polymer to match experimental and theoretical encounter probability at a given scale. Once the connectivity fractions are extracted, we are able to compute the mean square displacement of single monomers (genomic segments) using Eq. 14, allowing to reliably simulate SPTs, constrained by the empirical data.

This result by itself is new and extend the static information embedded in the empirical data, in contrary to reviewer's statement. Furthermore, the focus of our method is to describe the dynamics of TADs as the cell undergoes differentiation. We have devoted Figure 3 to this topic. We show how acquisition of intra and inter-TAD connectors alter the steady-state and dynamic properties of 3 TADs, which display synchronous compaction and decompaction. These findings static HiC map can reveal dynamical properties.

To emphasize the applicability of our RCL model to study transient properties of specific loci of the chromatin and to connect with single particle trajectories experiments, we have now added in the main text the section "Distribution of anomalous exponents for single monomer trajectories", where we compute using the calibrated RCL model (Fig. 3, main text) the influence of chromatin architecture on the measured anomalous exponent: we simulate an ensemble of fixed configurations and computed the anomalous exponent per monomer in three stages of cell differentiation from the 5C data. The results are shown in Fig. 5, below:

Figure 5. Anomalous exponents in three stages of differentiation. A. Anomalous exponents computed for 302 monomers of the RCL polymer reconstructed in Fig. 3, corresponding to the empirical 5C data of three TADs, TAD D (dark blue), TAD E (cyan), and TAD F (brown) of the X chromosome [1]. A hundred realizations are simulated for each configuration using Eq. 7. For each realization we choose the positions of added connectors uniformly distributed within and between TADs and repeated simulations 100 times from relaxation time (SI Eq. 23) to $t=25$ s. The anomalous

exponent α_i , $i=1,\dots,302$ are obtained by fitting the MSD curve of each monomer using the model 16. **B.** Probability density of the anomalous exponent in TAD D (left) TAD E (middle) and TAD F (right) for three cell stages: mESC (dark blue), NPC (cyan), and MEF (brown). The average anomalous exponents in TAD D, E, and F, are $\alpha_D=0.46, 0.41, 0.435$, $\alpha_E=0.425, 0.41, 0.426$, and $\alpha_F = 0.443, 0.405, 0.44$ for mESC (circle), NPC (square), and MEF (triangle) stages, respectively.

To conclude, our approach allows to connect two type of statistics: 5C/HiC (average ensemble over cells) to SPT (time average over trajectories). We think that this approach does not only provide new methodology, but also new ideas and concepts.

We thank the reviewer for comments, that helped us clarify the manuscript and for suggestions to make it available for a large and mixed audience.

Reviewer #3

This manuscript reports a theoretical and data-analysis study on a model consisting of a heterogeneous ensemble of macromolecules consisting of a set of defined polymers with additional inter- and intra-connectivity expressed in terms of random Gaussian springs. Focusing on the important problem of the modeling of the emergence of topological association domains (TADs) in chromosome conformation capture experiment, the authors formulate their model in terms of a stochastic differential equation describing Brownian motion in a network of Gaussian potentials. As a follow-up on their previous theoretical work (same authors PRE 2017), where the mean field theory for a single randomly cross-linked polymer was developed, they chose in this work a more complex network connectivity matrix: consisting of N_T independent polymers of length $\{ N_1, N_2, \dots, N_{NT} \}$ connected by a random matrix B^G . The matrix is generated according to the square-symmetric connectivity fraction matrix x_{ij} that for each couple of independent polymers i and j prescribes the number of crosslinks.

From this model, they make analytical mean-field calculation substituting, in the stochastic equation, the random matrix, B^G , by its mean over the ensemble of random matrices. Obtaining formulas describing the encounter probability, the mean square radius of gyration and the monomer dynamics. The mean-field calculations are shown to work excellently in reproducing simulated configurations. I consider this part of the paper the most insightful: I think it is novel, well executed and presented. It will be generally useful to the community.

Finally, the authors utilize their model to describe results from an already published chromosome conformation capture data of the mammalian X chromosomes on a specific region consisting of 3 TADs, in 3 different differentiation stages. In this part of the work I feel major fundamental issues that I would like to be taken into consideration and replied by the authors. I do not think that the work is ready to be published by a selective Journal for a general audience such as Nature communications in the present form. However I think that the authors should have the opportunity to reply to my comments and possibly improve their manuscript.

Major issues:

Comment 1

The authors merge the data from the two replicas before performing the analysis, could this data be used instead to evaluate the level of noise of the measurements in relation to the differences identified between the different cell lineages, in order, also, to assess their significance?

>Answer 1: This is an excellent question: we have now performed fitting on a single replica of the 5C data, Nora et al., Nature, 2012, and validated our reconstruction on the second replica.

This validation stage was further done when we reversed the roles of the two replicas. To assess the quality of the fit, we computed the norm of the difference between the encounter probabilities fitted on one replica with the empirical encounter probability of the second replica. We further compared the resulting average number of added connectors within and between TADs to those found by using the average of the two replica. We found an excellent agreement between the fitting of both replica and the fitted to the average of the two replicas (Fig. 3, main text).

We have now added a SI Figure 4 to the Supplementary Information in the subsection “Independent fit and comparison of the two replicas of the 5C data at 10kb resolution”. The figure is shown below:

SI Figure 4 **Comparing the number of connectors between two replica of 5C data at a resolution of 10kb.** **A.** Average numbers of inter and intra-TAD connectors for TAD D, E, and F, obtained by fitting the RCL model EP (Eq. 42, 52) to each monomer of replica 1 of the dataset presented in [3], binned at 10 kb resolution, and for three stages of cell differentiation: mESC (left), NPC (middle), and MEF (right). **B.** Same as in panel A for replica 2. **C.** Absolute difference between the average number of connectors found independently for replica 1 and 2 (panels A and B), and for the three cell stages. **D.** Norm of the difference $|P^{(1)} - E^{(2)}|$ (Eq. 59), between the EP $P^{(1)}$ fitted to replica 1, and the empirical EP $E^{(2)}$ of replica 2 (left) and then we show the opposite (right), where we compute the norm of the difference $|P^{(2)} - E^{(1)}|$.

In addition, we have computed the mean radius of gyration and compaction ratio for each replica individually, and discovered the compaction and decompaction of TADs follows as in the average replica case (Fig. 3, main text), we now add these result in Supplementary Information Figure 5, below:

SI Figure 5: **Statistics of individual replicas of the 5C data.** **A.** Mean Radius of Gyration (MRG) of replica 1 of the X chromosome (left) at 10 kb resolution for TADs D (blue diamonds), E (orange squares), and F (yellow circles), and for three stages of cell differentiation, shows a synchronous TAD compaction in the transition from mESC to NPC cells, and the compaction ratio at the MEF stage. The compaction ratio (right, Eq. 15, main text) further reveals the chromatin changes across cell differentiation. **B.** Mean radius of gyration (left) of TADs D (blue diamonds), E (red, squares), and F (yellow, circles) of 5C replica 2 of the X chromosome indicates a synchronous compaction of all three TADs in the transition from mESC to NPC. The changes in the compaction ratio (right) of replica 2 is comparable to that of replica 1 (panel A, right).

To conclude, we obtain an excellent agreement between the fitting of the two replica, with a maximal of 5 connectors difference in TAD F of the mESC stage. Examining the fit of replica 1 on the empirical EP of replica 2 shows that the norm of the difference between the two is 0.17 on average, with higher norm at monomers with persistent long-range connectors.

Reversing the roles of replica 1 and two, and examining the norm of the difference, shows a similar average norm of difference of 0.17. These results strengthen our method claim and demonstrate the robustness of the RCL model approach. We thank the reviewer for bringing this idea to our attention.

Comment 2

The fit from the data to the model first consists in the identification of TADs: the authors use the reported position of TAD boundaries consisting of the values N_D , N_E and N_F . This step consists alone in the fitting of 4 parameters (the number of TADs and their length) necessary as input to the definition of the model and I wonder if the choice of those parameters does not affect the results? What would happen to results by splitting a TAD in two?

>Answer 2: We agree with the reviewer: the identification of TAD boundaries is not very satisfying in the entire literature, but we have here relied for the present on pre-existing information about TAD boundary published in Nora et al. 2012 [Ref. 1]. However, such a procedure can be performed by already established TAD calling

algorithms, such as TADtree [Weinreb, *Bioinformatics* 32, 1601–1609 (2016)], Arrowhead [Rao et al., *Cell* 2014].

However, to address the point of the review, we have now examined the sensitivity of the RCL fitting procedure to TAD boundaries positions: we now perform a fit of the theoretical encounter probability (Eqs. 10, 12, main text) to the 5C Nora, 2012 data, TADs D, E, and F, at a scale of 6kb, where we further divide each TAD into two sub-TADs. We found that the average number of connectors associated with the sub-TADs sum up to that of the initial TADs (Fig. 3B, main text, and SI Fig. 6), with a difference of 5 intra-TAD connectors. Although we observe slight variations we find the overall results consistent. We now report this new result in the SI Figure 6 below

SI Figure 6 **Robustness of TAD boundaries in RCL representation.** **A.** Average number of connectors for the genomic section spanning TAD D, E, and F, when each TAD is further divided in half to give rise to six TADs, binned at 6kb resolution, for mESC (left), NPC (middle), and MEF (right) cell types. **B.** Average number of connectors obtained by fitting SI Eq. 42, 52 to the empirical EP, in

each non-overlapping 2x2 blocks of the number of connectors in panel A, corresponding to the initial subdivision of the genomic segment into TADs D, E, and F. **C.** Average number of added connectors for TADs D, E, and F presented in Fig. 3 (main text). **D.** Difference between the average number of connectors shown in panel B and panel C.

In addition, we have investigated the dynamics of the TADs throughout the three differentiation stages, where the six sub-TADs are detected, and compared it to the case of three TADs. We find a qualitative agreement between the cases of three and six sub-TADs, in which a synchronous chromatin compaction and decompaction occurs throughout the three stages of differentiation.

To conclude, this result shows that these phenomena of synchronous compaction is consistently captured by the RCL model despite the different division of the 5C data into TADs, which can be the result of the use of different TAD calling algorithms.

We now added Supplementary Figure 7, summarizing the similarity and differences in MRG of the three and six sub-TADs and the compaction ratio throughout cellular differentiation. The SI Fig. 7 is below:

SI Figure 7 **Comparing three and six sub-TADs throughout cellular differentiation.** **A.** Mean radius of gyration of the 5C data [3] at 6 kb for TADs D, E, and F when each TAD is further divided in half (left), such that TADs D E and F comprises TADs 1-2, 3-4, and 5-6, respectively, and (right) with no TAD sub-division. **B.** Compaction ratio (left, Eq. 15, main text) of TADs 1-6 throughout differentiation, and with no TAD sub-division (right).

We added in Result section, subsection “Reconstructing genome reorganization with multiple TADs from 5C data during cellular differentiation” 6th paragraph, the sentence:

“we find that the compaction of TADs throughout differentiation is preserved for the six TAD case (SI Fig. 7A), and further find a qualitative agreement between the compaction ratios of the three and six TADs case (see comparison, SI Fig. 7B).”

Comment 3

Is there an advantage in using a physical model to measure the two quantities in question respect to the measurements of raw number of contacts between TADs from experiments?

>Answer 3: The loci-loci 5C contact are reported in millions of reads per interacting pair. Therefore, we cannot use the count directly but only within the context of a physical polymer model which takes into account the ensemble of chromatin structures. Using a physical model further allows us to predict dynamics and transient statistical properties (transient mean first encounter times, see Shukron PRE 2017), which cannot be extracted directly from raw 5C maps. It also allows us to connect HiC data with single particle trajectories of a locus. The link between the two methods is interesting because it connects two type of different experiments.

We address here specifically the question: what is the architecture of a polymer model which can reproduce the ensemble steady-state chromosome conformation encounter maps?

Reading a single realizations' CC map cannot reveal this information, and population 5C maps do not reveal the level of heterogeneity in chromatin architecture across the ensemble. Our model, therefore, solves this problem, by forming a link between the ensemble encounter probability decay, and the average number of connectors in a single realization.

To emphasize this point we have now added in the Introduction, last paragraph, the sentence:

“These properties cannot be studied by direct comparison of the overall number of interactions within and between TADs from 5C data, because individual realizations cannot be discerned from the ensemble.”

Comment 4

One naively would expect the degree of compaction to be proportional to the intra-over inter-TADs contacts and the number of crosslinks to be proportional to the specific inter-interactions between the TADs in question vs all the rest. Does your model provide additional insight regarding the identification of direct vs indirect interactions?

>Answer 4: To measure compaction, we suggested here to take into account the size of the linear genome inside each TAD and not only the number of connectors within and between TADs. We then compare the volume of the genomic region associated with the radius of gyration of the linear vs. the folded chain in each TAD. By doing so, we eliminate effects of chromatin size and the incomplete knowledge that we have about the parameter b (std of connectors length). Directly comparing the number of Inter vs Intra-TAD connectivity is insensitive to the size of the genomic region.

Measuring direct and indirect interactions between TADs is related to find the exact numbers of connectors that we have solved here. How many connections are needed to reproduce the EP map? And where shall they be positioned and do they need to be precisely positioned?

The method developed here provide an optimal answer for the number of connectors but show their precise location is not that important, as long as there are located on the correct TAD or between TADs.

We think that having estimated the correct average intra and inter-TAD connectivity matrix, which result from the RCL fitting procedure (Fig 3, main text) is

the key to the reconstruction and the statistical analysis and more important the simulation to determine the transient properties.

Comment 5

Does your model provide results more apt to be tested by other experimental methods?

>Answer 5: Our model predicts the number of binding molecules in a given genomic section, which reproduces the empirical encounter probability at a given resolution. We have now added new data about 5C, where we use the present model to compare HiC with 5C for the reconstruction of TADs and their statistics (Figure 4, main text). Our method reveals similarities, but also some local changes in the average number of connectors for TAD D, E and F. Specifically, we find an excellent agreement between the intra-TAD connectivity of TAD D and E in mESC and NPC stages, with the only difference of 11 connectors in TAD F. In addition, the HiC data included less inter-TAD connectivity, which resulted in some differences in the radius of gyration between HiC and 5C, these differences are of the order of 40nm on average.

We think the present approach can be used to analyze and compare any HiC data. The result of comparison of the HiC (Bonev et al.) and 5C (Nora et al.) is summarized Fig. 4, below.

Figure 4: Comparing TADs reconstruction during cell differentiation between HiC and 5C. A. Average number of connectors within and between TADs D, E, and F of HiC data [31] of the X chromosome binned at 10 kb, obtained by fitting the empirical EP with Eq. 10, 12, where TAD boundaries were obtained in [1] for mouse embryonic stem cells (mESC, left), neuronal progenitor cells (NPC, middle), and cortical neurons (CN, right). The average number of connectors within and between TADs are presented in each blue box. **B.** Mean radius of gyration (left) for TAD D, E, and F, throughout three successive stages of differentiation of the HiC data, with $b=0.18\mu\text{m}$ obtained from SI Eq. 62, and the compaction ratio (right, Eq. 15). **C.** Average number of connectors within and between TADs D, E, and F of the 5C data [1] of the X chromosome binned at 10 kb, obtained by fitting the empirical EP with Eq. 10, 12 for mESC (left), NPC (middle), and MEF (right). **D.** Mean radius of gyration (left) for TAD D, E, and F of the 5C data, and the compaction ratio (right, Eq. 15).

The present method could be use to reveal how changing the concentration of binding molecules such as cohesin and CTCF would affect chromatin organization, for

example during dsDNA break induction. The MSR analysis could be validated by using single particle trajectories, tagging several loci in a given TAD.

We have now added Figure 5 to show how the RCL model can be used to predict the distribution of anomalous exponents, based on the inter-and intra-TAD connectivity, that could be compared to experimental data. Figure 5 is attached below

Figure 5: **Anomalous exponents in three stages of differentiation.** **A.** Anomalous exponents computed for 302 monomers of the RCL polymer reconstructed in Fig. 3, corresponding to the empirical 5C data of three TADs, TAD D (dark blue), TAD E (cyan), and TAD F (brown) of the X chromosome [1]. A hundred realizations are simulated for each configuration using Eq. 7. For each realization we choose the positions of added connectors uniformly distributed within and between TADs and repeated simulations 100 times from relaxation time (SI Eq. 23) to $t=25$ s. The anomalous exponent α_i , $i=1, \dots, 302$ are obtained by fitting the MSD curve of each monomer using the model 16. **B.** Probability density of the anomalous exponent in TAD D (left) TAD E (middle) and TAD F (right) for three cell stages: mESC (dark blue), NPC (cyan), and MEF (brown). The average anomalous exponents in TAD D, E, and F, are $\alpha_D=0.46$, 0.41 , 0.435 , $\alpha_E=0.425$, 0.41 , 0.426 , and $\alpha_F=0.443$, 0.405 , 0.44 for mESC (circle), NPC (square), and MEF (triangle) stages, respectively.

We have also added in the Result section the section C “Distribution of anomalous exponents for single monomer trajectories“, summarizing the procedure of computing the anomalous exponents and description of our results in Figure 5.

Comment 6

The author choose a Rouse chain as the base for their polymer model, I consider the choice rational since it allows the authors to make very precise analytical calculations. However it neglects steric and knot-topological interactions between different chromatin sections. Could the authors provide some convincing arguments about the adequacy of such a model in this specific context?

>**Answer 6:** The use of a phantom chain in previous studies and have shown that models with exclusion forces provide little advantage for recovering genome architecture and dynamics over a phantom chain. The interplay between mathematical and numerical tractability and the evidence from literature support out choice of the phantom chains. We have now added the following references.

However, to address the reviewer comments in full we now added a simulation of volume exclusion to account for steric interactions. Not that any topological

interactions that would have led to a local increase in the EP matrix would have been captured here by a connector.

We have now added in the Supplementary Information the subsection III “Heterogeneous RCL model with volume exclusion forces”, where we report the results of simulations of the RCL polymer with volume exclusion, and compared with theoretical encounter probability and mean square displacement. We summarize now the new results and figures:

We compare our theoretical results for the steady-state encounter probability, radius of gyration and mean squared-displacement with simulation of the RCL with volume exclusion, where we test for two values of c , the exclusion radius. The added exclusion potential is modeled by the potential

$$\phi_V(\mathbf{R}) = \frac{\kappa}{2} \sum_{m,n}^N (r_m - r_n)^2 H(r_m, r_n, c),$$

where the indicator function H is defined by

$$H(r_m, r_n, c) = \begin{cases} 1, & \|r_m - r_n\| < c; \\ 0, & \text{else.} \end{cases}$$

and c is the exclusion sphere radius. The total potential of the RCL polymer is then

$$\phi(\mathbf{R}) = \phi_\xi^G(\mathbf{R}) + \phi_V(\mathbf{R})$$

with $\phi_\xi^G(\mathbf{R})$, the spring potential from the linear backbone and that of added random connectors (Eq. 1, main text).

For $c=b/4\mu\text{m}$, with b the mean square distance between adjacent monomers, we find that the steady-state encounter probability of the RCL model with volume exclusion matches the theoretical results (Eq. 8-13, main text). The results for 10,000 simulations of the RCL model with volume exclusion of radius $c=0.04 \mu\text{m}$, $b=0.2 \mu\text{m}$, $D=8 \times 10^{-3} \mu\text{m}^2/\text{s}$, $\Delta t=0.01 \text{ s}$ are shown in the figure below.

SI Figure 2: **Statistical properties of the heterogeneous RCL polymer with added volume exclusion.** **A.** Encounter frequency matrix of a polymer with three TAD blocks (TAD₁, TAD₂, TAD₃) of $N_1=50$, $N_2=40$, $N_3=60$ monomers is computed from 10,000 simulations of the RCL polymer using Eq. 7 (main text) with the added volume exclusion potential 58, and $\Delta t = 0.01$ s, $D=8 \times 10^{-3}$ $\mu\text{m}^2/\text{s}$, $d=3$, $b=0.2$ μm , $\varepsilon=0.02$ μm , and $c=0.04$ μm . The number of added connectors within and between TADs appears in each block. Three distinct diagonal TADs are visible (red boxes) where secondary structure appears (black lines) due to weak inter-TAD connectivity. **B.** Encounter probability (EP) of the heterogeneous RCL described in panel A, where the simulated EP (red) agrees with the theoretical one (blue, Eqs. 42, 52). We plot the EP of the middle monomer in each TAD: monomer r_{20} (top left), monomer r_{70} (top right) and monomer r_{120} (bottom left), where TAD boundaries are in vertical dashed lines. **C.** Averaged mean squared displacement of monomers in TAD₁ (blue), TAD₂ (orange) and TAD₃ (yellow), using simulations of RCL polymer, as described in panel A: simulation (continuous line) vs theory (dashed, Eq. 54).

Numerical simulations of the mean radius of gyration for TAD₁, TAD₂, and TAD₃, reveal that values 0.179, 0.134, 0.168 vs. 0.178, 0.132, and 0.167 μm (from theory, Eq. 8, main text), respectively. An average error of 1%.

However, when we increased the exclusion radius to $c=0.0667 \mu\text{m}$, we found several discrepancies between the theoretical and simulation encounter probability and mean square displacement, as shown in the figure below

SI Figure 3: **Statistical properties of the heterogeneous RCL polymer with added volume exclusion.** **A.** Encounter frequency matrix of a polymer as in Fig. SI Fig. 2A, with radius of exclusion $c=0.067 \mu\text{m}$. The number of added connectors appears in each block. Three distinct diagonal TADs are visible (red boxes) where secondary structure appears (black lines) due to weak inter-TAD connectivity. **B.** Encounter probability (EP) of the heterogeneous RCL described in panel A, where the simulation EP (orange) deviates from the theoretical EP (blue, Eqs. 42, 52), plotted for the middle monomer in each TAD: monomer r_{20} (top left), monomer r_{70} (top right) and monomer r_{120} (bottom left), and TAD boundaries appear in dashed red lines. **C.** Averaged mean squared displacement of monomers in TAD₁ (blue), TAD₂ (red), and TAD₃ (yellow), using simulations of RCL polymers, as described in panel A: simulations (continuous line) vs. theory (dashed, Eq. 54), that do not agree for $t > 5$ s.

The results of the simulation mean radius of gyration for radius of exclusion 67nm , for TAD₁, TAD₂, and TAD₃, are 0.192 , 0.144 , and $0.181 \mu\text{m}$ vs. 0.178 , 0.132 , and $0.167\mu\text{m}$ from theory, for TADs 1 to 3, respectively. An average error of 8.5% .

To conclude, these result show that for $b=0.2\mu\text{m}$, $c=0.0667\mu\text{m}$, we can expect a clear deviation from the theory presented in this paper. The theory presented in this work is accurate for polymers with exclusion radius of up to 40 nm .

In addition, we have added in the Results subsection “Statistical properties of the heterogeneous RCL and numerical validations“, the last paragraph:

“In addition, we found that adding an exclusion forces with a radius of 40nm did not lead to any modifications of the statistical quantities defined above (see SI Fig. 2 compared to Fig. 3). However, when the exclusion radius increases to 67nm , deviations started to appear (SI Fig. 3). To conclude, an exclusion radius of the order of 40 nm , also used in [29], is consistent with the physical crowding properties of condensin and cohesin [30] to fold and unfold chromatin.”.

We also added two references:

[29] T. M. Cheng, S. Heeger, R. A. Chaleil, N. Matthews, A. Stewart, J. Wright, C. Lim, P. A. Bates, and F. Uhlmann, *Elife* 4 (2015).

[30] D. E. Anderson, A. Losada, H. P. Erickson, and T. Hirano, *The Journal of cell biology* 156, 419 (2002).

Comment 7: Importantly, regarding meta-TADs, the authors state that the hierarchical organization is a consequence of weak inter-connectivity properties. Personally, I am convinced that the hierarchical organization comes often from the fact that the chromosome is a single polymer and that, for this reason, TADs will have a decreasing contact probability in function of genomic distance, despite any special interactions. How could those views on the matter be conciliated?

>Answer 7: Meta TADs are off-diagonal blocks encompassing TADs, in which the encounter frequencies does not decay continuously, but rather in a stepwise fashion. In Fig. 2A-B we show an example of such organization (although not motivated by any biological example, it shows that these structure cannot be captured by a single decaying function, and are the direct consequence of inter-TAD connectivity (compare with Rouse $(|m-n|)^{-3/2}$, encounter probability decay).

We have already compared in Shukron et al PRE, the Rouse decay with decay generated by configuration connected by connectors and found a significant deviation. We further added now in Fig. 2B an EP generated for Rouse polymer to show the deviation, showing that it cannot account for meta-TADs. Thus this shows that without interaction, in our hand, it is not possible to recover the statistics of EP HiC data.

Comment 8

In particular I would like the author to consider matrix M of Eq. 6, that being a block matrix, neglects the link constraints between the frontiers of single-TAD Rouse chains: in this matrix the order of TADs on the diagonal does not matter, while this is generally not true in reality. What changes in the results if adjacent blocks would be connected by single links between their borders?

>Answer 8: The block matrix in Eq. 2 represents the linear backbone of the polymer. The order of appearance of matrices $[M^{(j)}]$ is actually important due to the inter-TAD connectivity associated to them (which is assumed to be non-vanishing, SI Eq. 50).

We do not impose strong connection between the ends of chain in the analytical derivation, but we do connect the ends in simulations. This result in the minor discrepancies between simulation and analytical results in Fig. 2B, which shows a jump discontinuity in the analytical curve (blue) at the boundaries between TADs.

We have now added in the Method section, subsection “RCL polymer model for multiple interacting TADs”, first paragraph, the sentence:

“Note, that during the numerical simulations, we connect the last monomer of block $[M^{(j)}]$ to the first one of $[M^{(j+1)}]$.”

In simulations we construct the polymer such that the ends of TADs are all connected using a harmonic spring with spring constant k .

Minor issues

Comment 9

There is a typo at the definition of NN: non-nearest neighboring should be non-NN considering the use of NN later in the manuscript. Otherwise better a less ambiguous rephrasing

>**Answer 9:** We have now corrected the definition in Method section, first paragraph to non nearest-neighboring (non-NN) and throughout the manuscript.

Comment 10

Before eq. 10: appendix 37 -> appendix Eq. 37

Answer 10: We have made corrections to the reference. Note, that we have made major revision and appendix has now been replaces with SI and equation numbering has been changes.

We thanks the reviewer for constructive comments.

Reviewer 1

Comment 1

The manuscript aims at reconstructing the three-dimensional structure of a fraction of the chromatin fiber from chromosome conformation capture experiments data. In the first part, the authors extend the randomly cross-linked (RCL) polymer model (also known as random-loop model) to capture the experimentally observed higher contact probability within topologically associated domain (TAD) in contrast to contact probability between different TADs. Each TAD i is modeled as a linear Rouse-like polymer backbone where a fraction $x_{i,ii}$ of all possible non-consecutive monomer pairs are cross-linked at random by other harmonic springs. Similarly, the contacts between TADs i and j are realized by a fraction $x_{i,ij}$ of randomly chosen harmonic cross-links. The authors use mean-field (MF) approximation to derive various quantities characterizing the conformation and dynamics of TADs, such as mean squared radius of gyration (MSRG), encounter (contact) probability (EP) and mean squared displacement (MSD). The MF approximation in essence means that instead of counting with the fraction of the cross-linked monomers, all monomers are considered cross-linked, however with a coupling constant that is $_$ -times smaller. Using this, Brownian dynamics and an assumption of dominating intra-TAD contacts over inter-TAD, the authors derive closed form expressions for the above-mentioned quantities. In the second part, the authors use their results to fit the parameters of their model to a chromatin segment representing three TADs at three different differentiation stages of the mouse embryonic stem cell.

Answer 1:This is summary contains some information about what we have done, but to make sure, we would like to provide further clarifications about what we have done here, that are not present in this summary: the most important contribution is a direct reconstruction of chromatin organization of several interacting TADs and their statistical properties, such as the radius of gyration and mean square displacement, that we compare in three stages of differentiation. We have now extracted the distributions of anomalous exponent for a given realization and use new data to validate our model.

The results presented here are new and have not been possible before because we have indeed made a tour-de-force by solving a reverse engineering computational problem about reconstruction of TAD from HiC.

This new method and data analysis bring to the community interested in chromosome organization, a new information and a new tool to analyze the chromatin dynamic organization throughout cellular differentiation. Consequently, our method allows to connect HiC with single particle experiment data in three-dimensional, by

allowing stochastic simulations of locus located inside a reconstructed TADs (see new data and simulations added, Fig 4-5).

Comment 2

The main ideas of the manuscript are written clearly with the aid of the appealing visual presentation. However, as detailed below, the extension of the RCL model represents a small improvement to the known model (introduced more than ten years ago) and does not seem to bring a substantial new viewpoint on the TADs or chromatin organization.

>Answer 2: We disagree with this view point, as revealed by the recent publications, polymer models are actually revealing new features of chromatin organization.

A few examples are:

- 1. Barbieri et al. 2014, scaling regimes based on concentration of diffusing binding molecules, PNAS.**
- 2. Pombo PNAS 2012 and 2017 (Nature) - three way genomeic loci encounters.**
- 3. Thirumalai 2018 (Nat. Com) - a recent paper of the same spirit of the current manuscript**
- 4. D. Jost and C. Vaillant, Nucleic Acids Research, 2018- the importance of long range connectors for epigenomic maintenance.**
- 5. L. Giorgetti et al Cell, 2014.**

First, the result we presented here is very far from being a routine extension of our previous model, it is new and based on a new algorithm that allows to find the number of added connectors, which has never been achieved in the past.

Our RCL model differs from that of the random loop model in two critical aspects:

- 1) the cross-links in our model are not dynamic as in the random loop model, and
- 2) we introduce heterogeneity in connectors within and between TADs, which is not achieved in the random loop model.

Our construction by itself can be considered as a breakthrough in polymer models and its application to chromatin representation. However, we should not underestimate the recent breakthrough of polymer models in the community of genetics and cell biology, which are published in general journals, because it targets a mixed audience of physicists, computational biologist, geneticists, and cell biologists.

Comment 3 Moreover, the validity of the model, its assumptions, and simplifications, are not assessed by detailed comparison with experiments and the reconstruction of the 3D structure is not compared to other methods or models.

>Answer 3: The goal of the present data analysis and methods is not to provide a spatial 3D structure of the chromatin, the interpretation of which is problematic in population HiC/5C. We provided here a tool to extract the average number of connectors within and between TAD, such that the steady-state encounter probability of the RCL polymer and empirical data agree. We further provide means of quantifying the structural changes of TADs as the cell undergoes differentiation. This is very different than the consensus reconstruction in the literature and was not achieved by other authors.

Therefore, unfortunately contrary to what the reviewer is asking there is at this stage no ground for comparison with other similar method, however, we have now added new data so we can use the result of the present approach to compare HiC and 5C data at two different resolution 6 and 10 kb. We have now perform this new analysis. The results are shown in the new Figure 4, which shows that the present

method can be used to compare different EP matrices, coming from different variant of experimental techniques.

Figure 4: **Comparing TADs reconstruction during cell differentiation between HiC and 5C.** **A.** Average number of connectors within and between TADs D, E, and F of HiC data [31] of the X chromosome binned at 10 kb, obtained by fitting the empirical EP with Eq. 10, 12, where TAD boundaries were obtained in [1] for mouse embryonic stem cells (mESC, left), neuronal progenitor cells (NPC, middle), and cortical neurons (CN, right). The average number of connectors within and between TADs are presented in each blue box. **B.** Mean radius of gyration (left) for TAD D, E, and F, throughout three successive stages of differentiation of the HiC data, with $b=0.18\mu\text{m}$ obtained from SI Eq. 62, and the compaction ratio (right, Eq. 15). **C.** Average number of connectors within and between TADs D, E, and F of the 5C data [1] of the X chromosome binned at 10 kb, obtained by fitting the empirical EP with Eq. 10, 12 for mESC (left), NPC (middle), and MEF (right). **D.** Mean radius of gyration (left) for TAD D, E, and F of the 5C data, and the compaction ratio (right, Eq. 15).

Comment 4

To summarize, based on the limited novelty and insufficient model validation I would suggest to reject the manuscript from the publication in Nature Communications. This does not mean that I do not consider the work interesting. I think it is important and interesting to have exactly solvable models and their various extensions, even if their direct applicability is somewhat questionable. From this point of view, after some restructuring and clarifications I think the manuscript would be more appropriate for a publication in a specialized journal that favors comprehensive subject exploration over novelty and paradigm shift. An option would be Physical Review E, where a close predecessor of this model was presented by the authors last year.

>**Answer 4** Following our answer above, we think that our explanations have not been adequate enough to express the novelty of the present results, so that the referee feels that we have done is not new enough. We could not find any references about published works where a similar approach has been developed, and more important has been applied to HiC data of differentiated cells, using the extracted physical contacts within and between TADs.

It should not be underestimated the vast application of polymer model to chromatin in the past 5 years (see reference in Answer 2) where polymer models are now a key player for understanding and reconstructing chromatin organization, especially when the HiC map can be converted systematically into a polymer structure. This structure can then be used to recover statistical properties. The difficulty is to find a systematic approach to convert EP into polymer model. We think we present in this manuscript such a generic approach that should brought to the attention of a vast majority of scientists in genetic, cell biology, computational biology, biophysics working on this question, because they could use and improve our method.

We have not seen before an equivalent approach and for that reason, we think that the present approach and results should be communicated to a large audience, including biologists and not simply for a restricted physics community which is only readership of Phys. Rev E. We did this mistake several times in the past and it took years for the biology community to realize the relevance of our findings.

In addition, we would like to say that a similar subject has been addressed by many theoretical works, based on polymer models, and data analysis published in Nat Com, NAR or PNAS with similar approaches (MSD analysis, polymer simulations, etc...), and we do not see why the present manuscript should receive a different treatment, under the reason that it contains a minor part about analytical formula, which is not the point of the manuscript. Examples are :

Javer A, Long Z, Nugent E, Grisi M, Siriwatwetchakul K, Dorfman KD, Cicuta P, Cosentino Lagomarsino M., Short-time movement of E. coli chromosomal loci depends on coordinate and subcellular localization. Nat Commun. 2013;4:3003.

Javer A, Kuwada NJ, Long Z, Benza VG, Dorfman KD, Wiggins PA, Cicuta P, Lagomarsino MC. Persistent super-diffusive motion of Escherichia coli chromosomal loci. Nat Commun. 2014 May 30;5:3854.

Jost D, Carrivain P, Cavalli G, Vaillant C., Modeling epigenome folding: formation and dynamics of topologically associated chromatin domains. Nucleic Acids Res. 2014

Barbieri M, Chotalia M, Fraser J, Lavitas LM, Dostie J, Pombo A, Nicodemi M. Complexity of chromatin folding is captured by the strings and binders switch model. Proc Natl Acad Sci U S A. 2012

And many more.

We have now added new experimental HiC data from our experimental collaborators to strength the methodological part (see Figure 4 in the answer to comment 3 above). This should emphasize the applicability of our method.

Major questions and issues in order of appearance

Comment 5

In the RCL model and its present extension the chromatin fiber is assumed to be phantom: there is neither (i) excluded volume interaction nor (ii) any kind of topological constraints that would prevent co-localization or self-crossings of different segments. In dilute conditions in theta solvent this could be a good approximation, but what makes these to be valid assumptions in the case of highly compacted chromatin structure within TADs? This affects not only the statistical properties of the conformation, but also the dynamics of the polymer.

>Answer 5: The discussion of phantom vs. non phantom chain, exclusion etc do not apply to the first part of the manuscript about chromatin reconstruction because the difficulty in that part is to identify the number of inter and intra TAD connectivity. This part by itself is new and allows the coarse-grained chromatin reconstruction. For the second part of the model, which consists in comparing new analytical results that have never been obtained before with simulation, we indeed neglected exclusion forces.

To show that this exclusion effect does not affect significantly our results, we have now follow the reviewer recommendation and added to the Supplementary Information, subsection F, “Heterogeneous RCL model with volume exclusion forces”, the results of a simulation where we now calculate the statistical quantities of interest with exclusion forces, using two values of the exclusion radius, namely $c = 40$ nm, and $c = 67$ nm.

We find that the theoretical statistical properties are in agreement with stochastic simulation of system Eq. 7 up to an exclusion volume of 40 nm. We added two supplementary figures, SI Fig. 2 and 3, summarizing these results, and shown below:

SI Figure 2: **Statistical properties of the heterogeneous RCL polymer with added volume exclusion.** **A.** Encounter frequency matrix of a polymer with three TAD blocks (TAD₁, TAD₂, TAD₃) of $N_1=50$, $N_2=40$, $N_3=60$ monomers is computed from 10,000 simulations of the RCL polymer using Eq. 7 (main text) with the added volume exclusion potential 58, and $\Delta t = 0.01$ s, $D=8 \times 10^{-3}$ $\mu\text{m}^2/\text{s}$, $d=3$, $b=0.2$ μm , $\varepsilon=0.02$ μm , and $c=0.04$ μm . The number of added connectors within and between TADs appears in each block. Three distinct diagonal TADs are visible (red boxes) where secondary structure appears (black lines) due to weak inter-TAD connectivity. **B.** Encounter probability (EP) of the heterogeneous RCL described in panel A, where the simulated EP (red) agrees with the theoretical one (blue, Eqs. 42, 52). We plot the EP of the middle monomer in each TAD: monomer r_{20} (top left), monomer r_{70} (top right) and monomer r_{120} (bottom left), where TAD boundaries are in vertical dashed lines. **C.** Averaged mean squared displacement of monomers in TAD₁ (blue), TAD₂ (orange) and TAD₃ (yellow), using simulations of RCL polymer, as described in panel A: simulation (continuous line) vs theory (dashed, Eq. 54).

The simulation mean radius of gyration for TAD₁, TAD₂, and TAD₃, are 0.179, 0.134, 0.168 vs. 0.178, 0.132, and 0.167 μm , from theory (Eq. 8, main text), respectively. An average error of 1%.

By increasing the exclusion radius to $c=0.0667$ μm , we start seeing several discrepancies between the theoretical and simulation encounter probability and mean square displacement, as shown in SI Fig. 3, below.

To conclude, our model captures the phenomenology that would be obtain by an exclusion of 40 nm, but not above 67 nm.

SI Figure 3: **Statistical properties of the heterogeneous RCL polymer with added volume exclusion.** **A.** Encounter frequency matrix of a polymer as in Fig. SI Fig. 2A, with radius of exclusion $c=0.067 \mu\text{m}$. The number of added connectors appears in each block. Three distinct diagonal TADs are visible (red boxes) where secondary structure appears (black lines) due to weak inter-TAD connectivity. **B.** Encounter probability (EP) of the heterogeneous RCL described in panel A, where the simulation EP (orange) deviates from the theoretical EP (blue, Eqs. 42, 52), plotted for the middle monomer in each TAD: monomer r_{20} (top left), monomer r_{70} (top right) and monomer r_{120} (bottom left), and TAD boundaries appear in dashed red lines. **C.** Averaged mean squared displacement of monomers in TAD₁ (blue), TAD₂ (red), and TAD₃ (yellow), using simulations of RCL polymers, as described in panel A: simulations (continuous line) vs. theory (dashed, Eq. 54), that do not agree for $t > 5$ s.

The results of the simulation mean radius of gyration for radius of exclusion 67nm , for TAD₁, TAD₂, and TAD₃, are 0.192 , 0.144 , and $0.181 \mu\text{m}$ vs. 0.178 , 0.132 , and $0.167 \mu\text{m}$ from theory, for TADs 1 to 3, respectively, with an average error of 8.5% .

To conclude, these new results show that for $b=0.2 \mu\text{m}$, $c=0.0667 \mu\text{m}$, we can expect a deviation from the theory presented in this paper. But, and this should be considered a remarkable achievement: the present theory, which neglected initially the exclusion volume is actually quite accurate for polymers with an exclusion radius of up to 40 nm , which is a reasonable scale.

We have now added in the Results, subsection “Statistical properties of the heterogeneous RCL and numerical validations“, one before last paragraph:

“In addition, we found that adding an exclusion forces with a radius of 40nm did not lead to any modifications of the statistical quantities defined above (see SI Fig. 2 compared to Fig. 2C). However, when the exclusion radius increases to 67nm , deviations started to appear (SI Fig. 3). To conclude, an exclusion radius of the order of 40 nm , also used in [29], is consistent with the physical crowding properties of condensin and cohesin [30] to fold and unfold chromatin”.

We also added two references:

[29] T. M. Cheng, S. Heeger, R. A. Chaleil, N. Matthews, A. Stewart, J. Wright, C. Lim, P. A. Bates, and F. Uhlmann, *Elife* 4 (2015).

[30] D. E. Anderson, A. Losada, H. P. Erickson, and T. Hirano, *The Journal of cell biology* 156, 419 (2002).

Comment 6

The mean-field approximation is also discussed in one of the early works on random-loop model (Bohn et al PRE (2007) - Ref [15] in the manuscript). It was shown there numerically that the size of a chromatin domain could be fitted with the model with MF approximation, but it also says: "Although one could fit the data with these averaged attraction potentials, we see no biological reason for such a potential to exist in the cell." Indeed, the MF represents interactions (although weak) between all the monomers within the domain that is easily 1_m large. What would be the physical and biological mechanisms behind such long-range attractions? I understand that this assumption is necessary for having an exactly solvable model, but isn't this on the expense of its direct applicability? i.e. What is the validity of the mean-field assumption? (See also related question 9 below.)

>Answer 6: The nice model by Bohn et al., (2007) introduced dynamic linking and thus differ from our model, where the cross-links are fixed for each realization. In addition, the paragraph above mentions several questions that we will treat one by one:

1. The mean field approximation is used here precisely because the 5C/HiC map is by definition an average over cell population, which is equivalent to average many polymer realizations. There are actually little assumptions about using mean-field here, because this is precisely the equivalent of what is done in the statistical analysis of the experimental data. The mean field approximation is simply used here to derive analytical results for system Eq. 4 of the main text, in which no averaging is made. The derivation of these analytical properties from the mean-field to a single realization was done for one TAD case in Shukron and Holcman PRE 2017. We could not do it here.

2. There are no attraction potentials here, which is a physical abstract concept. We clearly specify here that a connector represent a binding molecule (such as CTCF, cohesin) and the model is built upon finding minimum number of the connectors that represent the data. Contrary to previous work, including the interesting work of Bohn et al 2007, the present model is parsimonious and thus reveal how random cohesin binders influence the folding of the chromatin.

3. Following point 2, there are here no long-range interactions of potential attractors. Each connector is a specific force acting on a pair only and nothing else.

4. The specificity and quality of the present model is not to be solvable but to allow reverse engineering problem, that is: given the connectivity map find the minimal number of connectors such that the average realization matrix for the encounter probability of simulation and the data matches (their difference in any norm is minimal).

To conclude, the mean-field procedure is not an assumption but a method of averaging, which is exactly reproduce the treatment of the empirical data. There are no generic potential wells but specific monomer-monomer interactions and the difficulty

we have solved here is to identify the number of these interactions and not much about the analytical solvability of the present model.

Comment 7

Along similar lines, as far as I understand, the 5C data represent the contact probability as a population average over many cells, while contacts within single cell can exhibit large variability. Why should we hope that for the RCL model the procedures of averaging and dynamical evolution commute? I.e. evolving an average system is not the same as averaging over evolved systems. (See also related question 9 below).

>Answer 7: As we will answer in comment 9, we think there is a misunderstanding here. We did not simulate the average system, rather we simulate the full stochastic equation 4 that do represent single cell realization and not averaging. With this clarification comment 7 is now moot. The derivation of the statistical properties of the RCL in one TAD case, with no averaging was presented in Shukron and Holcman, PRE 2017.

Comment 8

It would help to clarify what are the different scaling regimes of the MSRSG with polymer length. From the presented results, the older paper (Ref [15]) as well as similar result in the authors preceding paper (eq. (33) of Shukron and Holcman PRE (2017) - Ref [10] in the present manuscript) it's clear that the MSRSG initially grows with N , but then the growth is decreasing. This is because adding one monomer adds other N attractive springs i.e. the number of cross-links scales with N^2 , while the length of the polymer is only linear in N . For fixed spring constants this would eventually lead to a collapse in the absence of other repulsive interactions. To my understanding, the collapse in the present model is prevented by the fact that the spring constants decrease as N^2 . Could such a fine tuning be present in a biological system? What would be its physics? Moreover, the above-mentioned scaling could give an answer to the range of polymer lengths the model could be applicable to, by comparison of different scaling regimes of chromatin fiber. What are the lower and upper bounds on the lengths of the polymer, where the model is assumed to work? How would it behave under a different coarse-grained representation, e.g. when a smaller fraction of chromatin is represented by a single bead? This deserves some comments.

>Answer 8: Before answering the general question about relating the total length N to the size of the chromatin, we would like to clarify our model because comment 8 contains a description that does not apply to the present manuscript:

There is no connection between the number of added connectors and the spring constant. However, we recall that using the identity for the spring constant

$$k/\gamma = dD/b^2,$$

where d is dimension, D is the diffusion coefficient, and γ is the friction coefficient, performing fitting at various coarse-grain scales of the 5C or HiC data, the value of b can be calibrated at each scale, which imposes a value for the spring constant. We have now added in the supplementary Eqs. 61, 62, the procedure of computing the value of b , the mean square length of connectors, from a given coarse-grained scale, e.g. 6kb to e.g. 10 kb. The computation is based on the invariance of the MSRSG across scales. Note, that the number of added connectors might also change with the scale (SI Figs. 5 and figure 4, main text), thus Eq. 54, states the condition of constant MSRSG, and requires

$$(61) \quad \frac{\langle R_{6kb}^2 \rangle^{(i)}}{\langle R_{10kb}^2 \rangle^{(i)}} = 1, \quad i = D, E, F.$$

For each TAD. This computation leads to the new value of b at 10 kb, stated in SI Eq 62,

$$b_{10kb}^{(i)} = b_{6kb}^{(i)} \sqrt{\frac{(1 - \xi_{ii}^{(2)})(\zeta_0^{(i)}(\Xi^{(2)}) - \zeta_1^{(i)}(\Xi^{(2)}))}{(1 - \xi_{ii}^{(1)})(\zeta_0^{(i)}(\Xi^{(1)}) - \zeta_1^{(i)}(\Xi^{(1)}))}} \quad (62)$$

with superscript (i) indicates the value of b for TAD i.

We emphasize that these changes have no consequences for the reconstruction of the polymer from HiC matrix, but only to the value of the spring constant.

We would like to recall the procedure of reconstruction. Once a HiC resolution has been defined, a matrix is generated which represent a chromosome at a given scale; then the user defines the coarse-grain scale of the polymer model, such that a monomer represents x kb. Once a monomer is specified it leads to the total number of monomers that will be used such that (1) to represent the length of the chromatin.

Thus the analysis would be valid for this specific size, and clearly choosing a different resolution could lead to a different representation that would be defined by our procedure of selecting the number of connectors.

In general, there is no clear relation between the number of connectors to be added and the coarse-grain resolution. In the case of a single TAD, we have already found (Shukron 2017, PRE) how the number of connectors is changing with respect to the resolution $\sqrt{N\xi}$. Indeed, we kept the MSRG fixed for any given N . This leads to the relation that b^2 is proportional to $\sqrt{N\xi}$. Here, for a polymer representing multiple TADs, this relationship is given in Eqs. 61, 62 above.

Comment 9 The scaling exponent of the average contact probability with the genomic separation length of the polymer is a good measure to distinguish between different polymer models [see e.g. S. Sazer, H. Schiessel, Trafic (2017)]. Is the exponent and the behavior predicted here (eq. 15) consistent with experiments? I understand one can fit the Eps to reproduce the behavior to some extent, but I also expect that the exponent will not depend on this fit.

>Answer 9: In general the decay of the encounter probability is computed either across the entire genome or inside a single TAD. What we have proposed here is to extend this encounter probability across TADs. Being able to distinguish between polymer models is our recent contribution through the beta-polymer model or the RCL polymer [Amitai, PRE, 2013], where we could convert the power law into local forces between monomers. In Shukron et al, PRE 2017, we have shown that the variance of the distance between monomers inside TADs can be approximated for low N_c , number of added connectors, by

$$\sigma_{mn}^2 = 1 - \text{Exp}(-|m-n|(2N_c/N)^{0.5}),$$

and therefore, the encounter probability in three dimensions behaves as

$$E_p \sim (\sigma_{mn}^2)^{-3/2} = 1 / (1 - \text{Exp}(-|m-n|(2N_c/N)^{0.5}))^{3/2}$$

Therefore, for large inter monomer distance, $|m-n| \gg 1$, the EP is a constant. For small $|m-n|$, the behavior similar to the Rouse model, i.e $E_p \sim |m-n|^{-3/2}$.

Comment 10

Is the dynamics prediction (eq. 17) and Fig 2C consistent with any experimental evidence?

>**Answer 10:** Yes. The anomalous exponent of the scale of 0.4 was previously measured using SPTs by several groups such as Gasser's group, Bistriky group, P. cicuta's group, see for example: Amitai et al. Cell report, Seeber et al, Nat Struct 2017, and references

We have now added the values of the anomalous exponent α , resulting from a fit βt^α to the MSD curves.

Comment 11

Are the values chosen for the diffusion coefficient and the standard deviation of the spring connector b realistic? What would such a coupling $b = p3_m$ represent? How do the model properties depend on this choice?

>**Answer 11:** The values we have previously chosen for numerical simulations were such that $dD/b^2 = 1$. These values were used for validation purposes only in Fig. 2, and do not correspond to any biologically relevant scenario.

Thus, we have now simulated the heterogeneous RCL polymer with values taken from literature $D=0.008 \mu\text{m}^2/\text{s}$, $b=0.2 \mu\text{m}$ (Shukron, PLoS Computational Biology, 2017) and report the results in a new Fig. 2, below. Note that there are little changes compared to the initial figure, because we are simulating steady-state configurations, that do not depend much on time scales.

Figure 2: **Statistical properties of the heterogeneous RCL polymer.** **A.** Encounter frequency matrix of a polymer with three TAD blocks (TAD₁, TAD₂, TAD₃) of $N_1=50$, $N_2=40$, $N_3=60$ monomers, computed from 10,000 simulations of Eq. 7 with $\Delta t = 0.01$ s, $D=8 \times 10^{-3} \mu\text{m}^2/\text{s}$, $d=3$, $b=0.2 \mu\text{m}$, $\epsilon=0.02 \mu\text{m}$. The number of added connectors appears in each block. Three distinct diagonal TADs are visible (red boxes) where secondary structure appears (black lines) due to weak inter-TAD connectivity. **B.** Encounter probability (EP) of the heterogeneous RCL described in panel A, where the simulation EP (orange) is in agreement with theoretical EP (blue, Eqs. 10, 12), plotted for the middle monomer in each TAD: monomer r_{20} (top left), monomer r_{70} (top right) and monomer r_{120} (bottom left). **C.** Average mean square displacement of monomers in each TAD of the heterogeneous RCL

polymer, simulated as described in panel A, simulation (full) versus theory (dashed Eq. 14) are in good agreement for time up to 25 s.

Comment 12

Why is the encounter distance chosen so large i.e. $b/10$? Does the simulation encounter frequency matrix depend on this choice?

>Answer 12: We have taken epsilon equals to $b/10$, to be tenth of the average distance between adjacent monomers. In practice, this value affect the number of simulation time needed to reach the encounter maps in Fig. 2.

Smaller values will require more simulations. The steady-state encounter probability should not be affected, unless epsilon is of order $>b$. We have now used more realistic parameters and produced a new Fig. 2 and 3: we use $D=8 \times 10^{-3} \mu\text{m}^2/\text{s}$, $d=3$, $b=0.2 \mu\text{m}$, and the encounter distance of 40 nm. The distance of 40 nm was found D. E. Anderson, A. Losada, H. P. Erickson, and T. Hirano, *The Journal of cell biology* 156, 419 (2002).

To characteristic the size of cohesin/CTCF molecule and was also used in T. M. Cheng, S. Heeger, R. A. Chaleil, N. Matthews, A. Stewart, J. Wright, C. Lim, P. A. Bates, and F. Uhlmann, *Elife* 4 (2015), in simulation of chromatin dynamics.

We have now added these two research articles as references 29-30 in our manuscript. For the results of the heterogeneous RCL simulation with the new parameters see the figure in Answer 11 above.

Comment 13

The description of the simulations is not completely clear. Firstly, the simulations are running within the MF approximation i.e. using eq. (9). Wouldn't it be better to run it without the MF approximation (because we don't need the MF for numerics) and compare it with the analytical results with MF? This would to some extent elucidate whether MF makes sense.

>Answer 13: We think that indeed there is a misunderstanding here probably due to our lack of explanation. This misunderstanding stems from our mistake in the reference to the correct system we simulate. We have referred to the MF system SI Eq. 12 instead of system Eq. 7 (main text). Therefore, we do not run the MF-approximation in Fig. 2 and Fig. 3 as the reviewer understood. We have actually ran the full stochastic equation for a given G configuration. Later on, in Fig. 2, we then averaged over many realizations to compute the statistical quantity of interest (MRG, encounter probability and so on).

To conclude, it is correct that our analysis does make sense and justify our analytical approximation for the computation of the different statistical quantities (both numerical and analysis results coincide).

Comment 14

Secondly, the authors say "The connectors are placed uniformly in each A_i ..." do they mean uniformly or uniformly-randomly? This is not clear, because eq. (9) suggests connectors between all the pairs of monomers.

>Answer 14: We apologize for the confusion. We are not simulating SI Eq. 12 but Eq. 7, main text, where the connectors are placed randomly with a uniform distribution (by definition the number of connected pairs is characterized by the parameter Ξ). We have now explicitly mentioned that we have simulated Eq. 7 in the caption of Fig. 2.

Comment 15

Thirdly, the relation between the time step used (dt) and a natural time scale of the model ($D=b^2$?) should be clarified.

>**Answer 15:** To compute the encounter probability matrix, the exact time scale does not matter much, because the probability does not depend on any dynamic factor such as diffusion or any time scale. For that reason, we chose initially the following parameters $D=1 \mu\text{m}^2/\text{s}$, $dt=0.1\text{s}$, $b=\sqrt{3}$. However, for the computation of the mean squared displacement, the time scale does matter. We agree that parameters do play a role, the present figure (Fig. 2C) is used as a proof for the agreement between simulation and our analysis.

We have now redone Fig. 2C using the diffusion coefficient and Δt extracted from (Amitai et al, Cell report 2017) and we found similar result. The new Figure 2 is attached in the answer to comment 11, above.

Comment 16

Lastly, the time step in the text $dt = 0:05\text{s}$ is not consistent with the one stated in the caption of Figure 2 $dt = 0.1\text{s}$.

>**Answer 16:** Thanks. We have now generated a new Fig. 2, and corrected the typo (see also figure in Answer 11).

Comment 17

Why is there no comparison for the MSR_G between the simulations and the theory?

>**Answer 17:** First, note that in the new version of the manuscript we compute the mean radius of gyration (MRG) instead of the MSR_G. The comparison of the three MRGs of TAD₁, TAD₂, and TAD₃ between theory and simulations, corresponding to the system in Fig. 2, already appeared in the previous version of the manuscript in Result section we previously wrote:

Finally, the MSR_G for TAD₁, TAD₂ and TAD₃ is 2.42, 1.5, 2.15 μm (simulations), compared to 2.38, 1.31, 2.09, respectively, obtained from expression 10, which are in good agreement.”

We have now changed the value of the diffusion coefficient and b (see Fig 2, in answer 11) and computed the MRG. We now report in the Result section, subsection”statistical properties of heterogeneous RCL and numerical validations”, the comparison between the mean radius of gyration computed from stochastic simulations vs. Theory:

“ Finally, the mean radius of gyration $\langle R_g^{(i)} \rangle$ for TAD₁, TAD₂, and TAD₃ are 0.031, 0.017, 0.027 μm (simulations), compared to 0.031, 0.017, 0.027 μm , respectively, obtained from expression 8, which agree”.

Comment 18

Why are the 5C data coarse-grained to 6kb before fitting? As far as I understand this is to “smooth out” the peaks found in the data to resemble the model's EP. But doesn't this say simply that the model does not represent the data well and therefore should not be used for this purpose? What predictive power has it then? (See related questions 4 and 5)

>**Answer 18:** The number of long-range peaks that were discounted by the coarse graining procedure at 6 kb is around 10 per TAD. These peaks could represent statistical error or long-range interactions that we do not account for here.

We have already described a method to account for these peaks (PLoS 2017, Shukron). In that paper, we have assessed the consequences of disregarding long-range peaks. We found, for example, that it modulates the encounter probability by few percents. Thus we decided not to include them here. In addition, we emphasize that at

6kb resolution, the peaks are smoothed out and thus the analysis we present reflect the organization of the chromatin at this scale. It is completely possible that a refined coarse-graining could reveal more subtle structures, which could be the focus of further studies.

Comment 19

It was found that “the MSR_G increases and decreases in correlation with the acquisition and lose of connectors in all TADs”. This seems obvious. Is there a reason why it is not a trivial result? If so, this should be stated. (lose ! loss)

>Answer 19: Had we considered only intra-TAD connectors it would have been indeed an obvious statement that the MSR_G increases with the loss of connectors. However, because we account for inter-TAD connectors, the increase or decrease of the MSR_G with the number of connectors is not trivial. Indeed, adding connectors from opposite TADs can pull the mid TAD in one direction, while adding other connectors can pull in other direction.

Note that it is not a priory predictable how the MSR_G will evolve (Fig. 3) by adding connectors. To answer this question, we derived an expression (Eqs. 8, main text, and subsection B in the SI) for the MSR_G for each TAD based on intra-and inter-TAD connectivity.

We have now edited in the Result section “Reconstructing genome reorganization with multiple TADs from the 5C data during cellular differentiation“, third paragraph, the sentence:

“We found that the MRG can both increase and decrease depending on the number of connectors within TADs, but is also affected by inter-TAD connectivity, as revealed by Eq. 8.”.

Comment 20

The presentation of the resulting MSR_G from the simulations is a bit confusing. Firstly, the numbers are presented as e.g. $b^2 \mu m^2$ which is inconsistent with MSR_G having units of μm^2 and b having units of μm . I presume the μm unit is just a typo and the authors meant e.g. b^2 . More importantly, MSR_G results are compared only with random walk model and no connection to experiment is made. Plugging in the value of b gives MSR_G about $6 \mu m^2$, which seems different from the Fluorescence In-Situ Hybridization (FISH) experiment performed on the very same TADs (Nora et al. Nature 2012 - Ref. [1] in the manuscript), where they found a 3D distance below $1 \mu m$ - see TAD E in Fig. 1C and D of that publication.

>Answer 20: We specifically computed the ratio of the MSR_G for TAD to that of the linear Rouse polymer with the same number of monomers as the RCL, to avoid estimating the parameter b (standard deviation of the length between two neighboring monomers). Indeed, it is not clear in general how to estimate b from these Hi-C data.

However, we do not need such an estimation because the ratio provides information about different TADs. The choice of b from Fig. 2 is used only to calculate the probability map, which is independent of the dynamics.

We have now clarified this point by running a new simulation (Fig. 2) where we used the value of $b=0.2 \mu m$. We have now redone Figure 2 and 3 to account for parameters estimated from literature. Computing b for simulations and computations in other scales (SI Figs 2-7) is done using SI Eqs. 61-62, as appear in the answer to Comment 8 above. The new Figure 2 is presented in Answer 11 above. We computed in Figure 3 the mean radius of gyration instead of the MSR_G computed in the previous version of the manuscript, which resulted in new panel C in Fig. 3, below

Comment 21

The equation (18) seems to depend on the number of monomers used and therefore on the chosen resolution. Why is it interesting to present? What is behind the resulting huge compaction ratios - is it only different scaling of MSRG with number of monomers, or the negligence of some other effects such as the excluded volume interactions?

>Answer 21: The compaction ratio (Eq. 15, main text) is now plotted Fig. 3C: it is important to note that this ratio depends on the number N of monomers, because we are comparing the RCL of length N with a Rouse polymer of the same length.

The dependency with N represents specifically the radius of gyration of the Rouse polymer $\langle R_g^2 \rangle = Nb^2/6$. The large compaction ratio comes from the comparison between the volumes of the balls associated with the radius of gyration (elevated to the cube). We now computed both the mean radius of gyration and the compaction ratio using the new parameter $b=0.2 \mu m$ for the 5C data of the X chromosome, TADs D, E, and F, corrected a numerical error, and produced a new Figure 3C for the compaction ratio throughout differentiation, attached below

In addition, we have simulated the heterogeneous RCL polymer with three interacting TADs, where we add volume exclusion pair-wise between monomers for two values of the exclusion radius: $c=40$, and $67nm$. We find that the expressions we derived for the encounter probability (Eq. 10-12, main text), radius of gyration (Eq. 8) is in agreement with numerical simulations for exclusion radius of up to $40 nm$. This result is summarized in the Supplementary Information SI Fig. 2 and SI Fig. 3 attached in answer to Comment 5 above.

Comment 22

The authors claim “Inter-TAD connectivity determines the compaction of TADs and therefore recovering their exact number is a key step of the reconstruction method and allows to precisely recover genome organization from 5C data.”. In the light of the assumptions of

the model and the absence of its physical microscopic understanding, this seems to me as an excessively strong statement.

>Answer 22: We meant here that inter-TAD connectivity cannot be neglected for reconstructing the properties of the genome, as shown in Fig. 3. It should insist that to our knowledge, this is the first time that inter-TAD connectivity is used to reconstruct chromatin and we wanted simply to emphasize such progress.

We have now clarified the sentence: “Inter-TAD connectivity affect the compaction of TADs and can now be used to reconstruct the genome organization from 5C data”.

Comment 23

The present method is claimed to be a new tool to reconstruct chromatin structural reorganization from contact probability matrices. However, no comparison with other such methods is shown and no discussion on advantages of the present tool over others is presented. An example of such tool is e.g. [L. Rieber, S. Mahony, Bioinformatics (2017)], which cites (e.g. section 2.4.) at least about ten other such tools.

>Answer 23: The comparison here is not accurate because the bioinformatic tools used in the Refs mentioned above are not about polymer models and are not relevant to describe the dynamics and statistical quantities of interest.

The focus of the method we are offering here is not about the 3D spatial organization, which is the case of minMDS [L Rieber, S Mahony, 2017] but with extracting measurable quantities such as the number of binding molecules, to reproduce the steady-state encounter probability in the 5C maps. We provide here a computational and statistical framework to extract the properties of the chromatin directly from the 5C, and we are not giving any consensus 3D structure. To conclude, we do not think in this already long manuscript to compare the chromatin representation we obtain with structural reconstruction. It would clearly be an important subject for a new study about comparing all polymer methods that is currently emerging from us, Thiromalai group, D. Marenduzzo, Jost, etc...

Minor issues

Comment 24

There is a typo in the equation for the connectivity matrix B^G_{mn} . The sum should go over $j \neq m$ in the case $m = n$

>Answer 24: We have now corrected the index of summation in the definition of B^G (below Eq. 3) to $j \neq m$

Comment 25

There is typo in the encounter frequency plot label in Fig. 2A and missing units for the quantities D and b in the caption of Fig. 2A.

>Answer 25: We have corrected the Z label to “encounter frequency” in Fig. 2A and have added the units $D=1\mu m^2 /sec$ and $b= 0.2 \mu m$ in the caption of Fig. 2A.

Comment 26: There are missing units in the MSD plot in Fig 2C

>Answer 26: We have now added the units of μm^2 to the y axis label in Fig 2C.

Comment 27

The units in Fig 3C are wrong.

>Answer 27: We have corrected the units to μm in Fig. 3C and plotted the mean radius of gyration rather than the MSRG.

Comment 28

There is extra "the" in the first line below eq. (34).

>Answer 28: We have removed the word "the" from the sentence (below SI Eq. 20, in the new version of the manuscript) to read "where $G_{ij}(m,n)$ is a $N_i \times N_j$ matrix made of zeros except for $G_{ij}(1,1)=1$."

Comment 29

There is missing "of" in the first line below eq (48).

>Answer 29: Eq 48 is now SI Eq 30 in the new version of the manuscript. We have corrected the sentence below SI Eq. 48 to: "We recall that the variance of the normal coordinates is..".

Comment 30

There is an extra "gku698" in Reference [21]

>Answer 30: We have now corrected the reference (Ref. 23 in the new version of the manuscript) to: D. Jost, P. Carrivain, G. Cavalli, and C. Vaillant, Nucleic Acids Research , (2014).

We thanks the reviewer for his comments that helped us clarify the present manuscript.

Reviewers' comments:

Reviewer #1 (Remarks to the Author):

Referee Report on NCOMMS-17-34203 Statistics of chromatin organization during cell differentiation revealed by heterogeneous cross-linked polymers

I have read the replies of the authors and the new version of the manuscript. I greatly appreciate the amount of work and effort put in to improve the work. Some of the major issues were solved to my satisfaction. This includes Comments/Answers (CA) number 7, 9, 13, 14, 18, 19 and 21. Particularly, I appreciate, the authors clarified that the simulations did not use the mean-field (MF) approximation, but were run with random realizations of the connector placements.

To my opinion the work presents two classes of results. First, it is the extension of the RCL model and the mean-field analytical results that agree with the numerical simulations of the specific polymer model. I found this result very nice, definitely worth publication, but probably will be appreciated by specialist (polymer/physics) readership only. The second class of results relates to the application of the model to the biological data, to reveal the statistics of chromatin organization. At first I state these with a brief description why I do not find them completely convincing. More details are then provided below in my comments on the answers provided by the authors.

1. Prediction of MRG - the numerical values obtained from the model seem now to be closer to the experiment. What is different here from the previous version of the manuscript? The difference is probably related to the (fitted) value of b , which is also somewhat poorly justified.
2. Compression ratios - as the authors state a way to partly avoid the difficulties of the MRG comparison is to state only relative compressions with respect to linear polymer. Truly, these are giving some qualitative predictions, as shown on the analysis of the experiment replicas (which I appreciate). However the quantitative prediction is a bit problematic as shown on comparison of HiC and 5C data.
3. Dynamic data and exponents - I found the presentation unclear and hard to assess its validity.
4. Number of connectors - to my understanding this has been stressed as the important result in several answers. My main problem with it is the following: Are these connectors real molecules or they are just an abstract way to model organization statistics?
(i) If they are real (which was suggested by saying they could be cohesin or CTCF) then why the size b does not match the size used for the steric repulsion interaction ($40nm$) and why does their number depend on the scale? If they are not real, why should we be interested in their number? (ii) I do not

find the assessment of the error bars of the number of connectors fair. For example, I found the following statement misleading: "We found that the number of added connectors in replica 1 and 2 are in agreement.... most significant difference was 5 connectors...". The 5 connectors is about 30% difference and in the other cases in Fig S4 where the difference is only one or two connectors - the relative error is in many cases also around 30% and in a few cases almost 50% as the number changes e.g from 4 to 7 connectors. Similarly when Hi-C and 5C or when 3 and 6 TADs are compared. These relative differences are about the same as those found between different differentiation stages. How can be the authors then sure that we see for the differentiation stages is a real effect and not just the lack of precision of the method?

In the following I comment on the answers that I found unsatisfying or only partially satisfying and explain my reasons.

- CA 2 and 4: I believe the authors misunderstood my comment. I was not saying polymer models are not useful. They are. I was trying to explain that this work does not convincingly show, how novel or different is the present model from its predecessors e.g. what advantage it gives over the old ones.

The works of Barbieri et al or Jost et al mentioned or Fudenberg et al, describe different physics (paradigm shift) that can be behind the organization (diffusing binders, epigenomic interactions, active maintenance). In this, although very interesting work I miss such a shift. The distinction from the Random Loop Model (RLM) was mentioned in the answer - static crosslinks here vs. dynamic ones in RLM, but I haven't seen this in the manuscript and an assessment of what advantage the permanent crosslinks have.

I understand and agree with the authors that they want to present their results for a mixed audience.

- CA 3: By "3D structure" I meant statistics of the ensemble of 3D structures, such as MRG of different segments. Therefore I don't understand why the authors claim that there is no ground for comparison with other methods or experiments. For example the work Di Pierro et al (PNAS 2017) also presents a method to obtain spatial distances from ChIP-Seq and Hi-C and tested their method with fluorescence experiment. Another example is Stevens et al (NAT 2017) who are also extracting RG from single cell Hi-C and compared with

fluorescence experiments. Yet another is Li et al (TCBB 2018). A comparison with experimental fluorescence data of Nora et al who have measured the spatial distances for the same chromosome segment as in the present manuscript is missing.

I stated my objections on the number of connectors above.

Yes, I appreciate the investigation of the structural changes during differentiation, if the model is sufficiently justified.

- CA 5: I appreciate the simulation with exclusion volume interaction although one can of course ask what is the range of c for which it is negligible. More importantly, there was no comment on the topological constraints.
- CA 6: Thank you for the explanations. I don't agree that there is no long-range interaction. With high b the interaction is quite long-range (e.g. for the HiC data $b = 1.8\mu m$), but specific to certain monomers. If the connectors represent the cohesins, why the b does not match the c value that characterizes the size of the molecule?
- CA 8: I found the fitting procedure still unclear. b is tuned to maintain MSRG the same for all scales. As D is fixed, the value of b affects the value of k and the number of connectors is different for different scales. This suggests then that these connectors are not real or there is some renormalization scheme under the hood which should be clarified.
- CA 10, 11 and 15: I found the presentation of the MSD data and procedures unsatisfying. Firstly, if exponents are extracted, the plots of MSD vs time should be in log-log to see the robustness and range of the slope. Second, the equation (14) holds in intermediate times and represents the change of the exponent of 0.5 (second term) at intermediate times to the exponent 1 (first term) at late times. This means also that the prefactors affect the effective slope measured (so the chosen values of b or D are important). Third, I don't understand why on one hand the structure is represented by the steady states, but the MSD is evaluated in intermediate times and why these times are those below 25s. Fourth, is there a justification for the value of the diffusion coefficient $D = 0.008\mu m^2/s$? In answer 11 it is stated that the change of the parameters (D and b) has little effect on the figure, because the steady states don't depend on time scales. The intermediate MSD scaling however does depend on the time scales, so this match should be explained.
- CA 12: Why c is different from b ? The authors have the simulations with other parameter sets so can asses and present the sensitivity of the model to these choices.
- CA 17 and 20: I apologize that I missed the MSRG comparison between theory and simulations in the first version of the paper. Now the values 0.21, 0.19 (note there is a typo in the manuscript stating 0.9) and 0.2 seem to be close to the FISH experimental values mentioned before (Nora et al). What is behind the change from the previous version?
- CA 22: If inter-TAD connectivity is crucial in reconstructing the properties correctly, then isn't it necessary to consider also the connections of the presented three TADs with other TADs around?
- CA 23: Clearly the quantities such as the compaction ratio or the MRG (that the authors present) could be calculated from the 3D structures too, therefore a comparison is possible. I understand that the manuscript is already long, but it seems to me unsatisfying extracting e.g. MRG data with this involved procedure and then not comparing it with other methods to show how much better/worse this is. My concerns about the number of binding molecules are stated above.

One more question: Is the comparison of simulation and theory of the EP also good for monomers that are not in the center of the TAD, but let's say on the boundary of a TAD? These should be included in Fig. 2 as well.

I have one more comment on the Answer 10 to the question of the Reviewer 2: I believe the Reviewer 2 meant that the connector could detach and subsequently reconnect different sites. Such a dynamics certainly affects the steady state end configurations. The authors assume only permanently fixed connectors and the reviewer was asking for a justification of such a model.

Thank you for improving the minor issues. One more: pg. 2: "The maximal possible number N_L of NN connected pairs..." \rightarrow "The maximal possible number N_L of non-NN connected pairs...". This should be also fixed later below.

To summarize: I think a substantial improvement of the paper has been done and this paper should be suitable for a wider audience. Nevertheless based on the novelty criteria (relevance and comparison with older models, paradigm shift) and the remaining issues above I would not recommend it for publication in Nature Communications until all this is satisfactorily solved.

Reviewer #2 (Remarks to the Author):

Review of Shukron and Holcman

While the authors have responded to many of the reviewer's comments, I do not find compelling a model with 3 linear chains with linear crosslinks randomly distributed and floating in space to be a reasonable model for chromosome TADs.

Specific comments to Rebuttal:

Bottom p. 1 Linear springs- this would be ok if the distances never stretch beyond the FENE or WLC cutoff for chromosomes. This is not known.

Random connectors: is this an insertion of random crosslinks? are they permanent or transient? if permanent, that is very unlikely to be biological.

Answer 3, p. 4: as far as I can tell, these are permanent crosslinks. That turns Rouse chain melts into a gel. If the cross-links are transient, that would be more interesting and physiologically relevant.

Answer 5: This is an instance where the model does not capture the biology. Why are the ends cross-linked to inner monomers? Chromosome ends (telomeres) are often tethered to the confined nuclear envelop.

Answer 6: If I understand, they are putting permanent crosslinks between 3 chains and acting like that is confinement? This does not make sense to me. They are studying a super dilute, 3 Rouse chain gel. This is not anything like the highly crowded environment of chromosomes and TADs.

I disagree with the comment about boundaries bottom p5. The cited papers do not allow beads to escape the nuclear wall, they are hypothetical only by not adding active forcing from the wall, which could be. It is true that a boundary is hypothetical, but it seems to capture the critical biology.

Answer 9 (bottom p. 6) The authors restate in different language that they have a 3 chain w/ random permanent crosslinks model. This is not a physiological a model of TADs and encounter probabilities. They ignore all the other chromosomes in this crowded confined environment.

Answer 10. The reviewer understands that there will be different realization. However 3 chain permanently crosslinked gels with different realization of the crosslinks is not transient binding. The steady state of one does not relate to the other.

The statement (bottom p. 7) "Thus, the dynamics of springs in live cells has no role in the computation of steady-state properties, because no temporal dynamics is involved" is not correct. I am aware of models with transient cross-linking springs that bring temporal dynamics into play.

Reviewer #3 (Remarks to the Author):

The authors submitted an improved version of their paper: I am very positive in regards to the extra amount of work done. The authors provide additional results confirming the robustness of the model. I suggest that the paper would be accepted as it is since the analytical results obtained are original and very insightful for the field, and somehow fit the data analysed; the paper provide a detailed description of the methodology used to obtain these results. For this reason, I think that

the work will have a significant impact on the part of HiC community that work on chromatin modelling with both simulations and pen-and-paper.

To recapitulate, the authors address my first two comments in a satisfying manner, showing how the results does not depend on biological noise, by comparing two biological replicates. They also show that the results are consistent in respect to the change of definition of TADs.

I accept the reply to comment 3 and 4 as generally satisfying. And I am very positive about the excluded volume analysis performed to answer question 6 that shows that steric effects may not be important in physiological conditions.

In regard to question 5, I find the added Figure 5 a bit awkward: while it shows that the presence of cross-linkers can reflect heterogeneities of the "measured" diffusion exponent at a certain time scale, I think it is contradicting with the results of equation 14 to use such an observable in the first place. In their approximations the corrections from the Rouse behaviour are additive and this does not translate in any well defined exponent. I appreciate the effort to provide additional ways to test their results but I prefer that the authors would be driven more directly by the predictions of their models instead of by methodologies used in antecedent experimental results.

Regarding comment 7, I have to admit that, the HiC data I've been working on, looks very different from the data analysed in this article. In micro-organisms the hierarchy of interactions and timescales depends strictly on the polymer natural ordering, and I have never seen stepwise features in the contact probability that looks suspiciously like different baselines of the signal, a part, obviously, for the borders defined by whole chromosomes or in case of chromosomal rearrangements. I have to say that the presence of marked TAD borders is a known difference between micro-organisms and multicellular organisms that makes the latter at the same time more complex and interesting. As such, it is fine for me, this is food for thought. I am convinced enough that the model presented in this paper describes well this kind of data.

Response to reviewers

Reviewer #1

I have read the replies of the authors and the new version of the manuscript. I greatly appreciate the amount of work and effort put in to improve the work. Some of the major issues were solved to my satisfaction. This includes Comments/Answers (CA) number 7, 9, 13, 14, 18, 19 and 21. Particularly, I appreciate, the authors clarified that the simulations did not use the mean-field (MF) approximation, but were run with random realizations of the connector placements. To my opinion the work presents two classes of results. First, it is the extension of the RCL model and the mean-field analytical results that agree with the numerical simulations of the specific polymer model. I found this result very nice, definitely worth publication, but probably will be appreciated by specialist (polymer/physics)

readership only. The second class of results relates to the application of the model to the biological data, to reveal the statistics of chromatin organization. At first I state these with a brief description why I do not find them completely convincing. More details are then provided below in my comments on the answers provided by the authors

Comment 1:

Prediction of MRG - the numerical values obtained from the model seem now to be closer to the experiment. What is different here from the previous version of the manuscript? The difference is probably related to the (fitted) value of b , which is also somewhat poorly justified.

Answer 1:

1-In the last version of the manuscript, we replaced the values previously used in Fig. 2, i.e., $b=\sqrt{3\mu\text{m}}$, $D=1\mu\text{m}^2/\text{s}$ by the values $b=0.2\mu\text{m}$, $D=0.008\mu\text{m}^2/\text{s}$, which resulted in the new MRG values, due to the decrease in the value of b ;

2-The second comment about the value and the meaning for b is indeed important for the following reason: The classical Rouse chain consists of monomers connected by springs and the parameter b refers to the mean-square-distance between two consecutive monomers (along the linear backbone). However, after cross-links are added, the mean-distance between two consecutive monomers shrinks (proportional to the number of added connectors N_c (computed for a single TAD, Shukron et al., PRE, 2017). In that context, the interpretation of the value of b is different from the case of the Rouse polymer. However, we kept for b the value of Rouse polymer (Amitai et al, Cell report 2017), because we see chromatin as a Rouse polymer where connectors have been added.

To clarify this point we have now added in the Methods section 1, after Eq. 7:

-" the mean distance b between neighboring monomers is defined \red{for a linear chain when there are no added connectors ($N_c=0$),..."

And at the end of the paragraph:

-"In addition, after connectors have been added the mean distance between two consecutive monomers can be smaller than the initial mean distance b for a Rouse chain ($N_c=0$)".

We have also added a reference to Amitai et al. PRL 2013, in Figure 2, caption, and in Result section p.4, left column, one before last paragraph.

Comment 2:

Compression ratios - as the authors state a way to partly avoid the difficulties of the MRG comparison is to state only relative compressions with respect to linear polymer. Truly, these are giving some qualitative predictions, as shown on the analysis of the experiment replicas (which I appreciate). However the quantitative prediction is a bit problematic as shown on comparison of HiC and 5C data.

Answer 2: We did not expect a perfect statistical reconstruction when comparing HiC and 5C data, because they are produced with a different kb resolution. Here, we wanted to demonstrate that the present polymer method can be applied to each type of data separately, so precisely to compare the polymer reconstruction for each. The present result (Fig. 4) suggests that there are inherent differences between the two types of data.

We now added in p.8, first paragraph, the following sentence:

“Note that the polymer model reconstructed from HiC and 5C data, are not necessary identical, although they share some similar statistics, because for each one, the data are generated at a different resolution”.

Comment 3:

Dynamic data and exponents - I found the presentation unclear and hard to assess its validity

Answer3: the reviewer is referring to the subsection C, p.8: “*Distribution of anomalous exponents for single monomer trajectories*”. We agree with the reviewer comment about clarity and we have now rewritten this section:

“Multiple interacting TADs in a cross-linked chromatin environment, mediated by cohesin molecules can affect the dynamics of single loci trajectories. Indeed, analysis of single particle trajectories (SPTs) \cite{Javer2013,Javer2014, Kepten2013,Amitai2017,Hauer2017} of a tagged locus revealed a deviation from classical diffusion as measured by the anomalous exponent. We recall briefly that the MSD (Eq. \ref{eq:MSDmultipleTADs}) is computed from the positions $r_i(t)$ of all monomers $\$i=1,\dots,N_T\$$. In that case, the MSD, which is an average over realization, behaves for small time t , as a power law

$$\langle (r_i(s+t) - r_i(s))^2 \rangle \sim t^{\alpha_i}.$$

It is still unclear how the value of the anomalous exponent α_i relates to the local chromatin environment, although it reflects some of its statistical properties, such as the local cross-link interaction between loci \cite{Amitai2013,Amitai2017}. Thus we decided to explore here how the distribution of cross-links extracted from EP of the HiC data could influence the anomalous exponents. For that purpose, we simulate a heterogeneous RCL model, where the number of cross-link was previously calibrated to the data. The number and position of the connectors remain fixed throughout all the simulations (for tens of seconds).

We started with a heterogeneous RCL model with three TADs, reflecting the inter and intra-TAD connectivity as shown in Fig. 3. We generated a hundred chromatin realizations G_1, \dots, G_{100} . In each realization G_k , the position of added connectors is not changing. We then simulated in time each configuration a hundred times until relaxation time (SI Eq. \ref{SIeq:relaxationTimeMultiTADdef}). After relaxation time is reached defined as $t=0$, we followed the position of each monomer and computed the MSD up to time $t=25s$. To compute the anomalous exponent α_i , we fitted the MSD curves using a power law Eq. \ref{eq:MSDmodelGeneral} to estimate the anomalous exponents α_i , $i=1, \dots, 302$ along the polymer chain. We repeated the procedure for each stage of cell differentiation: mESC, NPC, and MEF.”

We note that this procedure of extracting the anomalous exponents is classically used on SPT data (see Amitai et al, Cell report 2017, Plos CB 2015, Hauer..Gasser, Nat Structr Biology 2017,..). We have used exactly this procedure on synthetic simulations, calibrated from the data, which is the new added value of the present method. Indeed, in Figure 5, we have related HiC/5C with the statistics of SPTs. We now added to the Discussion section, p.11, the following sentence:

“Finally, we reconstructed a polymer model from HiC and with the present method, we generated numerical simulations and estimated the MSD and the anomalous exponents, which allows us relating HiC with SPT statistics.”

Comment 4:

Number of connectors - to my understanding this has been stressed as the important result in several answers. My main problem with it is the following: Are these connectors real molecules or they are just an abstract way to model organization statistics?.

(i) If they are real (which was suggested by saying they could be cohesin or CTCF) then why the size b does not match the size used for the steric repulsion interaction (40nm) and why does their number depend on the scale? If they are not real, why should we be interested in their number?

(ii) I do not find the assessment of the error bars of the number of connectors fair. For example, I found the following statement misleading: “We found that the number of added connectors in replica 1 and 2 are in agreement.... most significant difference was 5 connectors...”. The 5 connectors is about 30% difference and in the other cases in Fig S4 where the difference is only one or two connectors - the relative error is in many cases also around 30% and in a few cases almost 50% as the number changes e.g from 4 to 7 connectors. Similarly when Hi-C and 5C or when 3 and 6 TADs are compared. These relative differences are about the same as those found between different differentiation stages. How can be the authors then sure that we see for the differentiation stages is a real effect and not just the lack of precision of the method?

Answer 4: Because the heterogeneous RCL polymer model is a coarse-grained model, quantities such as the number of monomers, N , and number N_c of connectors represent approximated averages. But thus this is true for polymer models.

Specifically, the average number N_c of connectors does not refer here to an absolute number of molecules, but a relative number that depends on the HiC resolution. Converting N_c to an endogenous concentration is indeed an important question, that we do not address here, however we will give below a general explanation about the interpretation of this number.

We will discuss below (Answer 9) how connectors are related to the HiC resolution (6 vs. 9kpbs). This number N_c represents the number of connectors at a given scale, which suggests that in the limit of 1bp resolution N_c would represent indeed the endogenous number of linkers, but as the resolution decreases, the number of connectors that are reported by the HiC data also decreases, because two connectors in the same bin are not anymore accounted for. To clarify this key issue, we have now added at the end of the main text of the manuscript a new schematic Figure 6 that precisely summarize this coarse-graining procedure:

Figure 6. Illustration of polymer reconstruction of chromatin at different kb-scales. A. Two examples of polymer reconstruction at 2 (top) and 10 kb (bottom) resolution, where the number of connectors N_c vary with the scale $N_c=11$ (2kb) and $N_c=5$ (10kb). For coarse scale, connectors within the same bin (orange boxes) are discarded. B. Effect of coarse-graining on the encounter probability. It is possible to coarse-grain (yellow arrow), while the construction of a refined polymer model from a coarse grained resolution (red arrow) is an ill-posed problem.

We have also added to the Discussion section, one before last paragraph, the following sentences:

“The interpretation of the number of connectors N_c is not straightforward. This number characterizes the amount of connectors at a given scale, which suggests that in the limit of 1bp resolution N_c , it would be equal to the endogenous number of linkers, but as the resolution decreases, the number of connectors that are reported by the HiC data should also decrease, because two connectors in the same bin are not counted (Fig. 6A). It is always possible to coarse-grain a polymer model, but reconstructing a refined polymer from a coarse grained kb-scale is an ill-posed problem, because the refined encounter probability at high kb-resolution cannot be inferred from a low resolution (yellow and red arrows in Fig. 6B).

(i) In Answer 2 we provided some explanations for the meaning and justification for the value of b . Indeed, for the RCL chain ($N_c > 0$), b does not necessarily refer to the actual mean square distance between connected monomers, but the value of b is used to initialize the polymer and it reflects the standard deviation of a linear chain ($N_c = 0$). The mean distance between monomers for $N_c > 0$ can be computed using equations 11-13 for consecutive monomers m and $m+1$;

(ii) To our knowledge, we presented for the first time such a method. We reported the error bars. We have now changed the mentioned sentence and simply report the result and the 30% variability, which can result from fluctuations in the data across different experiments and scales. We have now changed the sentence in the second paragraph, p.7, second column, to:

“We found that the number of added connectors in replica 1 and 2 differ by at most five connectors for TAD F. This difference between replica may arise from intrinsic fluctuations in the statistics of encounter frequencies.”

Comment 5:

In the following I comment on the answers that I found unsatisfying or only partially satisfying and explain my reasons

CA 2 and 4: I believe the authors misunderstood my comment. I was not saying polymer models are not useful. They are. I was trying to explain that this work does not convincingly show, how novel or different is the present model from its predecessors e.g. what advantage it gives over the old ones. The works of Barbieri et al or Jost et al mentioned or Fudenberg et al, describe different physics (paradigm shift) that can be behind the organization (diffusing binders, epigenomic interactions, active maintenance). In this, although very interesting work I miss such a shift. The distinction from the Random Loop Model (RLM) was mentioned in the answer - static crosslinks here vs. dynamic ones in RLM, but I haven't seen this in the manuscript and an assessment of what advantage the permanent crosslinks have. I understand and agree with the authors that they want to present their results for a mixed audience

Answer 5: There are several novelties of our approach and major differences with existing models that we did not emphasize enough:

1-Our construction of polymer model from HiC is parsimonious in order to be as close as possible from the biological processes where linker are indeed endogeneous binders: the mode uses a minimal number of added connectors at a given scale to match the experimental steady-state of HiC/5C data. This property is key compared to other model, where this number is not necessarily related to HiC or concentration. We are trying to reproduce contact maps assuming that they are generated by linkers such as cohesin, and we do not want to hide in the complexity of the polymer model any other parameters. For example, the model used in L. Giorgetti et al, Cell 2014 does not include physical contact because it is based on a fully monomer-monomer interaction, described by pair potential wells. Here we exactly use an opposite approach. In addition, contrary to Barbieri et al., we do not assume the presence of several types of diffusing molecules and their hypothetical binding points along the chromatin, and we think this is not necessary.

2-Contrary to the Random Loop Model (RLM), we do not consider here transient binding, because we showed that the position of random connectors within TADs does not matter (as long as they are randomly placed uniformly). Indeed, by placing connectors randomly, we captured the heterogeneity in chromatin organization across cell population, as verified by experimental observations in the 5C and HiC data. One other difference between our RCL model and the RLM model, is the possibility to account for several interacting TADs and obtain specific statistical laws (Radius of Gyration, encounter probability, MSD,...), that was never done before. This approach is key to recover the mean number of connectors needed to construct the polymer model, the contact map that best approximate the HiC one (as shown in Fig. 3). Note that we can do this reconstruction by simulations and using exact formula.

3-Using our RCL model, we study the chromatin structural reorganization throughout stages of differentiation. To our knowledge this is the first time that such an approach is used to study chromatin reorganization for snapshots of 5C.

4-Using the present approach, we linked the statistical properties associated to dynamics of

single loci to the HiC/5C, which consist of ensemble organization of the chromatin. Indeed, by placing the minimum number of connectors at a given bp resolution (scale), we can capture the constraints revealed by HiC. We recall that connectors are randomly positioned inside a TAD and this accounts for the inherent cell-to-cell variability.

5-Finally, for each configuration, we simulated each polymer and computed the anomalous exponents for all monomers: we found a spectrum of possible exponents, for the same loci, because its local environment is different. We concluded in the manuscript that the anomalous exponents are modulated by the number and the positions of linkers. Conversely, we are also suggesting that measuring the anomalous exponents of loci, positioned at different locations inside a TAD, could reveal the amount of connectors and thus chromatin condensation beyond the exact position of these loci. We think that is an important and new concept.

To our knowledge, no other models have presented the advantages outlined above, and connected so closely to experimental data, based on first principles of polymer physics. The method of reconstruct is generic and can be applied to any 5C/HiC data.

Finally, the present approach can also be used to study the phase-liquid transition at the chromatin level, depending on the number of connectors. We computed here the Radius of Gyration that clarifies how chromatin compaction depends on the number of added connectors. Similar questions were recently discussed in a general perspective (D. Hnisz, Phase Separation Model for Transcriptional Control, Cell 2017) and we presented here some answers.

Note that the article of Fudenberg et al. proposes a mechanism of loop formation, but this idea has not yet been supported by a precise physical mechanism: a polymer model with a prescribed number of loops is yet to be made.

The approach of Jost et al. concerns polymer model, that exhibit clustering based on epigenetic markers, which is quite different from our approach.

We have now added in the first paragraph of the Discussion the description of previous polymer models, as suggested by the reviewer:

“The present polymer model differs from previous ones by several aspects: Our construction of polymer model from HiC is parsimonious: it uses a minimal number of added connectors at a given scale to match the experimental steady-state of HiC/5C data, in contrast to the model [8], based on a fully monomer-monomer interaction, described by potential wells. Furthermore,

we do not have here several types of diffusing binders that need to finding a binding site to generate a stable link, as introduced in [40]. In addition, contrary to the Random Loop Model (RLM) [16], we do not consider here transient binding, because we showed that the position of random connectors within TADs does not matter (as long as they are uniformly randomly placed). By placing connectors randomly, we capture the heterogeneity in chromatin organization across cell population. Here we fix connectors which are stable in the time scale of minutes to hours. The present polymer construction is motivated by the evidence of many stable loci-loci interactions, which are common to the majority of chromosomes in 5C (peaks of the 5C data) [1,33]. These stable interactions are also

found at TAD boundaries, which are conserved in both human and mouse. There are several conflicting studies [5,33,41] about the binding time of connectors (CTCF-cohesin, etc...), which suggest that cross-links can remain stable for minutes to hours and even during the entire phase cycle. Here, we study the chromatin dynamics within this time range where cross-links are stable [cite{Hansen2017,Rao2014}]. One final difference between the present RCL model and the RLM model, is the possibility to account for several interacting TADs and our new expressions for some statistical quantities such as Radius of Gyration, encounter probability, MSD,....

Comment 6:

CA 3: By “3D structure” I meant statistics of the ensemble of 3D structures, such as MRG of different segments. Therefore I don’t understand why the authors claim that there is no ground for comparison with other methods or experiments. For example the work Di Pierro et al (PNAS 2017) also presents a method to obtain spatial distances from ChIP-Seq and Hi-C and tested their method with fluorescence experiment. Another example is Stevens et al (NAT 2017) who are also extracting RG from single cell Hi-C and compared with fluorescence experiments. Yet another is Li et al (TCBB 2018). A comparison with experimental fluorescence data of Nora et al who have measured the spatial distances for the same chromosome segment as in the present manuscript is missing. I stated my objections on the number of connectors above. Yes, I appreciate the investigation of the structural changes during differentiation, if the model is sufficiently justified.

Answer 6: The article of Di Pierro et al, PNAS 2017 presents spatial distances between genomic loci, measured by FISH in autosomal chromosomes of human B-lymphoblastoid cells, that are compared to chromatin reconstruction of the same genomic section, whereas in the present work we reconstruct TADs from X chromosomes of mouse embryonic stem cells, and therefore we cannot directly compare with Di Pierro. However, to address the question of the reviewer, we now compare the distribution of spatial distances from DNA FISH data reported in Giorgetti et al., Cell, 2014 to the predictions of the heterogeneous RCL model. We note, however, that the current work is not meant to reproduce the work in Giorgetti et al. 2014.

First, we have now added in the Supplementary Information, a new subsection F, with details of the derivation of the distribution of distances between monomers of the heterogeneous RCL polymer. The new subsection II F is described below:

“We now derive an expression for the distribution of the distance between any two monomers r_m and r_n of the heterogeneous RCL polymer model. Since the RCL polymer belongs to the Gaussian chain family [12], and therefore the vector $r_m - r_n$ is Normally distributed in any dimension $j=1, \dots, d$ with mean $\mu_{mn}=0$ and standard-deviation $\sigma_{mn}(\Xi)$ (square root of Eqs. 41,51), where Ξ is the connectivity fraction matrix (Eq. 4, main text). The distance D_{mn} between monomers r_m and r_n is defined as

$$D_{mn} = ||r_m - r_n|| = \sqrt{(\sum_{j=1}^d (r_m^{(j)} - r_n^{(j)})^2)},$$

and is a χ -distributed random variable with d degrees of freedom, as the norm of d -

dimensional Normally distributed vector $r_m - r_n$. The normalized norm to the variance

$$Z_{mn} = \left\| (r_m - r_n - \mu_{mn}) / \sigma_{mn}(\Xi) \right\| = D_{mn} / \sigma_{mn}(\Xi),$$

is distributed according to standard χ distribution, as the norm of d -dimensional Normally distributed random vector with mean zero and standard-deviation one. Thus, we compute the distribution $f_{D_{mn}}(x)$ of the distances D_{mn} using the standard χ distribution

$$f_{D_{mn}}(x) = f_{Z_{mn}}(x / \sigma_{mn}(\Xi)) = 2^{1-d/2} \Gamma(d/2) (x / \sigma_{mn}(\Xi))^{d-1} \exp(-(x / \sqrt{2} \sigma_{mn}(\Xi))^2), \quad (58)$$

where Γ is the Gamma function.

Finally, using the average of the standard χ distribution $\langle Z_{mn} \rangle$, the average spatial distance $\langle D_{mn} \rangle$ between monomers r_m and r_n is

$$\langle D_{mn} \rangle = \langle \sigma_{mn}(\Xi) Z_{mn} \rangle = \sigma_{mn}(\Xi) \langle Z_{mn} \rangle = (\sqrt{2}) \sigma_{mn}(\Xi) \Gamma((d+1)/2) / \Gamma(d/2)."$$

We also added in the Supplementary Information the new Section III, presenting the comparison of the distribution of distances derived above, with seven DNA FISH probes from Giorgetti et al, Cell, 2014. The new Section III: *Comparison of the distribution of distances of the heterogeneous RCL model with DNA FISH data*, is:

“We compare the prediction of the distribution of three-dimensional distances between monomers of the heterogeneous RCL polymer to measurements of DNA FISH probe pairs. For this comparison, we use seven DNA FISH probe pairs of lengths 4-16kb, reported in [18]. We first binned the 5C data [3] of TAD D, E, and F, at a monomer resolution of 3kb avoiding the two ends of a probe within the same monomer. We obtained a coarse-grained genomic section of $N=605$ monomers, with $N_D=123$, $N_E=174$, and $N_F=308$. We then mapped the bp positions of probes to monomers (see SI of [18]). In Table I, we summarize the details of mapping FISH probes at 3kb to monomers.

Probe	Length (bp)	Start monomer (r_m)	End monomer (r_n)
pEN1	9839	43	46
pEN2	9612	109	112
pLG1	9430	54	57
pLG10	4503	116	117
pLG11	4938	96	98
X3	16060	73	78
X4	12606	87	92

Table I: Mapping seven DNA FISH probe ends [18] at 3kb resolution from bp to monomers of the coarse-grained, three TAD, RCL polymer model with $N=605$ monomers

We construct a three TAD RCL polymer with $N=605$ monomers. To obtain the average connectivity

fraction matrix Ξ , we fit the EP of the RCL polymer (Eqs. 42,52) to the empirical 5C EP data of TAD D, E, and F (see Methods section, main text) and obtained:

$$\Xi = 10^{-3} [\begin{matrix} 3.5723, 1.7856, 1.5828 \\ 1.7856, 3.6240, 1.7966 \\ 1.5828, 1.7966, 3.2404 \end{matrix}] \quad (60)$$

To compute the distribution of the three-dimensional distances $f_{Dmn}(x)$ (Eq. 58), we first compute the variance $\sigma^2_{mn}(\Xi)$ (Eqs. 41-51) of the distance between probes (monomers m, n Table I), where we substitute the fitted Ξ (Eq. 60) in Eq. 58, with $b=0.2 \mu\text{m}$. In Fig. 2, we plot the distribution of 3D DNA FISH probe distances (black) vs. predicted $f_{Dmn}(x)$ (red), and plotted mean DNA FISH (green circles) and predicted distances $\langle D_{mn} \rangle$ (Eq. 59) of the RCL model, which are in good agreement.

”

The new Fig. S2 shows the very nice agreement, reproduced below:

Figure S2 **Comparing 3D DNA FISH with RCL model prediction.** Distribution of three-dimensional distances between seven DNA FISH probe pairs (black) of lengths 4.5kb-16kb (measured in [18]) and predicted distributions f_{Dmn} (Eq. 60) of the RCL polymer model at 3kb resolution. Average values for DNA FISH (green circle) and predicted (Eq. 61, red squares).

In addition, we have now added in the Discussion section, in the third paragraph, the sentence:

“In addition, using Eq. 15, we were able to reproduce the distributions of three-dimensional distances between seven genomic loci, measured by DNA FISH probes (SI Fig. 2).”

We added in the Result section, subsection B, second paragraph, the last sentence:

“In addition, using the calibrated RCL model at 3kb, we were able to reproduce the distributions of three-dimensional distances between seven DNA FISH probes reported in [8] (SI Fig. 2).”

Finally, we added in the Result section, the new Eq. 15:

“The distribution $f_{D_{mn}}(x)$ of the distance $D_{mn} = ||r_m - r_n||$ between any two monomers r_m and r_n (see SI subsection IIF) is given by

$$f_{D_{mn}}(x) = 2^{1-d/2} \Gamma(d/2) (x / \sigma_{mn}(\Xi))^{d-1} \exp(-(x / \sqrt{2} \sigma_{mn}(\Xi))^2), \quad (15)$$

where Γ is the Gamma function.”

Comment 7:

CA 5: I appreciate the simulation with exclusion volume interaction although one can of course ask what is the range of c for which it is negligible. More importantly, there was no comment on the topological constraints

Answer 7: To answer the reviewer question about commenting topological constraints, we just mention that we provided an upper estimation of the exclusion radius. Indeed, we found using simulations that for an exclusion radius $c > 0.667 \mu\text{m}$ (Fig. S4), the present simulations cannot account for exclusion volume. In addition, we do not account for any real topological constraint at this resolution. We have now added a sentence about this in the method section .

In Supplementary Information Section III, after Eq. 58, we have added

“In the present model, we do not account for any topological constraint on the chromatin at any resolutions”.

Comment 8:

CA 6: Thank you for the explanations. I don't agree that there is no long-range interaction. With high b the interaction is quite long-range (e.g. for the HiC data $b = 1.8 \mu\text{m}$), but specific to certain monomers. If the connectors represent the cohesins, why the b does not match the c value that characterizes the size of the molecule?

Answer 8: The value of the exclusion radius c is determined by structural proteins (about $0.40 \mu\text{m}$ radius) at our scale of simulations, whereas the value of the mean monomer distance b , as explained in Answer 2, represents the mean-square distance between connected monomers on the linear chain (i.e., zero added cross-links, $Nc=0$).

In principle the value of the parameters $c=0.04\mu\text{m}$ and $b=0.2\mu\text{m}$ are not linked. The value of b , although uniform throughout the linear chain, is not the mean-square distance between monomers when connectors are added within and between TADs. These additional connectors renders the computation of the mean square distance for each TAD difficult.

However, for adjacent monomers m and $m+1$, of a homogeneous RCL (one TAD, $N_c>0$), the mean square distance b^* was computed in Shukron PRE 2017 (Eq. 30 therein), given (for non-vanishing connectivity fraction ξ) by:

$$(b^*)^2 \sim \sigma^2(m, m+1) = b^2 / \sqrt{N\xi} [1 - \exp(-\sqrt{N\xi})],$$

When $\sqrt{N\xi} \ll 1$ (almost no connectors), we converge back to the initial value b of the Rouse polymer:

$$(b^*)^2 = \sigma^2(m, m+1) \sim b^2 / \sqrt{N\xi} [1 - (1 - \sqrt{N\xi})] = b^2.$$

In general, for a heterogeneous RCL model, using Eqs. 11-13 for monomers m and $m+1$, we can calculate the value of b^* , which is clearly smaller than b . For example, using parameters $N_D=62$, $N_E=88$, $N_F=162$, $b=0.2\mu\text{m}$, and connectivity matrix Ξ as in Fig. 3, we obtain from Eq. 11, that the average (over TADs) mean distance is $b^* = 0.16\mu\text{m}$.

To clarify the present question, we have now added in the paragraph before the Results section the following sentence:

“In addition, after connectors have been added, the mean distance between two consecutive monomers r_m, r_{m+1} can be smaller than the initial mean distance b for a Rouse chain ($N_c=0$), as computed below, based on Eqs. 11-13.”

We also edited the sentence in the Supplementary Information, subsection A, below Eq. 12:

“... b is the standard deviation of connected monomers for $N_c=0$ (Rouse chain),...”

Comment 9:

CA 8: I found the fitting procedure still unclear. b is tuned to maintain MSR G the same for all scales. As D is fixed, the value of b affects the value of k and the number of connectors is different for different scales. This suggests then that these connectors are not real or there is some renormalization scheme under the hood which should be clarified

Answer 9: As we mentioned in Answers 2 and 8, the parameter b refers to the mean square distance for the Rouse Chain ($N_c=0$). In addition, it does not enter into the computation of the encounter probability (EP, Eqs. 10,12), and hence we do not use b in the fitting procedure between empirical and theoretical EP to find the mean number of connectors N_c .

The number of connectors N_c is defined for a given kb-resolution. For example, if N_c is known at 6kb, then at 12kb, we expect to find less connectors, because some will disappear in the process of coarse-graining (smoothing of the EP). However, going in the opposite direction would be more interesting: suppose we know the number of connectors at resolution 12kb, can we infer the number of connector at the 1bp resolution? This would mean that we could predict the total amount of linkers, which would reveal the exact concentration of linkers. However, inferring the EP at high resolution

from a smoothed EP at a coarse resolution is an ill-posed problem (see new fig. 6, shown above).

The number of connectors at a given scale follows the abstraction of HiC data at a same scale: it represents chromatin interaction at this scale. Is it the real chromatin, maybe not, but it is considered as a good approximation, that should be refined as the resolution decreases.

To address this question 9 and 4, we have added Figure 6 and the associated discussion.

Comment 10:

CA 10, 11 and 15: I found the presentation of the MSD data and procedures unsatisfying. Firstly, if exponents are extracted, the plots of MSD vs time should be in log-log to see the robustness and range of the slope. Second, the equation (14) holds in intermediate times and represents the change of the exponent of 0.5 (second term) at intermediate times to the exponent 1 (first term) at late times. This means also that the prefactors affect the effective slope measured (so the chosen values of b or D are important). Third, I don't understand why on one hand the structure is represented by the steady states, but the MSD is evaluated in intermediate times and why these times are those below 25s. Fourth, is there a justification for the value of the diffusion coefficient $D = 0.008\mu\text{m}^2/\text{s}$? In answer 11 it is stated that the change of the parameters (D and b) has little effect on the figure, because the steady states don't depend on time scales. The intermediate MSD scaling however does depend on the time scales, so this match should be explained.

Answer 10:

1- In the Log-log, we generally miss refined behavior. We always used the linear axis for estimation, as we did in many of our previous works Amitai et al, Cell report 2017, Shukron PLOS 2017 and PRE. 2017), we do not think that this makes a real difference.

2-the parameters D and b do not affect the steady-state, but we agree, they are determinant for the transient characteristics.

3- We reconstructed the polymer model from the steady-state 5C/HiC encounter matrices. Once the polymer model has been calibrated to have the same statistics as the HiC, we can use it to study transient properties, revealed by the MSD. To avoid that cross-links may have been removed transiently (time scale of minutes to hours, [Hansen et al., eLife,2017]), we decided to evaluate the MSD up to 25s.

4-We use the diffusion coefficient $D=0.008\mu\text{m}^2/\text{s}$ from [A. Amitai et al, PRL, 2013]. We note that the values of D and b do not affect the steady-state encounter probability, as shown in Fig.2 A-B. However, the MSD is computed along trajectories, and it not a steady-state property, and indeed as reviewer mentioned, it depends on D and b .

Comment 11:

CA 12: Why c is different from b ? The authors have the simulations with other parameter sets so can assess and present the sensitivity of the model to these choices.

Answer 11:

In Answers 2 and 8, the value of b represents the mean square distance between connected monomers of the linear Rouse chain, and its interpretation in the context of multiple interacting TADs ($N_c > 0$) is not immediate. The non-homogeneous connectivity in multiple-TAD polymers, resulted in different values for mean-square distance between monomers of different TADs. Furthermore, the mean square distance depends on both inter and intra-TAD connectivity.

The value of the parameter b depends on internal organization of the polymer, and is a property of the polymer structure. Note that the value of the exclusion radius c is based on the characteristics of external forces, such as those generated by crowding and activity of structural proteins (e.g. CTCF, cohesin) at a given coarse-grained scale. In Answer 8, computed the mean square distance based on the input $b=0.2$ for a heterogeneous RCL model and we found $b^* = 0.16\mu\text{m}$.

To clarify the difference between the parameters c and b , we have now added in Supplementary Information Section III, below Eq. 58, the sentence:

“The exclusion radius c characterizes physical properties of external forces applied on the chromatin at a given coarse-grained scale. This parameter is thus independent of the mean distance b between connected monomers at that scale (see Method section, main text).”

Comment 12:

CA 17 and 20: I apologize that I missed the MSR comparison between theory and simulations in the first version of the paper. Now the values 0.21, 0.19 (note there is a typo in the manuscript stating 0.9) and 0.2 seem to be close to the FISH experimental values mentioned before (Nora et al). What is behind the change from the previous version?

Answer 12: Thank you for spotting the typo; **we have now corrected it**. In the current version of the manuscript, we use the parameters $D=0.008\mu\text{m}^2/\text{s}$ and $b=0.2\mu\text{m}$, which resulted in MRG of $0.21\mu\text{m}$ and $0.19\mu\text{m}$ (see also Answer 1).

Comment 13:

CA 22: If inter-TAD connectivity is crucial in reconstructing the properties correctly, then isn't it necessary to consider also the connections of the presented three TADs with other TADs around?

Answer 13: We agree with the reviewer that to model the entire chromosome, we need to account for all TADs. However, in the current work, we provided a proof of principle for our modeling approach, using only three TADs, which harbor the X inactivation center, located between TAD D and E. We further included an extra TAD F to show the impact of long-range connectivity between the relatively distal TADs D and F, and to account for TAD E, which is connected on both its ends. We however note that long-range interactions beyond the scale of two TADs (2 Mbps) are rare (Nora et al. 2012, Dixon et al. 2012, Rao 2017).

Comment 14:

CA 23: Clearly the quantities such as the compaction ratio or the MRG (that the authors present) could be calculated from the 3D structures too, therefore a comparison is possible. I understand that the manuscript is already long, but it seems to me unsatisfying extracting e.g. MRG data with this involved procedure and then not comparing it with other methods to show how much better/worse this is. My concerns about the number of binding molecules are stated above.

Answer 14: For a comparison of our model predicted 3D distances with FISH data, please see Answer 6 above, and the new subsections F and SI Fig.2. section III of the Supplementary Information.

Comment 15:

One more question: Is the comparison of simulation and theory of the EP also good for monomers that are not in the center of the TAD, but let's say on the boundary of a TAD? These should be included in Fig. 2 as well.

Answer 15: The theoretical EP (Eqs. 10, 12) matches the simulated EP for monomers on TAD boundaries. To show the agreement between them, we have now added to Fig.2B a bottom panel with a comparison of the theoretical and simulation EP for monomers at boundaries of TADs, monomers: r_1 , r_{51} , and r_{91} . The new Fig.2 is shown below:

*Figure 2: **Statistical properties of the heterogeneous RCL polymer.** A. Encounter frequency matrix of a polymer with three TAD blocks (TAD_1 , TAD_2 , TAD_3) of $N_1=50$, $N_2=40$, $N_3=60$ monomers, computed from 10,000 simulations of the Eq. 7 with $\Delta t = 0.01s$, $D= 8 \times 10^{-3} \mu m^2/s$, $d=3$, $b=0.2 \mu m$. The number of added connectors appears in each block. Three distinct diagonal TADs are visible (red boxes) where secondary structure appears (black lines) due to weak inter-TAD connectivity. B. Encounter probability (EP) of the heterogeneous RCL described in panel A, where the simulated (orange) and theoretical (blue, Eqs. 10, 12) EP agree. In the top panel, we plotted the EP for middle monomers in each TAD: monomer r_{20} (top left), monomer r_{70} (top right), and monomer r_{120} (bottom left). In the bottom panel we plotted the simulation EP (orange) vs. theoretical EP (blue) for monomers r_1 (left), r_{51} (center), and r_{91} (right) located at TAD boundaries. C. Average mean square displacement of monomers in each TAD of the heterogeneous RCL polymer, simulated as described in panel A, simulation (continuous) versus theory (dashed, Eq. 14) are in good agreement for time up to $t=25s$.*

Comment 16:

I have one more comment on the Answer 10 to the question of the Reviewer 2: I believe the Reviewer 2 meant that the connector could detach and subsequently reconnect different sites. Such a dynamics certainly affects the steady state end configurations. The authors assume only permanently fixed connectors and the reviewer was asking for a justification of such a model.

Answer 16: The steady-state EP does not depend on dynamics of connectors. The steady-state configuration of the chromatin is embedded in the 5C data, on which we calibrated the RCL model. Therefore, by definition we capture the steady-state EP, regardless of how cross-links are formed.

Comment 17:

Thank you for improving the minor issues. One more: pg. 2: “The maximal possible number NL of NN connected pairs...” → “The maximal possible number NL of non-NN connected pairs...”. This should be also fixed later below. To summarize: I think a substantial improvement of the paper has been done and this paper should be suitable for a wider audience. Nevertheless based on the novelty criteria (relevance and comparison with older models, paradigm shift) and the remaining issues above I would not recommend it for publication in Nature Communications until all this is satisfactorily solved.

Answer 17: We have now corrected.... NN to non-NN in the Method section above and below Eq. 3.

We thank again the reviewer for helping us increase the quality of the manuscript.

Reviewer #2 (Remarks to the Author):

While the authors have responded to many of the reviewer's comments, I do not find compelling a model with 3 linear chains with linear crosslinks randomly distributed and floating in space to be a reasonable model for chromosome TADs.

Specific comments to Rebuttal:

Comment 1:

Bottom p. 1 Linear springs- this would be ok if the distances never stretch beyond the FENE or WLC cutoff for chromosomes. This is not known.

Answer 1: Extreme stretching of cross-links in the RCL requires a large fluctuations, leading to extremely rare events. Once the polymer simulation has reached relaxation, the statistics collected over realizations do not show any of these rare events. Indeed, in our simulations, all polymers are simulated after relaxation time is reached, so we examined their structure at equilibrium. In addition, our formal computations are in agreement with our simulations, confirming that at steady-state, we are confident that the present approach is sufficient.

Comment 2:

Random connectors: is this an insertion of random crosslinks? are they permanent or transient? if permanent, that is very unlikely to be biological.

Answer 2: By permanent, we refer to the time scale of polymer relaxation. For example, during interphase, chromatin contains many loci-loci interactions, which are common to the majority of chromosomes (peaks of the 5C data, see also Rao et al., Cell, 2017, Nora, Nature, 2012, etc...). These interactions are found at TAD boundaries, which are conserved in both human and mouse. There are several conflicting studies (Andrey 2017 Genome research, in development. Dixon, 2012, CTCF located at stable TAD boundaries) about the time scale of connectors (CTCF-cohesin, etc...). Some suggest that they can stay stable for minutes and others, for hours and even the entire phase cycle (REF). Here, we studied the chromatin dynamics within this time range where cross-links are stable (see e.g. Hansen et al., eLife,2017, Rao et al, Cell, 2017).

We have in the present approach capture the conformation of chromatin that could last minutes to hours. However, since we are averaging over realizations, where connectors have been randomly placed, we have thus accounted for structural heterogeneity at steady-state which could be equivalent (ergodicity) to obtained the statistics over polymer models, where the connectors

could have been placed transiently. Indeed, we recall, that we have shown that the location of the connectors does not matter as long as they are placed in the correct region (see our material and method: for example in a TAD: only the number of connectors does matter), as we reconstructed from 5C/HiC experiments. Our model aims at capturing the steady-state configuration in a multi-TAD organization and study the relationship between organization and dynamics rather than at fast transient dynamics between two monomers.

For that specific fast transient, the dynamics of cross-links could be relevant, as discussed in Bohn and Herrmann PRE, 2007. Although even in that article, the mean field derivation is used, which is equivalent to consider fixed added cross-links. Now the only part of the present article, where we indeed considered a transient regime is to estimate the anomalous exponents. In that case, we specifically mentioned that we did not use a time series longer than 25 seconds, which is much shorter than the binding time of connectors. Thus considering here fixed connectors did not impact the results that we have found.

Comment 3:

Answer 3, p. 4: as far as I can tell, these are permanent crosslinks. That turns Rouse chain melts into a gel. If the cross-links are transient, that would be more interesting and physiologically relevant.

Answer 3: In the previous Answer 2, we explained that at steady-state, having transient or fixed connectors would lead to a similar steady-state statistic. We also mentioned that at the time scale of $t < 25s$, we considered the connectors as fixed. At this time scale, we use our model, where molecules such as CTCF or cohesin are making stable cross-link, with a residence time of 1-22 minutes (Hansen et al., CTCF and cohesin regulate chromatin loop stability with distinct dynamics, 2017, eLife).

There are other conditions where fixed connectors are stable such as during interphase. It is true that during mitosis, the present approach does not work, but this is really a fast changing chromatin organization. However, this is not the situation we are exploring here.

Comment 4:

Answer 5: This is an instance where the model does not capture the biology. Why are the ends cross-linked to inner monomers? Chromosome ends (telomeres) are often tethered to the confined nuclear envelop.

Answer 4: Telomeres can indeed be attached to the nuclear envelop or in some cases (depending on Sir over expression), they can detach from the nuclear membrane in Yeast (Gasser and Taddei's group JCB 2011). In addition, the telomeres seem to be freely moving on the nucleus surface. Modeling the entire chromosome motion remains challenging and yet to be performed. Here we restricted our model to a small chromosomal region, containing three TADs, central to the chromosome and no lamina interactions was observed in the HiC data. Thus we did not account for any external interactions such as with boundary of the nucleus.

In addition, we would like to emphasize that the end monomers do NOT represent telomeres in our model. In principle, the referee is correct that the end monomers should also be connected to other monomers that represents a different genomic region. We did not connect because there would indeed always a last monomer, if we do not go until the telomere. However, we think that our approach still brings very relevant description of the inner monomers, even if the real connections are accounted for the two ends. Indeed, the many added connectors have strong consequences on the indeed monomer statistics, compared to the forces that the two extreme monomers would have on the inner chain. Thus, we think that disregarding the exact forces on these extreme monomers, do not affect much that dynamics and structure of the three TADs, excepted in a very tiny boundary layer of these monomers. For example, we reported the value for the anomalous exponent for all monomers and there is a large deviation for the ends one. In addition, they only represent 2 out of more than 150 to 300 monomers. So their statistics do not contribute much.

Finally, we note that the statistical properties of the end of monomers of TAD E, connected on both ends to the other TADs, are not that different from that of monomers 1 and N, which are connected to other parts of the chromosome, but not to other TADs outside the genomic region we model. To clarify this point, we have now added a bottom panel B in Fig. 2 main text, showing the statistics of end monomers in the new Fig. 2, reproduced below

Figure 2 **Statistical properties of the heterogeneous RCL polymer.** A. Encounter frequency matrix of a polymer with

three TAD blocks (TAD1, TAD2, TAD3) of $N_1=50$, $N_2=40$, $N_3=60$ monomers, computed from 10,000 simulations of the Eq. 7 with $\Delta t = 0.01s$, $D = 8 \times 10^{-3} \mu m^2/s$, $d=3$, $b=0.2 \mu m$. The number of added connectors appears in each block. Three distinct diagonal TADs are visible (red boxes) where secondary structure appears (black lines) due to weak inter-TAD connectivity. B. Encounter probability (EP) of the heterogeneous RCL described in panel A, where the simulated (orange) and theoretical (blue, Eqs. 10, 12) EP agree. In the top panel, we plotted the EP for middle monomers in each TAD: monomer r20 (top left), monomer r70 (top right), and monomer r120 (bottom left). In the bottom panel we plotted the simulation EP (orange) vs. theoretical EP (blue) for monomers r1 (left), r51 (center), and r91 (right) located at TAD boundaries. C. Average mean square displacement of monomers in each TAD of the heterogeneous RCL polymer, simulated as described in panel A, simulation (continuous) versus theory (dashed, Eq. 14) are in good agreement for time up to $t=25s$.

Comment 5:

Answer 6: If I understand, they are putting permanent crosslinks between 3 chains and acting like that is confinement? This does not make sense to me. They are studying a super dilute, 3 Rouse chain gel. This is not anything like the highly crowded environment of chromosomes and TADs.

Answer 5: Confinement refers to the physical situation, where the polymer motion is restricted by crowding. We have not introduced anything like this here. However, it is true that neighboring chromosomes can restrict the motion of chromatin. However, we showed that the effect of connectors significantly reduces the radius of gyration by at least a factor 10. Thus, connectors generate a very condensed polymer state. At the scale of 6-10kb, with a total of 2Mbps, crowding interactions with other chromosomes that are not contained in the HiC data, are disregarded. We have not found any of these long-range interactions in the HiC data sets, thus we did not include any in the modeling part. Moreover, possible forces occurring from the nuclear boundary or from the lamina are also absent at the scale of the three TADs that we are studying (protocol described in Nora et al., Nature, 2012).

To comply with this comment, we have now added in the Discussion section, first paragraph:

“In addition, at the scale of few μm occupied by TAD D, E and F of chromosome X, we neglected crowding effects from neighboring chromosomes”

Comment 6:

I disagree with the comment about boundaries bottom p5. The cited papers do not allow beads to escape the nuclear wall, they are hypothetical only by not adding active forcing from the wall, which could be. It is true that a boundary is hypothetical, but it seems to capture the critical biology.

Answer 6: It is very hard in the absence of clear information to define a boundary. First, there is no information in the HiC/ 5C data on which the model could be calibrated. Second, we did not want to introduce an artificial boundary constraint that could represent the nucleus surface (forces from nuclear wall or lamina). Third, at the modeling scale we use $\sim 1-2Mbp$, we are far from an entire chromosome. Finally, had we introduced a crowded environment, characterizing a boundary for the TADs we model, we would have needed to give a dynamics of interactions with the boundary, which

is unknown.

Comment 7:

Answer 9 (bottom p. 6) The authors restate in different language that they have a 3 chain w/ random permanent crosslinks model. This is not a physiological a model of TADs and encounter probabilities. They ignore all the other chromosomes in this crowded confined environment.

Answer 7: We have answered this question several times above, but we shall add here that to our knowledge there are no canonical physiological models of TADs, thus many groups are trying to propose accurate models such as the group of Heerrmann, Pombo, Barbieri, Rosa, Mirny, Jost, Gioergetti, Grossberg, Thiromalay and many others.

Finding a method to reconstruct a parsimonious polymer model from the EP is indeed the challenge that we have undertaken here, and we think that we have pushed in that direction the farthest we could. This remains challenging and indeed, we hope that the community will be able to build on our model to one day account for the entire chromosome.

Comment 8:

Answer 10. The reviewer understands that there will be different realization. However 3 chain permanently crosslinked gels with different realization of the crosslinks is not transient binding.

The steady state of one does not relate to the other.

Answer 8: We have answered this question above where we explained actually that the time scale for a transient regime is of the order of few minutes, before connectors have time to change and here we studied the transient properties before this time scale is reached. Thus the difference between transient and permanent does not impact our predictions.

Comment 9:

The statement (bottom p. 7) “Thus, the dynamics of springs in live cells has no role in the computation of steady-state properties, because no temporal dynamics is involved”; is not correct. I am aware of models with transient cross-linking springs that bring temporal dynamics into play.

Answer 9:

-we confirm that in our model, since we are taking snapshots of the polymer models at steady-state, the dynamics of spring should not lead to any effect.

-To remove any ambiguity, we **have now removed the quoted sentence.**

We thank the reviewer and appreciate his comments throughout this revision process.

Reviewer #3 (Remarks to the Author):

The authors submitted an improved version of their paper: I am very positive in regards to the extra amount of work done. The authors provide additional results confirming the robustness of the model. I suggest that the paper would be accepted as it is since the analytical results obtained are original and very insightful for the field, and somehow fit the data analyzed; the paper provide a detailed description of the methodology used to obtain these results. For this reason, I think that the work will have a significant impact on the part of HiC community that work on chromatin modelling with both simulations and pen-and-paper.

To recapitulate, the authors address my first two comments in a satisfying manner, showing how the results does not depend on biological noise, by comparing two biological replicates. They also show that the results are consistent in respect to the change of definition of TADs.

I accept the reply to comment 3 and 4 as generally satisfying. And I am very positive about the excluded volume analysis performed to answer question 6 that shows that steric effects may not be important in physiological conditions.

Comment 1:

In regard to question 5, I find the added Figure 5 a bit awkward: while it shows that the presence of cross-linkers can reflect heterogeneities of the "measured" diffusion exponent at a certain time scale, I think it is contradicting with the results of equation 14 to use such an observable in the first place. In their approximations the corrections from the Rouse behaviour are additive and this does not translate in any well-defined exponent. I appreciate the effort to provide additional ways to test their results but I prefer that the authors would be driven more directly by the predictions of their models instead of by methodologies used in antecedent experimental results.

Answer 1: We approximated here in Figure 5, the MSD by t^α . We recall here that we fixed the number of connector and their positions. This is in contrast with the situation under which we derived Eq. 14, where we average over cross-linker positions and monomers. This averaging procedure leads to an exponent of 0.5. However, we reported that the average anomalous exponent is ~ 0.4 , for the case of non averaging, which is very important. This value of 0.4 seems to be consistent with the published experimental literature based on SPT [see Lagomarsino Nat. Com, P. Ciccuta, S. Gasser].

Comment 2:

Regarding comment 7, I have to admit that, the HiC data I've been working on, looks very different from the data analyzed in this article. In micro-organisms the hierarchy of interactions and timescales depends strictly on the polymer natural ordering, and I have never seen

stepwise features in the contact probability that looks suspiciously like different baselines of the signal, a part, obviously, for the borders defined by whole chromosomes or in case of chromosomal rearrangements. I have to say that the presence of marked TAD borders is a known difference between micro-organisms and multicellular organisms that makes the latter at the same time more complex and interesting. As such, it is fine for me, this is food for thought. I am convinced enough that the model presented in this paper describes well this kind of data.

We thank the reviewer and appreciate his comments throughout this revision process.

Reviewers' comments:

Reviewer #1 (Remarks to the Author):

Referee Report on NCOMMS-17-34203 Statistics of chromatin organization during cell differentiation revealed by heterogeneous cross-linked polymers

I have had a fresh look at the manuscript and the replies of the authors. Unfortunately my opinion has not changed. I like the extension of the RCL, the analytic theory and its verification by the simulations. This can be a very nice paper on its own in a more specialized journal. Nevertheless, I find the connection of the model to chromatin organization not convincing and novel enough for publication in Nature Communications.

The claimed novelty of the work is in showing that such a simple model of only a few random connectors can explain many features - compaction, MSD of SPT, TADs and coarse-grained EP. Although, the difference from the Random Loop Model (RLM) model is in multiple interacting TADs and the permanent nature of the connectors, the compaction and the EP is captured by RLM too and the RCL for a single TAD presented earlier. The dynamics of RLM and the present model is the same on short time scales where the connectors can be considered fixed in the RLM.

A list of other issues:

A) I still find the presentation of the part on the dynamic exponents not convincing. For example,

1. The authors measured MSD for times smaller than 25s, but it was not explained how does this time scale compare to other time scales of the model. At what time scales are various dynamic regimes at play? What is the Rouse time and entanglement time for this model and how does it depend on the model parameters? How far on average each bead diffuses during the 25s and how does that compare to b and to the MRG?
2. Topological constraints and crowding have dramatic effects on polymer dynamics. Based on what grounds were these two effects neglected?
3. MSD is affected by D and b which depend on the

scale and also the location of the crossovers between various dynamic regimes depend on the scale so it is not clear if the measured exponents also depend on the scale and how.

4. From the presented plots it is not clear if the MSD exponent is measured in one scaling regime or over several regimes. Log-log plots that the authors do not want to present would shed light on this.
5. The MSD is measured after the structure relaxes. What is the relaxation time of the structure? Is this biologically relevant?

B) The issue with the number of connectors is not resolved satisfactorily. The two replicas differ in the number of connectors in some cases by about 30%. These kind of issues should not be hidden behind absolute numbers (difference of 5 connectors out of 18). The reason for the difference “intrinsic fluctuations in the statistics of encounter frequencies” sounds vague. As for the HiC and 5C differences in the number of connectors, the data could be coarse-grained to represent the same scale and the results could be directly compared. Last, most likely, the theoretical dependence of the number of connectors on the scale could be extracted, because their distribution is known (uniform).

C) Also the inter-TAD connections are rare in comparison to the intra TAD and their interplay was claimed important in the present work. I was not convinced that the consideration of other subsequent TADs would not change the game significantly.

D) The fitted values of b and their fluctuations are quite large. Why is this reasonable?

To summarize: Although I like some new parts of the work (e.g. comparison with FISH data), my opinion on the manuscript has not changed and I do not recommend its publication in Nature Communications.

Reviewer #2 (Remarks to the Author):

The authors have adequately addressed the 3 reviewers comments.

Response to reviewer 1

I have had a fresh look at the manuscript and the replies of the authors. Unfortunately my opinion has not changed. I like the extension of the RCL, the analytic theory and its verification by the simulations. This can be a very nice paper on its own in a more specialized journal. Nevertheless, I find the connection of the model to chromatin organization not convincing and novel enough for publication in Nature Communications.

Answer: The new FISH data, HiC/5C analysis and reconstruction, the analysis and simulations of polymer model, the analysis of single particle trajectories demonstrate the interdisciplinary nature of our approach, where the goal is to compare chromatin structure across cellular differentiation. We are unaware of any publication showing an equivalent reconstruction approach, revealing both static and dynamic properties of chromatin. We could not find in the reviewer comments a reference that we could use to compare with the present results.

Comment:

The claimed novelty of the work is in showing that such a simple model of only a few random connectors can explain many features - compaction, MSD of SPT, TADs and coarse-grained EP. Although, the difference from the Random Loop Model (RLM) model is in multiple interacting TADs and the permanent nature of the connectors, the compaction and the EP is captured by RLM too and the RCL for a single TAD presented earlier. The dynamics of RLM and the present model is the same on short time scales where the connectors can be considered fixed in the RLM.

Answer: The goal of the present manuscript is not to compete with the RLM model, but to present a procedure of chromatin reconstruction at a given scale, that could possibly be also implemented by RLM polymer model (For example, a new version of the RLM polymer has been explored recently by the group of O. Dulko, unpublished work, for different purpose to account for changes occurring within seconds, which is not the time scale we are studying here). In absence of any need for using RLM in this article, we do not consider this model. In addition, the RLM is not very robust to account for multiple interacting TADs. It is used to account for changes in the chromatin at the time scale of few seconds.

Finally, even in the recent work of A. Pombo, they use a sufficient amount of concentration of binders to be above the fluctuation regime (Answer of Pr. M. Nicodemi at our meeting, 2 weeks ago, <http://www.crm.sns.it/event/426/>).

To conclude, our tests, validation and examination of our model in various cases, including addition of volume exclusion and testing 5C, HiC and validation on FISH data demonstrate that the present model can be used to extract interesting and novel properties of the chromatin including dynamics, that was not done before us.

A list of other issues

Comment:

A) I still find the presentation of the part on the dynamic exponents not convincing. For example: The authors measured MSD for times smaller than 25s, but it was not explained how does this time scale compare to other time scales of the model. At what time scales are various dynamic regimes at play? What is the Rouse time and entanglement time for this model and how does it depend on the model parameters? How far on average each bead diffuses during the 25s and how does that compare to b and to the MRG?

Answer: In the last revision, we mentioned that we computed the MSD during 25s, after relaxation is achieved. Since the polymer is at steady-state, the MSD captures any internal fluctuations that could be generated by thermal noise and the internal modes. In addition, we recall that connectors are fixed, in contrast to the RLM model, so there is in principle no other hidden time scales to account for. There are no mileages to win by comparing our dynamics here with Rouse, because we have shown that the statistical properties are not comparable.

We have now computed the slowest relaxation time scale of the order of 16s-28s for the three TADs (see response below). These time scales are intrinsic, but as we describe below, they are not contributing to the anomalous exponent of the MSD. We show below the MSD curve in the log-log plot and discuss also the space explored by monomers.

Comment:

How far on average each bead diffuses during the 25s and how does that compare to b and to the MRG?

Answer: There is a confusion here about the motion of monomer; they do not diffuse (see the general review of polymer physics: Amitai Holcman, Physics report 2017 for a summary). This situation is the case for all monomers in a polymer model, because they are not characterized by a diffusion, but by an anomalous exponent.

To characterize the space of exploration, we must compute other quantities such as the length of constraint that we introduced in Amitai et al, Cell report 2017 or Hauer et al, Nature Struct Bio, 2017). In that case, here to estimate the space explored, we use the length of constraint L_c for three monomers in each TAD D, E, F: r_{20} , r_{70} , r_{120} , for a single realization, we obtain 0.3, 0.25, 0.26 μm , respectively, which is about twice the simulated MRG of TAD D, E, F: 0.18, 0.13, 0.17 μm , respectively

We have added before the Discussion section this clarification: "Complementary to the anomalous exponent, we estimated the space explored by monomers, by computing the length of constraint L_c [Amitai2017] (computed empirically along a trajectory of N_p points for monomer R as $L_c \approx 1/N \sum_i (R_i - \langle R \rangle)^2$) for three monomers in each TAD D, E, F: r_{20} , r_{70} , r_{120} , for a single realization, we obtain $L_c \approx 0.3, 0.25, 0.26 \mu\text{m}$, respectively, which is about twice the simulated MRG of TAD****

D, E, F: 0.18, 0.13, 0.17 μm , respectively. Thus we conclude that random distributions of fixed connectors can reproduce the large variability of anomalous exponents reported in experimental systems using single locus trajectories, especially for bacteria and yeast genome [Javer2013, Amitai2017] in various conditions."

Comment:

Topological constraints and crowding have dramatic effects on polymer dynamics. Based on what grounds were these two effects neglected?

Answer: Crowding has been captured at least in part by connectors, as we have shown in the revised version for a radius of $c < 40\text{nm}$: ignoring or not crowding give similar results, SI Fig. 3.

Topological constraints are not considered here and we are not aware of any model so far in the recent years that account for that for HiC reconstruction, because topological constraints cannot be reconstructed from the encounter probability matrix. In any cases, entanglements have NO *dramatic* effect in the scale of 3kb and above.

Comment:

MSD is affected by D and b which depend on the scale and also the location of the crossovers between various dynamic regimes depend on the scale so it is not clear if the measured exponents also depend on the scale and how.

Answer: How the MSD exponent depends on the scale is a generic question in polymer physics that goes beyond this manuscript: we recall it has been already solved in some cases such as Rouse, beta-polymer and its generalization (Amitai et al., PRE 2013).

In addition, contrary to what the reviewer says, the value of alpha is not affected by D or b: for Rouse, it is constant =0.5 and for the beta-polymer, it depends only on the force between connectors. Here, we have shown in Fig.5 that the anomalous exponent depends on the distribution of connectors. In the mean-field approximation, we recall that the mean exponent is 0.5 with does not depend on the scale, D and b. A thorough exploration of this relationship in the non mean-field case, is very interesting, but is beyond the scope of this current manuscript.

We have added a sentence: "We note that the value of anomalous exponent α does not depend on the diffusion coefficient D or b, as known in various other polymer models such as for Rouse or β -polymers [Amitai2013, Amitai2017b]. We have shown here (Fig.5) that the anomalous exponent crucially depends on the number and the distribution of connectors (see also Fig. S2D). In the mean-field approximation, the mean exponent is 0.5, which does not depend on the polymer scale. Finding the exact relation between the numbers of connectors for a specific configuration (not in the mean-field case) remains challenging and relevant to reconstruct the local connectors environment from measured anomalous exponents."

Comment:

From the presented plots it is not clear if the MSD exponent is measured in one scaling regime or over several regimes. Log-log plots that the authors do not want to present would shed light on this.

Answer: As shown below, the log-log plot does not provide more information than the one presented in the manuscript.

Fig: log-log plot of MSD for a given realization.

These plots confirm our normal analysis: the MSD exponent is estimated during the 25s (after polymer relaxation) using the mean-square procedure and we do not see here different regimes probably because the chromatin is at steady-state: the dotted lines show the fit approximation. **We have now added this fig as a subfig of Fig. S3 C, right panel.**

Comment:

The MSD is measured after the structure relaxes. What is the relaxation time of the structure? Is this biologically relevant?

Answer: The main relaxation time of the polymer is associated to the first non-vanishing eigenvalue of the mean-field system (expression in SI Eq. 23). Since we run simulations past relaxation time (~few 10000 of simulation steps, which is around few hundreds of seconds >> 25s), thus we are past relaxation time and the polymer has reach equilibrium.

Finally, using $b=0.2$, $D=0.008$, and N_c for polymer as in Fig 3 and 5., in Eq. SI 23, we have that the relaxation times are: TAD D: 28s, TAD E: 12s, TAD F: 16s.

We have added" Using $b=0.2\mu\text{m}$, $D=0.008\mu\text{m}^2/\text{s}$, and N_c for polymers as described in Fig. 3 and 5, and Eq. SI 23, we obtain for the relaxation times: for TAD D, E and F, $\tau_D\approx 28\text{s}$; $\tau_E\approx 12\text{s}$; $\tau_F\approx 16\text{s}$. All statistical quantities were computed here from simulations after we waited 10,000 time steps, which is much more than these relaxation times, ensuring that the polymer model has reached equilibrium. However the internal monomer position fluctuations due to thermal noise can contribute to the dynamics of each monomer, reported in the MSD (which is a second order moment)".

Comment:

The issue with the number of connectors is not resolved satisfactorily. The two replicas differ in the number of connectors in some cases by about 30%. These kind of issues should not be hidden behind absolute numbers (difference of 5 connectors out of 18). The reason for the difference “intrinsic fluctuations in the statistics of encounter frequencies” sounds vague. As for the HiC and 5C differences in the number of connectors, the data could be coarse-grained to represent the same scale and the results could be directly compared. Last, most likely, the theoretical dependence of the number of connectors on the scale could be extracted, because their distribution is known (uniform).

Answer: Contrary to what the reviewer said, we did not hide anything: we showed clearly the differences between both replica, which reflect the intrinsic differences between 5C and HiC. Such differences are well expected from the experimental methods Nora et al. Nature, 2012 and Bonev et al, 2017 (already mentioned in the manuscript).

These experimental differences are precisely discussed when we spoke about intrinsic fluctuations (main text, p.7, right column, second paragraph). The beauty is that our model captures well these fluctuations, revealed by the differences in the connectors. The suggestion to coarse-grain 5C and HiC "at the same scale" is not clear, because this is exactly what we have done in our manuscript: both data sets are coarse-grained at 10kb (See Figure 4). Finally, we made a specify effort during this revision process to link the number of connectors to the scale resolution. We highlighted our approach in the last new Fig. 6, that we added at that time.

Comment:

Also the inter-TAD connections are rare in comparison to the intra TAD and their interplay was claimed important in the present work. I was not convinced that the consideration of other subsequent TADs would not change the game significantly.

Answer: The reconstruction method we developed here is generic and can be used for any amount of TADs. Thus the consideration of the reviewer about adding or not subsequent TADs is not limiting. At this stage, this consideration is irrelevant for the understanding of genome dynamics, because long-range connectors above 1MB are extremely rare and the immediate neighborhood of TADS probably play the most important part.

Comment:

The fitted values of b and their fluctuations are quite large. Why is this reasonable?

>Answer: This new comment does not account for our previous answer: we previously explained that the parameter b is an input parameter chosen ($=0.2 \mu m$ in our case). In general, the value of b is not fitted, contrary to what the reviewer is suggesting. It is calibrated to scale of the data. We have explained (see SI Eq. 67) that b can be re-scaled from a known value in one scale to another coarse-graining scale.

Moreover, we also explained in the previous revision that b is not the inter-distance between monomers, because the connectors reduce these inter-distances.

But even in that case, our new analysis about FISH data shows that we could recover the experimental distances, so to conclude, our analysis is more than reasonable, it just works very well.

Reviewers' comments:

Reviewer #1 (Remarks to the Author):

Referee Report on NCOMMS-17-34203 Statistics of chromatin organization during cell differentiation revealed by heterogeneous cross-linked polymers. (4)

I looked again at the replies of the authors and the manuscript. I do not recommend its publication.

My main objection is about the MSD. Contrary to what the authors say, to my opinion, the newly-added figure of MSD vs. time in log-log scale does provide new information, namely

- The presented data of the MSD of any of the three TADs cannot be fitted to a single power-law. In the log-log scale a single power-law would correspond to a straight line. The colored curves are not straight lines. The fits are obviously not catching the real nature of the MSDs and therefore the extracted exponents do not make much sense.

What we see here is most likely a crossover between various dynamic regimes as I suggested in the previous round and that is the reason why I asked about the estimates of the different time-scales and this log-log plot. The different regimes can be seen as initial steeper power-law that smoothly crosses over to another one less steep at later times (In the log-log plot this could be maybe from 10s).

- The precise location (i.e. at what time a given regime starts to dominate) of these crossover points between the regimes will depend on the values of D and b or the level of coarse-graining. Therefore if one does not take this into account and blindly fit the MSD vs. t with one single power-law (as is

done in this work), the exponent will depend on the values of D and b . The value of the exponent α is independent of D and b only if a single regime is fitted, which is not the case in the present manuscript.

There are also other issues:

1. My question about topological constraints was answered without any supporting evidence, but with a capitalized assertion that they do not have a dramatic effect above the scale of $3kb$. For example the work of Rosa and Everaers PloS (2008) suggests otherwise, especially on large scales as also reviewed by Mirny Chrom. Res. (2011) and recently by Sazer and Schiessel Traffic (2017).
2. The authors say that the immediate neighborhood of TADs play the most important part for the correct results. If we agree on this, we can trust the data for the TAD in between the two other TADs, but, we should not necessarily trust the data for the two TADs, because all of their most immediate neighbors are not considered in the simulations.

Besides other persistent problems, the treatment of the dynamics is not correct. With, or without the dynamics part, I do not find a significant paradigm shift in the present work to be publishable in this journal. In the preceding rounds of the review process I have highlighted good and interesting aspects of this work, which should be published in a more specialized journal.

REVIEWERS' COMMENTS:

Reviewer #1 (Remarks to the Author):

Please find my comments in the attached pdf file.

The presented data of the MSD of any of the three TADs cannot be fitted to a single power law.

>ANSWER: it very well known that the MSD of locus belonging to a polymer, that deviates from classical Rouse, cannot be fitted by a single power law during the entire time interval due to the polymer modes, rather an interval of time has to be identified. This is not particular to the case at hand here. We have developed some of the theory in the following references for the beta and cross-linked polymer mode

-A. Amitai, D. Holcman, beta-model application to DNA modeling in the nucleus, *Phys. Rev E*. 88, 052604 (2013)

- O. Shukron D. Holcman, Polymer model reconstruction from chromosomal capture data and stochastic simulations of transient encounters, *PLoS Comput Biol* 13 (4), e1005469 2017.

-Shukron D. Holcman, Statistics of randomly cross-linked polymer models to interpret chromatin conformation capture data, *Physical Review E* 96, 012503 (2017).

A summary of the exact properties of the MSD can be found in our high impact review:

A. Amitai D. Holcman, Polymer physics of nuclear organization and function, *Physics Report*, 678, 1–83 (2017).

In the log-log scale a single power-law would correspond to a straight line. The colored curves are not straight lines. The fits are obviously not catching the real nature of the MSDs and therefore the extracted exponents do not make much sense. What we see here is most likely a crossover between various dynamic regimes as I suggested in the previous round and that is the reason why I asked about the estimates of the different time-scales and this log-log plot. The different regimes can be seen as initial steeper power-law that smoothly crosses over to another one less steep at later times (In the log-log plot this could be maybe from 10s).

>ANSWER:the MSD fit allow to approximate the anomalous exponent in a given time interval. Contrary to what the review seems to think, a fit is not about “catching the real nature” of anything, but just to estimate parameters from a given model, that should be reproducible. We have observed here some deviations from a single power law, but the interpretation of this deviation is unclear. So far, there are no general method in the literature to further characterize this crossover.

In the log-log plot this could be maybe from 10s.

>ANSWER: As the reviewer said: using conditional language: the conclusion from log-log plot is very hypothetical. We have added a sentence saying that:

“the deviation of the straight line in the log-log fit of the MSD may suggest that there are possibly two time scales in the 30s interval, one below 10s. The origin of these scales might be due to short and long range connection.

The precise location (i.e. at what time a given regime starts to dominate) of these crossover points between the regimes will depend on the values of D and b or the level of coarse-graining. Therefore if one does not take this into account and blindly fit the MSD vs. t with

one single power-law (as is done in this work), the exponent will depend on the values of D and b . The value of the exponent α is independent of D and b only if a single regime is fitted, which is not the case in the present manuscript.

>ANSWER: the anomalous exponent α characterizes the deviation from classical Brownian motion or Rouse polymer, where it is clearly independent from D and b . In general, the anomalous exponent should be defined from physical principles and not from empirical data (see Metzler and Klater 2001 Phys Rep. for the general theory and Amitai- Holcman Phys. Rep. 2017 for the case of polymer model). This is the classical approach in the physical sciences.

Using cross-over between regimes to define α empirically does not make much sense and obvious leads to the confusion that this coefficient might depend on D or b . It should not. Here we gave the precise condition under which we approximated the MSD curve by a power law and any deviation can be used for speculations at this stage. As mentioned by the reviewer, the time interval where the fit is done depends on the first eigenvalue of the connectivity matrix (reciprocal of the first relaxation time), which clearly depends on b and D . Contrary to what reviewer may have understood, we do not here “blindly fit the MSD vs. t ”, but we use the protocols used in SPTs analysis in order to compare our results with experimental data. We have already mentioned several time this point in the previous revision and refer to our own experience in this domain:

- A. Amitai, A. Seeber, S. M. Gasser D. Holcman, Statistical polymer simulation distinguishes DNA double-strand break movement due to chromatin expansion and nuclear oscillation, *Cell Report* 18(5):1200-1214 2017.

- M. Hauer A. Seeber R.Thierry A Amitai, J Eglinger D. Holcman T. Owen-Hughes S. Gasser, Histone degradation in response to DNA damage triggers general chromatin decompaction, *Nat Struct. Bio*, 24(2):99-107. 2017.

- A. Amitai, M Toulouze K. Dubrana D. Holcman, Analysis of single locus trajectories for extracting in vivo chromatin tethering interactions, *PLoS Comput Biol* 11 (8), e1004433 2015.

There are also other issues:

1. My question about topological constraints was answered without any supporting evidence, but with a capitalized assertion that they do not have a dramatic effect above the scale of 3kb. For example the work of Rosa and Everaers PloS (2008) suggests otherwise, especially on large scales as also reviewed by Mirny Chrom. Res. (2011) and recently by Sazer and Schiessel Traffic (2017).

>ANSWER: The goal of the manuscript is not to elaborate on hypothetical topological constraints that were suggested by the reviewers. We simply mentioned them briefly, but we do not think that there are bringing any deeper understanding here of the TAD reorganization across cell differentiation. The reviewer is not suggesting any specific interpretation.

2. The authors say that the immediate neighbourhood of TADs play the most important part for the correct results.

>ANSWER: This one sentence statement of the reviewer is not clear to us because it is quite vague and we do not see which “correct results” the referee is referring to. There is no absolute truth in any of the reconstruction methods of chromatin proposed so far by us and others.

If we agree on this,

>ANSWER: We clearly do not.

we can trust the data for the TAD in between the two other TADs, but, we should not necessarily trust the data for the two TADs, because all of their most immediate neighbors are not considered in the simulations.

>ANSWER:

Contrary to what the reviewer is mentioning, we are actually considering immediate neighbors that we called meta-TAD in Fig. 2A and thus we specifically estimated the number of connectors to match the EP matrix.

Besides other persistent problems, the treatment of the dynamics is not correct.

>ANSWER: It is not clear what is incorrect in the treatment of the dynamics, but connectors are not dynamics but static. This statement and not justified.

With, or without the dynamics part, I do not find a significant paradigm shift in the present work to be publishable in this journal. In the preceding rounds of the review process I have highlighted good and interesting aspects of this work, which should be published in a more specialized journal.

>ANSWER:

The paradigm shift that we started 10 years ago has already been accepted by the leading physics community M. Tamm, M. Consensinno, but also the experimental community such as S. Gasser, K. Dubrana, A. Spakowitz, Y. Garini and many others, as recently discussed in the conference we organized in Pisa:

<http://www.crm.sns.it/event/426/>

The Fractal globule promoted by L. Mirny is clearly insufficient to explain TADs, their specific properties and the local single particle trajectory statistics. Recent works in 2018 have now clearly proved that cross-linker such as the cohesin, condensin, lamin, are key component in the genome organization and TADs formation. Modifying TAD borders affect genome expression. The present manuscript presents for the first time an approach that reveals how to get more information from HiC and to connect with SPTs, which is clearly crucially missing in the field.